# Fundamental Bias in Inverting Random Sampling Matrices with Application to Sub-sampled Newton

Chengmei Niu [1]  Zhenyu Liao [1]  Zenan Ling [1]  Michael W. Mahoney [2]

## Abstract

A substantial body of work in machine learning (ML) and randomized numerical linear algebra (RandNLA) has exploited various sorts of random sketching methodologies, including random sampling and random projection, with much of the analysis using Johnson–Lindenstrauss and subspace embedding techniques. Recent studies have identified the issue of *inversion bias* – the phenomenon that inverses of random sketches are *not* unbiased, despite the unbiasedness of the sketches themselves. This bias presents challenges for the use of random sketches in various ML pipelines, such as fast stochastic optimization, scalable statistical estimators, and distributed optimization. In the context of random projection, the inversion bias can be easily corrected for dense Gaussian projections (which are, however, too expensive for many applications). Recent work has shown how the inversion bias can be corrected for sparse sub-gaussian projections. In this paper, we show how the inversion bias can be corrected for random sampling methods, both uniform and non-uniform leverage-based, as well as for structured random projections, including those based on the Hadamard transform. Using these results, we establish problem-independent local convergence rates for sub-sampled Newton methods.

## 1. Introduction

Randomized numerical linear algebra (RandNLA) significantly reduces computation, communication, and/or storage overheads by using randomness as an algorithmic resource.

[1]School of Electronic Information & Communications, Huazhong University of Science & Technology, Wuhan, Hubei, China. [2]ICSI, LBNL, and Department of Statistics, University of California, Berkeley, USA. Correspondence to: Zhenyu Liao <zhenyu_liao@hust.edu.cn>.

*Proceedings of the 42$^{nd}$ International Conference on Machine Learning*, Vancouver, Canada. PMLR 267, 2025. Copyright 2025 by the author(s).

As a pivotal technique in RandNLA, random sketching – which encompasses both random projection and random sampling – is becoming increasingly critical in many modern large-scale machine learning (ML) applications.

More precisely, for a tall matrix $\mathbf{A} \in \mathbb{R}^{n \times d}$ with $n \gg d$, random projection proposes to obtain a sketch $\tilde{\mathbf{A}} \in \mathbb{R}^{m \times d}$, of size $m \ll n$ of $\mathbf{A}$ by randomly and linearly combining the rows of $\mathbf{A}$; while random sampling, on the other hand, carefully selects a small subset (of size $m$ say) of the rows $\mathbf{A}$ and rescales them to obtain $\tilde{\mathbf{A}} \in \mathbb{R}^{m \times d}$. For both approaches, it follows from Johnson–Lindenstrauss (JL) type analysis (Johnson & Lindenstrauss, 1984) that the random sketch $\tilde{\mathbf{A}}$ can be used as a "proxy" of $\mathbf{A}$ in many downstream ML tasks, e.g., $\tilde{\mathbf{A}}^\top \tilde{\mathbf{A}} \approx \mathbf{A}^\top \mathbf{A}$ are close in some sense with high probability, even with $m \ll n$. This leads to a significant boost in the running time, communication time, and/or memory cost, for many numerical methods (Drineas et al., 2006b; 2011; 2012; Avron et al., 2017; Dereziński & Mahoney, 2021; Lacotte & Pilanci, 2022). See also Mahoney (2011); Halko et al. (2011); Woodruff (2014); Drineas & Mahoney (2018); Martinsson & Tropp (2020); Dereziński & Mahoney (2021); Murray et al. (2023); Dereziński & Mahoney (2024) and reference therein for an overview of RandNLA and the applications in modern ML.

Despite this promising "complexity-accuracy" trade-off achieved with random sketching, in many ML pipelines ranging from linear/ridge regression to scalable statistical estimation and fast stochastic optimization, the object of direct interest is the sketched matrix *inverse* $(\tilde{\mathbf{A}}^\top \tilde{\mathbf{A}} + \mathbf{C})^{-1}$ for some (perhaps all-zeros or diagonal) positive semi-definite (p.s.d.) $\mathbf{C}$ (instead of $\tilde{\mathbf{A}}^\top \tilde{\mathbf{A}}$ itself). Since the matrix inverse is a nonlinear operator, the fact that $\tilde{\mathbf{A}}^\top \tilde{\mathbf{A}}$ is an unbiased or nearly unbiased estimator of $\mathbf{A}^\top \mathbf{A}$ (i.e., $\mathbb{E}[\tilde{\mathbf{A}}^\top \tilde{\mathbf{A}}] = \mathbf{A}^\top \mathbf{A}$, which is a major guide in forming the sketch $\tilde{\mathbf{A}}$) does *not*, in general, imply the unbiasedness of its inverse, i.e., $\mathbb{E}[(\tilde{\mathbf{A}}^\top \tilde{\mathbf{A}})^{-1}] \not\approx (\mathbf{A}^\top \mathbf{A})^{-1}$. This phenomenon of *inversion bias* has been long known in the literature: in the case of Gaussian random projection with $\tilde{\mathbf{A}} = \mathbf{S}\mathbf{A}$ for $\mathbf{S} \in \mathbb{R}^{m \times n}$ having i.i.d. $\mathcal{N}(0, 1/m)$ entries, $(\tilde{\mathbf{A}}^\top \tilde{\mathbf{A}})^{-1}$ is known to follow the *inverse Wishart distribution* (Haff, 1979) with $\mathbb{E}[(\tilde{\mathbf{A}}^\top \tilde{\mathbf{A}})^{-1}] = \frac{m}{m-d-1}(\mathbf{A}^\top \mathbf{A})^{-1}$ for $m > d + 1$. However, much less is known beyond the Gaussian setting. Build-

ing upon recent progress in non-asymptotic random matrix theory (RMT), it has recently been shown by Dereziński et al. (2021b) that a similar inversion bias holds, and that it can be corrected, with $\mathbb{E}[(\tilde{\mathbf{A}}^\top \tilde{\mathbf{A}})^{-1}] \approx \frac{m}{m-d}(\mathbf{A}^\top \mathbf{A})^{-1}$, in the sense of partial order of p.s.d. matrices, for sub-gaussian *and* the so-called LEverage Score Sparsified (LESS) projections. The latter can be of greater practical interest than, e.g., dense Gaussian or sub-gaussian projections, as it is significantly sparser and can thus be evaluated more efficiently.

This precise characterization of inversion bias has direct implications for RandNLA and ML. As a telling example, Dereziński et al. (2021a) applied LESS sketches to Newton Sketch (Pilanci & Wainwright, 2017) and showed that this Newton-LESS approach enjoys almost the same local convergence rates as Newton Sketch with dense Gaussian projections. This leads to a significantly better "complexity-convergence" trade-off than the vanilla Gaussian projections in stochastic second-order optimization methods.

### 1.1. Our Contributions

In this paper, we consider the inversion bias of random sampling, including uniform and non-uniform sampling, as well as *structured* random projections such as the Subsampled Randomized (Walsh-)Hadamard Transform (SRHT) (Ailon & Chazelle, 2006). The analysis framework in Dereziński et al. (2021a;b) does *not* apply (to get non-vacuous results; see, e.g., Proposition 2.8 and Remark 2.9 below). Instead, we exploit novel and non-trivial connections between non-asymptotic RMT and RandNLA to show that this inversion bias can be precisely characterized and numerically corrected. We also show how this inversion bias result can be used to improve the local convergence rates of the popular sub-sampled Newton (SSN) method (Yao et al., 2018; Roosta-Khorasani & Mahoney, 2019; Xu et al., 2020).

Our main contributions can be summarized as follows.

1. We provide precise characterization of the inversion bias for general random sampling (in Theorem 3.1) and the corresponding de-biased approach (in Proposition 3.2). The proposed analysis and debiasing technique hold for exact and approximate leverage-based sampling (Corollary 3.4), as well as the structured SRHT (Corollary 3.7) as special cases.

2. With this precise inversion bias result, we further establish (in Theorem 4.3), the first *problem-independent* local convergence rates for sub-sampled Newton that approximately matches the dense Gaussian Newton Sketch scheme. Numerical results are provided in Section 5 to support these findings.

### 1.2. Related Work

**Inversion bias.** Given a matrix $\mathbf{A} \in \mathbb{R}^{n \times d}$, the matrix inverse $(\mathbf{A}^\top \mathbf{A})^{-1}$ is fundamental in ML, numerical computation, and statistics. Examples include linear functions $(\mathbf{A}^\top \mathbf{A})^{-1}\mathbf{y}$ that are crucial to Newton's methods (Boyd & Vandenberghe, 2004), quadratic forms $\mathbf{a}_i^\top (\mathbf{A}^\top \mathbf{A})^{-1}\mathbf{a}_i$ (with $\mathbf{a}_i^\top$ the $i^{th}$ row of $\mathbf{A}$) in computing matrix leverage scores (Drineas et al., 2012), and trace forms, $\operatorname{tr} \mathbf{L}(\mathbf{A}^\top \mathbf{A})^{-1}$ for some given $\mathbf{L}$, of interest in uncertainty quantification (Kalantzis et al., 2013) and experimental designs (Pukelsheim, 2006). In the case of tall matrices with $n \gg d$, random sketching applies to efficiently reduce the computational overhead of $(\mathbf{A}^\top \mathbf{A})^{-1}$, by using a sketch $\tilde{\mathbf{A}} = \mathbf{SA}$ of $\mathbf{A}$ for random matrix $\mathbf{S} \in \mathbb{R}^{m \times n}$ with $m \ll n$. It has been shown recently (Dereziński et al., 2020; 2021a;b) that these sketched inverses are *biased* for *unstructured* sub-gaussian $\mathbf{S}$, with $\mathbb{E}[(\tilde{\mathbf{A}}^\top \tilde{\mathbf{A}})^{-1}] \not\approx (\mathbf{A}^\top \mathbf{A})^{-1}$. In this paper, we consider the (often practically more interesting) case of *structured* random matrix $\mathbf{S}$, including random sampling matrices (Definition 2.1) and randomized Hadamard transforms (Definition 3.6).

Different from our approach that explicitly modifies the sketch to correct the bias, another line of work proposes to use shrinkage-based correction techniques (Zhang & Pilanci, 2023; Romanov et al., 2024).

**Random sampling.** Random sampling is at the core of RandNLA (Drineas et al., 2006a; Mahoney, 2011; Ma et al., 2015; Drineas & Mahoney, 2016; Dereziński & Mahoney, 2021; 2024), and it plays a central role in fast matrix multiplication (Drineas et al., 2006a), approximate regression (Drineas et al., 2006b), and low-rank approximation (Cohen et al., 2017), to name a few. It is of particular interest in scenarios where the dataset is massive and cannot be stored and/or computed on a single machine, e.g., the census data (Wang et al., 2018) and online network data (Deng et al., 2024). See Definition 2.1 below for a formal definition of random sampling and discussions thereafter for commonly-used sampling schemes including (exact and approximate) leverage score sampling (Mahoney, 2011; Cohen et al., 2017), shrinkage leverage sampling (Ma et al., 2015), as well as optimal subsampling (Wang et al., 2018; Wang & Ma, 2021; Yu et al., 2022; Ma et al., 2022). In this paper, instead of providing classical JL and subspace embedding-type results on random sampling, we precisely characterize (and correct) the inversion bias for a variety of commonly-used random sampling schemes.

**Sub-sampled Newton.** Sub-sampled Newton (SSN) methods propose to approximate the Hessian in Newton's method using a small subset of samples, and they have been extensively studied within the fields of ML, RandNLA, and optimization (Xu et al., 2016; Bollapragada et al., 2019;

Roosta-Khorasani & Mahoney, 2019; Xu et al., 2020; Ye et al., 2021). Although these fast optimization methods are easy to implement, their convergence rates are challenging to analyze. Existing results often depend on the Hessian condition number or the Lipschitz constant and fall short of, e.g., the *problem-independent* convergence rates achieved by sub-gaussian Newton Sketch (Lacotte & Pilanci, 2019; Dereziński et al., 2021a). In this paper, we establish the first *problem-independent* local convergence rates for SSN that closely align with Newton Sketch. This addresses the convergence guarantee gap identified in Iterative Hessian Sketch (Pilanci & Wainwright, 2016) for random sampling.

**Random matrix theory (RMT).** RMT studies the (limiting) eigenspectra of large-dimensional random matrices (Anderson et al., 2010) and finds its applications in signal processing and communication (Couillet & Debbah, 2011), statistical finance (Plerou et al., 2002), optimization (Paquette et al., 2021; 2023), and more recently in large-scale ML (Pennington & Worah, 2017; Fan & Wang, 2020; Mei & Montanari, 2022; Couillet & Liao, 2022). A recent line of work (Liao et al., 2020; 2021; Liao & Mahoney, 2021) has highlighted non-trivial connections between RMT and RandNLA that this paper further develops.

## 1.3. Notations and Organization of the Paper

We denote scalars by lowercase letters, vectors by bold lowercase, and matrices by bold uppercase. For a matrix $\mathbf{A} \in \mathbb{R}^{n \times d}$, we denote $\mathbf{A}^\top$, $\mathbf{a}_i^\top \in \mathbb{R}^d$, and $\|\mathbf{A}\|$ the transpose, $i^{th}$ row, and spectral norm of $\mathbf{A}$, respectively. We denote $\mathbf{A} \preceq \mathbf{B}$ if $\mathbf{B} - \mathbf{A}$ is p.s.d., and use $\mathbf{I}_d$ for the identity matrix of size $d$. For a vector $\mathbf{x} \in \mathbb{R}^d$ and a matrix $\mathbf{B} \in \mathbb{R}^{d \times d}$, we denote $\|\mathbf{x}\|_{\mathbf{B}} \equiv \sqrt{\mathbf{x}^\top \mathbf{B} \mathbf{x}}$, with the convention $\|\mathbf{x}\| = \|\mathbf{x}\|_{\mathbf{I}_d}$. For a random variable $x$, we denote $\mathbb{E}[x]$ the expectation of $x$ and $\mathbb{E}_\zeta[x]$ the expectation of $x$, conditional on the event $\zeta$. We use $\Theta(\cdot)$, and $O(\cdot)$ notations as in standard computer science literature.

The remainder of this paper is organized as follows. Section 2 presents preliminaries on random sampling and a *coarse-grained* characterization of its inversion bias, by directly (and naively) adapting the proof approach from Dereziński et al. (2021b) (which turns out to be vacuous in our setting). Section 3 delivers a *fine-grained* analysis of this inversion bias and proposes an efficient non-vacuous debiasing approach. Section 4 demonstrates how these technical results apply to establish *problem-independent* local convergence rates for SSN. Section 5 provides numerical results that support our theoretical findings. Section 6 provides a conclusion, summarizing our findings and discussing future perspectives. Additional material can be found in the appendices.

## 2. Preliminaries on Random Sampling

In this section, we introduce a few definitions that will be used in the remainder of this paper.

**Definition 2.1** (Random sampling). For a matrix $\mathbf{A} \in \mathbb{R}^{n \times d}$ with $n \geq d$, a sketch $\tilde{\mathbf{A}} \in \mathbb{R}^{m \times d}$ of $\mathbf{A}$ can be constructed by randomly sampling *with replacement* $m$ from the $n$ rows of $\mathbf{A}$ with an *importance sampling distribution*, $\{\pi_i\}_{i=1}^n$, $\sum_{i=1}^n \pi_i = 1$, and then rescaling by $1/\sqrt{m\pi_i}$. This can be expressed as $\tilde{\mathbf{A}} = \mathbf{S}\mathbf{A}$, with *random sampling matrix* $\mathbf{S} \in \mathbb{R}^{m \times n}$ having only one nonzero entry per row.

Definition 2.1 includes commonly-used random sampling schemes such as uniform sampling (with $\pi_i = 1/n$), row-norm-based sampling (with $\pi_i = \|\mathbf{a}_i\|^2/(\sum_{i=1}^n \|\mathbf{a}_i\|^2)$), exact or approximate leverage and ridge leverage score sampling (Mahoney, 2011; El Alaoui & Mahoney, 2015) defined below, as well as a mix between them, e.g., the shrinkage leverage sampling (Ma et al., 2015).

**Definition 2.2** (Leverage score sampling, Mahoney (2011)). For a matrix $\mathbf{A} \in \mathbb{R}^{n \times d}$ of rank $d$ with $n \geq d$ and a p.s.d. matrix $\mathbf{C} \in \mathbb{R}^{d \times d}$, the $i^{th}$ *leverage score* $\ell_i^{\mathbf{C}}$ of $\mathbf{A}$ given $\mathbf{C}$, is defined as $\ell_i^{\mathbf{C}} = \mathbf{a}_i^\top (\mathbf{A}^\top \mathbf{A} + \mathbf{C})^{-1} \mathbf{a}_i, i \in \{1, \ldots, n\}$. The exact leverage score sampling refers to the random sampling approach in Definition 2.1 with $\pi_i = \ell_i^{\mathbf{C}}/d_{\text{eff}}$, for $d_{\text{eff}} = \sum_{i=1}^n \ell_i^{\mathbf{C}}$ the *effective dimension* of $\mathbf{A}$ given $\mathbf{C}$.

The leverage score sampling has been extensively studied in RandNLA and ML. By taking $\mathbf{C} = \mathbf{0}_d$ in Definition 2.2, we obtain the standard leverage scores (Mahoney, 2011; Drineas et al., 2012); and by taking $\mathbf{C} = \lambda\mathbf{I}_d$, we obtain the $\lambda$-ridge leverage scores (El Alaoui & Mahoney, 2015). Given $\mathbf{A} \in \mathbb{R}^{n \times d}$, its leverage scores can be approximately computed in $O(\text{nnz}(\mathbf{A}) \log n + d^3(\log d)^2 + d^2 \log n)$ time, for $\text{nnz}(\mathbf{A})$ the number of non-zero entries in $\mathbf{A}$, see Drineas et al. (2012); Clarkson & Woodruff (2017); Cohen et al. (2017).

We also introduce an "approximation factor" to measure the extent to which one importance sampling distribution approximates another importance sampling distribution (the exact leverage score distribution, in our case).

**Definition 2.3** (Importance sampling approximation factor). For a matrix $\mathbf{A} \in \mathbb{R}^{n \times d}$ with $n \geq d$, a p.s.d. matrix $\mathbf{C} \in \mathbb{R}^{d \times d}$, and a random sampling matrix $\mathbf{S} \in \mathbb{R}^{m \times n}$ as in Definition 2.1, with importance sampling distribution $\{\pi_i\}_{i=1}^n$, the (min and max) *importance sampling approximation factors* of the random sampling scheme $\mathbf{S}$ is defined as the pair $(\rho_{\min}, \rho_{\max})$, with $\rho_{\min} \equiv \min_{1 \leq i \leq n} \ell_i^{\mathbf{C}}/(\pi_i d_{\text{eff}})$ and $\rho_{\max} \equiv \max_{1 \leq i \leq n} \ell_i^{\mathbf{C}}/(\pi_i d_{\text{eff}})$.

For us, the importance sampling approximation factors $(\rho_{\min}, \rho_{\max})$ in Definition 2.3 provide qualitative characterization on how the random sampling scheme under study

differs from the *exact* leverage score sampling in Definition 2.2. This extends the classical notion of sampling approximation factor in Drineas et al. (2006a) to include both the maximum and the minimum. While prior work primarily focuses on the max factor, here we focus on the inversion bias, where the min factor also plays a natural and significant role; see below in Section 3 and also Ma et al. (2015), who first noted the importance of the min factor in statistical style analysis. By the (generalized) median inequality, we have $\rho_{\min} \leq 1 \leq \rho_{\max}$, with equality for exact leverage score sampling.

It follows from Definition 2.1 that $\mathbb{E}[\tilde{\mathbf{A}}^\top \tilde{\mathbf{A}}] = \mathbf{A}^\top \mathbf{A}$, so that the randomly sampled matrix $\tilde{\mathbf{A}}^\top \tilde{\mathbf{A}}$ is an *unbiased estimator* of the true $\mathbf{A}^\top \mathbf{A}$. This, together with controls on the higher-order moments of $\tilde{\mathbf{A}}^\top \tilde{\mathbf{A}}$, allows one to conclude that $\tilde{\mathbf{A}}^\top \tilde{\mathbf{A}}$ fluctuates, with high probability and within a small "distance," around the true $\mathbf{A}^\top \mathbf{A}$ of interest. This can be made precise using the relative error approximation (for scalars and matrices) defined as follows.

**Definition 2.4** (Relative error approximation). For a non-negative scalar $\tilde{x} \geq 0$, we say $\tilde{x}$ is an $\epsilon$-approximation of (another scalar) $x$, denoted $\tilde{x} \approx_\epsilon x$, if

$$(1 + \epsilon)^{-1} x \leq \tilde{x} \leq (1 + \epsilon) x. \tag{1}$$

For $\tilde{x}$ being random, we say $\tilde{x}$ is an $(\epsilon, \delta)$-approximation of $x$ if (1) holds with probability at least $1 - \delta$. Similarly, for a p.s.d. matrix $\tilde{\mathbf{X}}$, we say $\tilde{\mathbf{X}}$ is an $\epsilon$-approximation (or an $(\epsilon, \delta)$-approximation when being random) if

$$\tilde{\mathbf{X}} \approx_\epsilon \mathbf{X} \Leftrightarrow (1 + \epsilon)^{-1} \mathbf{X} \preceq \tilde{\mathbf{X}} \preceq (1 + \epsilon) \mathbf{X}. \tag{2}$$

*Remark* 2.5 (Subspace embedding). For $\tilde{\mathbf{A}}$ a sketch of $\mathbf{A}$, the property that $\tilde{\mathbf{A}}^\top \tilde{\mathbf{A}} \approx_\epsilon \mathbf{A}^\top \mathbf{A}$ holds with probability $1 - \delta$ is known as $(\epsilon, \delta)$-*subspace embedding* in RandNLA. This concept was introduced by Drineas et al. (2006b); see also Mahoney (2011) for a history. It was subsequently used in data-oblivious form by Sarlós (2006); Drineas et al. (2011), and then popularized in data-oblivious form (and mis-attributed to Sarlós (2006)) by Woodruff (2014). It plays a central role in the statistical characterization of random sketching techniques.

The focus of this paper is to go beyond the subspace embedding-type results in Definition 2.4 and Remark 2.5, and to assess the *inversion bias* of the form $\mathbb{E}[(\tilde{\mathbf{A}}^\top \tilde{\mathbf{A}} + \mathbf{C})^{-1}]$ (versus the true inverse $(\mathbf{A}^\top \mathbf{A} + \mathbf{C})^{-1}$). To this end, we need the following measure of unbiased estimators.

**Definition 2.6** (Unbiased estimator). We say a random p.s.d. matrix $\tilde{\mathbf{X}}$ is an $(\epsilon, \delta)$-unbiased estimator of $\mathbf{X}$ if, conditioned on an event $\zeta$ that happens with probability at least $1 - \delta$,

$$(1 + \epsilon)^{-1} \mathbf{X} \preceq \mathbb{E}_\zeta[\tilde{\mathbf{X}}] \preceq (1 + \epsilon) \mathbf{X}, \text{ and } \tilde{\mathbf{X}} \preceq O(1) \mathbf{X}. \tag{3}$$

Note that the error parameter $\epsilon$ in subspace embeddings quantifies spectral approximation error, whereas the no-

tion of "unbiasedness" specifically refers to inversion bias. Furthermore, while subspace embeddings automatically ensure $\epsilon$-unbiasedness up to the *same* level of error $\epsilon$, they are generally *not* guaranteed to remain unbiased for a smaller $\epsilon$ (Dereziński et al., 2021b).

With these definitions and notations at hand, we are ready to assess the statistical properties of random sampling. A first quantity of interest to the design of random sampling is $m$, the number of trials needed to construct an $(\epsilon, \delta)$-subspace embedding, for some given importance sampling distribution $\{\pi_i\}_{i=1}^n$. A slightly more general result is given as follows.[1]

**Lemma 2.7** (Subspace embedding for random sampling). *Given $\mathbf{A} \in \mathbb{R}^{n \times d}$ of rank $d$ with $n \geq d$ and p.s.d. $\mathbf{C} \in \mathbb{R}^{d \times d}$, let $\mathbf{S}$ be a random sampling matrix with number of trials $m$ and importance sampling distribution $\{\pi_i\}_{i=1}^n$ as in Definition 2.1, and let $d_{\text{eff}} = \text{tr}(\mathbf{A}_{\mathbf{C}}^\top \mathbf{A}_{\mathbf{C}})$ be the effective dimension of $\mathbf{A}$ given $\mathbf{C}$ with $\mathbf{A}_{\mathbf{C}} \equiv \mathbf{A}(\mathbf{A}^\top \mathbf{A} + \mathbf{C})^{-1/2}$. Then, there exists $C > 0$ independent of $n, d_{\text{eff}}$ such that for $m \geq C \rho_{\max} d_{\text{eff}} \log(d_{\text{eff}}/\delta)/\epsilon^2$, failure probability $\delta \in (0, 1/2)$, $\epsilon > 0$, and $\rho_{\max}$ in Definition 2.3, $\mathbf{A}_{\mathbf{C}}^\top \mathbf{S}^\top \mathbf{S} \mathbf{A}_{\mathbf{C}}$ is an $(\epsilon, \delta)$-approximation of $\mathbf{A}_{\mathbf{C}}^\top \mathbf{A}_{\mathbf{C}}$.*

The proof of Lemma 2.7 uses standard matrix concentration techniques and is given in Appendix B. Note that Lemma 2.7 includes existing results of both leverage ($\mathbf{C} = \mathbf{0}_d$) and ridge leverage score ($\mathbf{C} = \lambda \mathbf{I}_d$) sampling as special cases, see Chowdhury et al. (2018, Theorm 3).

With Lemma 2.7, we are now ready to evaluate the inversion bias of random sampling. Since the matrix inverse is nonlinear, for $\mathbf{A}^\top \mathbf{S}^\top \mathbf{S} \mathbf{A}$ with $\mathbb{E}[\mathbf{A}^\top \mathbf{S}^\top \mathbf{S} \mathbf{A}] = \mathbf{A}^\top \mathbf{A}$, one should, a priori, *not* expect that $(\mathbf{A}^\top \mathbf{S}^\top \mathbf{S} \mathbf{A})^{-1}$ is an unbiased or nearly unbiased estimator of $(\mathbf{A}^\top \mathbf{A})^{-1}$. In the following result, we show (by adapting, in an almost straightforward fashion, the scalar debiasing proof approach of Dereziński et al. (2021a;b)) that this inversion bias can be corrected, *but only to some extent*, using the *same* scalar factor as for sub-gaussian or LESS projections. The proof of Proposition 2.8 is given in Appendix C for completeness.

**Proposition 2.8** (Coarse-grained debiasing of random sampling). *Given $\mathbf{A} \in \mathbb{R}^{n \times d}$ of rank $d$ with $n \geq d$ and p.s.d. $\mathbf{C} \in \mathbb{R}^{d \times d}$, let $\mathbf{S} \in \mathbb{R}^{m \times n}$ be a random sampling matrix with importance sampling distribution $\{\pi_i\}_{i=1}^n$ as in Definition 2.1 and max importance sampling approximation factor $\rho_{\max}$ as in Definition 2.3. Then, there exists $C > 0$ independent of $n, d_{\text{eff}}$ such that if $m \geq C \rho_{\max} d_{\text{eff}} (\log(d_{\text{eff}}/\delta) + \sqrt{d_{\text{eff}}}/\epsilon)$ with $\delta \leq m^{-3}$, $(\frac{m}{m - d_{\text{eff}}} \mathbf{A}^\top \mathbf{S}^\top \mathbf{S} \mathbf{A} + \mathbf{C})^{-1}$ is an $(\epsilon, \delta)$-unbiased estimator of $(\mathbf{A}^\top \mathbf{A} + \mathbf{C})^{-1}$.*

*Remark* 2.9 (On Proposition 2.8). While the debiasing factor

---

[1]Here we present the subspace embedding result in Lemma 2.7 on the (regularized) matrix $\mathbf{A}_{\mathbf{C}} \equiv \mathbf{A}(\mathbf{A}^\top \mathbf{A} + \mathbf{C})^{-1/2}$. This is of direct use in analyzing sub-sampled Newton methods in Section 4.

$\frac{m}{m-d_{\text{eff}}}$ is the same as that proposed for random (e.g., sub-gaussian or LESS) projections (Dereziński et al., 2021a;b), the resulting inversion bias is significantly larger. In particular, we have, in the case of Proposition 2.8 and for $m = \Theta(\rho_{\max} d_{\text{eff}} \log d_{\text{eff}})$, that $(\frac{m}{m-d_{\text{eff}}} \mathbf{A}^\top \mathbf{S}^\top \mathbf{S} \mathbf{A} + \mathbf{C})^{-1}$ has an inversion bias of order $O(\sqrt{d_{\text{eff}}}/\log d_{\text{eff}})$. This is a *vacuous* bound. It follows from Lemma 2.7 that for the same choice of $m$, $(\mathbf{A}^\top \mathbf{S}^\top \mathbf{S} \mathbf{A} + \mathbf{C})^{-1}$ is, *without* the debiasing factor, an $(O(1), \delta)$-approximation of $(\mathbf{A}^\top \mathbf{A} + \mathbf{C})^{-1}$, and thus has an inversion bias of order $O(1)$. See Lemma B.1 in Appendix B for a proof of this fact.

From Remark 2.9, it appears that the inversion bias result in Proposition 2.8 is disappointing: random sampling, in contrast with sub-gaussian or LESS random projections, while being numerically attractive and easy to implement, does *not* lead to a small inversion bias, at least with the $\frac{m}{m-d_{\text{eff}}}$ debiasing factor *and* the proof approach in Dereziński et al. (2021a;b) under Proposition 2.8. One may thus wonder:

> *Is it possible to get sharper control on the inversion bias of random sampling, either by introducing a different debiasing scheme and/or by using a more refined proof than Proposition 2.8?*

Below, we show that such improvement is indeed possible.

## 3. Fine-grained Analysis of Inversion Bias for Random Sampling

We have seen in Proposition 2.8 and Remark 2.9 that the scalar debiasing and the proof approach in Dereziński et al. (2021a;b) do *not* lead, in the case of random sampling, to a non-vacuous small inversion bias. In the following result, we provide fine-grained analysis of the inversion bias of random sampling (finer than that in Proposition 2.8), and we show that the inverse $(\mathbf{A}^\top \mathbf{S}^\top \mathbf{S} \mathbf{A} + \mathbf{C})^{-1}$ for random sampling $\mathbf{S}$ is biased in a more involved fashion than random projections studied in Dereziński et al. (2021a;b).

**Theorem 3.1** (Inversion bias for random sampling: fine–grained analysis). *Given $\mathbf{A} \in \mathbb{R}^{n \times d}$ of rank $d$ with $n \geq d$ and p.s.d. $\mathbf{C} \in \mathbb{R}^{d \times d}$, let $\mathbf{S} \in \mathbb{R}^{m \times n}$ be a random sampling matrix with importance sampling distribution $\{\pi_i\}_{i=1}^n$ as in Definition 2.1 and $(\rho_{\min}, \rho_{\max})$ as in Definition 2.3. Then, for diagonal matrix $\mathbf{D} = \text{diag}\{D_{ii}\}_{i=1}^n$ the solution to[2]*

$$D_{ii} = \frac{m}{m + \mathbf{a}_i^\top (\mathbf{A}^\top \mathbf{D} \mathbf{A} + \mathbf{C})^{-1} \mathbf{a}_i / \pi_i}, \quad (4)$$

*there exists $C > 0$ independent of $n, d_{\text{eff}}$ so that for $m \geq C\rho_{\max} d_{\text{eff}}(\log(d_{\text{eff}}/\delta) + 1/\epsilon^{2/3}), \delta \leq m^{-3}$, $(\mathbf{A}^\top \mathbf{S}^\top \mathbf{S} \mathbf{A} + \mathbf{C})^{-1}$ is an $(\epsilon, \delta)$-unbiased estimator of $(\mathbf{A}^\top \mathbf{D} \mathbf{A} + \mathbf{C})^{-1}$.*

[2]It can be checked that $\frac{m}{m+2\rho_{\max} d_{\text{eff}}} \mathbf{I}_n \preceq \mathbf{D} \preceq \frac{m}{m+\rho_{\min} d_{\text{eff}}} \mathbf{I}_n$ with $\rho_{\min}, \rho_{\max}$ in Definition 2.3. See Lemma D.3 in Appendix D.

*Heuristic derivation of Theorem 3.1.* For a more transparent understanding of the self-consistent equation in (4) of Theorem 3.1, we provide here a heuristic derivation. The detailed proof of Theorem 3.1 is deferred to Appendix D. Denote $\mathbf{x}_s^\top = \mathbf{e}_{i_s}^\top \mathbf{A}/\sqrt{\pi_{i_s}}$, $\mathbf{Q} = (\mathbf{A}^\top \mathbf{S}^\top \mathbf{S} \mathbf{A} + \mathbf{C})^{-1} = (\frac{1}{m}\sum_{s=1}^m \mathbf{x}_s \mathbf{x}_s^\top + \mathbf{C})^{-1}$ and $\mathbf{Q}_{-s} = (\sum_{j \neq s} \frac{1}{m} \mathbf{x}_j \mathbf{x}_j^\top + \mathbf{C})^{-1}$, for which we have $\sum_{s=1}^m \frac{1}{m} \mathbb{E}[\mathbf{x}_s \mathbf{x}_s^\top] = \sum_{i=1}^n \mathbf{a}_i \mathbf{a}_i^\top = \mathbf{A}^\top \mathbf{A}$. Then, we follow the *deterministic equivalent* framework (see, e.g., Couillet & Liao (2022, Chapter 2) for an introduction) and show that $\|\mathbb{E}[\mathbf{Q}] - \widehat{\mathbf{H}}^{-1}\| \simeq 0$, for $\widehat{\mathbf{H}} = \mathbf{A}^\top \mathbf{D} \mathbf{A} + \mathbf{C}$, for $\mathbf{D} \in \mathbb{R}^{n \times n}$ given in (4). First, note that $\|\mathbb{E}[\mathbf{Q}] - \widehat{\mathbf{H}}^{-1}\| = \|\mathbb{E}[\mathbf{Q}]\mathbf{A}^\top \mathbf{D} \mathbf{A} \widehat{\mathbf{H}}^{-1} - \mathbb{E}[\mathbf{Q} \mathbf{A}^\top \mathbf{S}^\top \mathbf{S} \mathbf{A}] \widehat{\mathbf{H}}^{-1}\|$, it then follows from Sherman-Morrison formula (Lemma A.3) that

$$\mathbb{E}[\mathbf{Q} \mathbf{A}^\top \mathbf{S}^\top \mathbf{S} \mathbf{A}] \widetilde{\mathbf{H}}^{-1} = \sum_{s=1}^m \mathbb{E}\left[\frac{\frac{1}{m}\mathbf{Q}_{-s} \mathbf{x}_s \mathbf{x}_s^\top \widetilde{\mathbf{H}}^{-1}}{1 + \mathbf{x}_s^\top \mathbf{Q}_{-s} \mathbf{x}_s/m}\right]$$
$$= \sum_{i=1}^n \mathbb{E}\left[\frac{\mathbf{Q}_{-s} \mathbf{a}_i \mathbf{a}_i^\top \widetilde{\mathbf{H}}^{-1}}{1 + \mathbf{a}_i^\top \mathbf{Q}_{-s} \mathbf{a}_i / m\pi_i}\right].$$

Using the rank-one perturbation lemma of matrix inverse, see, e.g., Silverstein & Bai (1995, Lemma 2.6), we obtain

$$\mathbb{E}[\mathbf{Q} \mathbf{A}^\top \mathbf{S}^\top \mathbf{S} \mathbf{A}] \widetilde{\mathbf{H}}^{-1} \simeq \sum_{i=1}^n \mathbb{E}\left[\frac{\mathbf{Q} \mathbf{a}_i \mathbf{a}_i^\top \widetilde{\mathbf{H}}^{-1}}{1 + \mathbf{a}_i^\top \widetilde{\mathbf{H}}^{-1} \mathbf{a}_i / m\pi_i}\right]$$
$$= \mathbb{E}[\mathbf{Q}] \mathbf{A}^\top \mathbf{D} \mathbf{A} \widetilde{\mathbf{H}}^{-1},$$

for $\mathbf{D} = \text{diag}\{m\pi_i/(m\pi_i + \mathbf{a}_i^\top \widetilde{\mathbf{H}}^{-1} \mathbf{a}_i)\}_{i=1}^n$. This leads to the self-consistent equation in (4) of Theorem 3.1. $\square$

Theorem 3.1 says that the (conditional) expectation $\mathbb{E}_\zeta[(\mathbf{A}^\top \mathbf{S}^\top \mathbf{S} \mathbf{A} + \mathbf{C})^{-1}]$, instead of being close to $(\mathbf{A}^\top \mathbf{A} + \mathbf{C})^{-1}$, is in fact close to $(\mathbf{A}^\top \mathbf{D} \mathbf{A} + \mathbf{C})^{-1}$, with $\mathbf{D}$ depending on $\mathbf{A}$ and the random sampling scheme per $m$ and $\{\pi_i\}_{i=1}^n$ in an implicit fashion. While seemingly uninterpretable and unusable at first sight, Theorem 3.1 can be tuned to design a de-biased random sampling approach. This is given in the following result.

**Proposition 3.2** (Fine-grained debiasing for general random sampling). *Under the settings and notations of Theorem 3.1, for $\ell_{i_s}^{\mathbf{C}}$ the $i_s^{th}$ leverage score of $\mathbf{A}$ as in Definition 2.2 and standard random sampling matrix $\mathbf{S}$ as in Definition 2.1, define the de-biased sampling matrix $\check{\mathbf{S}} \in \mathbb{R}^{m \times n}$ as*

$$\check{\mathbf{S}} = \text{diag}\left\{\sqrt{m/(m - \ell_{i_s}^{\mathbf{C}}/\pi_{i_s})}\right\}_{s=1}^m \cdot \mathbf{S}. \quad (5)$$

*Then, there exists constant $C > 0$ independent of $n, d_{\text{eff}}$ such that for $m \geq C\rho_{\max} d_{\text{eff}}(\log(d_{\text{eff}}/\delta) + 1/\epsilon^{2/3}), \delta \leq m^{-3}$, $(\mathbf{A}^\top \check{\mathbf{S}}^\top \check{\mathbf{S}} \mathbf{A} + \mathbf{C})^{-1}$ is an $(\epsilon, \delta)$-unbiased estimator of $(\mathbf{A}^\top \mathbf{A} + \mathbf{C})^{-1}$.*

*Heuristic derivation of Proposition 3.2.* To make the intuition behind Proposition 3.2 more accessible, we present here a heuristic derivation of (5). We refer the reader to Appendix E for the detailed proof of Proposition 3.2. Let $\check{\mathbf{S}}^\top \check{\mathbf{S}} = \sum_{s=1}^m F_{i_s i_s} \cdot \mathbf{e}_{i_s} \mathbf{e}_{i_s}^\top / m \pi_{i_s}$ for some *deterministic* $F_{ii}$ to be specified, $\check{\mathbf{Q}} = (\mathbf{A}^\top \check{\mathbf{S}}^\top \check{\mathbf{S}} \mathbf{A} + \mathbf{C})^{-1} = \left( \frac{1}{m} \sum_{s=1}^m F_{i_s i_s} \mathbf{x}_s \mathbf{x}_s^\top + \mathbf{C} \right)^{-1}$, and similarly $\check{\mathbf{Q}}_{-s} = (\frac{1}{m} \sum_{l \neq s} F_{i_l i_l} \mathbf{x}_l \mathbf{x}_l^\top + \mathbf{C})^{-1}$ as in the heuristic derivation of Theorem 3.1 above. We thus have $\frac{1}{m} \sum_{s=1}^m \mathbb{E}[F_{i_s i_s} \mathbf{x}_s \mathbf{x}_s^\top] = \sum_{i=1}^n F_{ii} \mathbf{a}_i \mathbf{a}_i^\top$. Our goal is to determine $\check{\mathbf{Q}}$ (and $F_{ii}$) such that $\|\mathbb{E}[\check{\mathbf{Q}}] - \mathbf{H}^{-1}\| \simeq 0$, for $\mathbf{H}^{-1} = (\mathbf{A}^\top \mathbf{A} + \mathbf{C})^{-1}$. To this end, observe that

$$\mathbb{E}[\check{\mathbf{Q}}] - \mathbf{H}^{-1} = \mathbb{E}[\check{\mathbf{Q}}] \mathbf{A}^\top \mathbf{A} \mathbf{H}^{-1} - \mathbb{E}[\check{\mathbf{Q}} \mathbf{A}^\top \check{\mathbf{S}}^\top \check{\mathbf{S}} \mathbf{A}] \mathbf{H}^{-1} \simeq 0.$$

By Sherman-Morrison formula (Lemma A.3), we obtain

$$\mathbb{E}[\check{\mathbf{Q}} \mathbf{A}^\top \check{\mathbf{S}}^\top \check{\mathbf{S}} \mathbf{A}] \mathbf{H}^{-1} = \mathbb{E} \left[ \frac{\check{\mathbf{Q}}_{-s} F_{i_s i_s} \mathbf{A}^\top \mathbf{e}_{i_s} \mathbf{e}_{i_s}^\top / \pi_{i_s} \mathbf{A} \mathbf{H}^{-1}}{1 + F_{i_s i_s} \mathbf{e}_{i_s}^\top \mathbf{A} \check{\mathbf{Q}}_{-s} \mathbf{A}^\top \mathbf{e}_{i_s} / m \pi_{i_s}} \right]$$
$$= \sum_{i=1}^n \mathbb{E} \left[ \frac{\check{\mathbf{Q}}_{-s} F_{ii} \mathbf{A}^\top \mathbf{e}_i \mathbf{e}_i^\top \mathbf{A} \mathbf{H}^{-1}}{1 + F_{ii} \mathbf{e}_i^\top \mathbf{A} \check{\mathbf{Q}}_{-s} \mathbf{A}^\top \mathbf{e}_i / m \pi_i} \right],$$

where we see the *exact* leverage score $\mathbf{e}_i^\top \mathbf{A} \mathbf{H}^{-1} \mathbf{A}^\top \mathbf{e}_i = \mathbf{a}_i^\top (\mathbf{A}^\top \mathbf{A} + \mathbf{C})^{-1} \mathbf{a}_i = \ell_i^{\mathbf{C}}$ as in Definition 2.2 naturally appears in the denominator from the derivation. Invoking the rank-one perturbation lemma once more, we have that

$$\mathbb{E}[\check{\mathbf{Q}} \mathbf{A}^\top \check{\mathbf{S}}^\top \check{\mathbf{S}} \mathbf{A}] \mathbf{H}^{-1} \simeq \mathbb{E}[\check{\mathbf{Q}}] \mathbf{A}^\top \sum_{i=1}^n \frac{F_{ii} \mathbf{e}_i \mathbf{e}_i^\top}{1 + F_{ii} \ell_i^{\mathbf{C}} / m \pi_i} \mathbf{A} \mathbf{H}^{-1}$$
$$= \mathbb{E}[\check{\mathbf{Q}}] \mathbf{A}^\top \mathbf{A} \mathbf{H}^{-1},$$

where we take the *debiasing factor* $F_{ii} = m \pi_i / (m \pi_i - \ell_i^{\mathbf{C}})$ such that $F_{ii} / (1 + F_{ii} \ell_i^{\mathbf{C}} / m \pi_i) = 1$. This leads to the form of the debiasing matrix $\check{\mathbf{S}}$ as in (5) of Proposition 3.2. $\square$

Comparing the fine-grained results in Proposition 3.2 to the coarse-grained results in Proposition 2.8, we see that the large inversion bias in Proposition 2.8 is indeed a consequence of the proof approach adapted from Dereziński et al. (2021a;b), that is *inadequate* for random sampling and for structured random projections such as the SRHT.

*Remark* 3.3 ($\check{\mathbf{S}}$ as a random sampling scheme). Note that $\check{\mathbf{S}} \in \mathbb{R}^{m \times n}$ in Proposition 3.2 is nothing but another random sampling matrix: it features exactly one nonzero entry per row that is equal to $(m \pi_{i_s} - \ell_{i_s}^{\mathbf{C}})^{-1/2}$, as opposed to $(m \pi_{i_s})^{-1/2}$ for the standard random sampling $\mathbf{S}$ in Definition 2.1. This non-standard re-weighting (that uses the leverage scores of $\mathbf{A}$) ensures that $(\mathbf{A}^\top \check{\mathbf{S}}^\top \check{\mathbf{S}} \mathbf{A} + \mathbf{C})^{-1}$ is a nearly unbiased estimate of $(\mathbf{A}^\top \mathbf{A} + \mathbf{C})^{-1}$, per Proposition 3.2. From a computational perspective, the re-weighted random sampling $\check{\mathbf{S}}$ in (5) can be computationally demanding due to the need for *exact* computation of leverage scores

$\ell_i^{\mathbf{C}}$ in (5). In Corollary E.1 of Appendix E, we consider approximate leverage scores (which are much faster to compute (Drineas et al., 2012; Clarkson & Woodruff, 2017; Cohen et al., 2017)). We show that for a given sampling scheme $\{\pi_i\}_{i=1}^n$, replacing exact leverage scores with their approximate counterparts in the de-biased sampling matrix $\check{\mathbf{S}}$ in (5) increases the inversion bias, but only very slightly.

Note from Proposition 3.2 that the proposed fine-grained de-biasing matrix $\check{\mathbf{S}}$ depends on the importance sampling distribution *only* via $\ell_i^{\mathbf{C}} / \pi_i$. (See Appendix E.1 for the RMT intuition on how the exact leverage scores arise from the derivation.) As such, for any random sampling method with $\pi_i \approx \ell_i^{\mathbf{C}} / d_{\text{eff}}$ close to those of exact leverage score sampling in Definition 2.2, we have $\check{\mathbf{S}} \approx \frac{m}{m - d_{\text{eff}}} \mathbf{S}$. This *coincides* with the scalar debiasing scheme in the coarse-grained result of Proposition 2.8, but it has a much smaller inversion bias. This special case is discussed in the following result, proven in Appendix E.3.2.

**Corollary 3.4** (Inversion bias using scalar debiasing under approximate leverage). *Under the settings and notations of Theorem 3.1, for random sampling scheme with sampling distribution $\pi_i \in [\ell_i^{\mathbf{C}} / (d_{\text{eff}} \rho_{\max}), \ell_i^{\mathbf{C}} / (d_{\text{eff}} \rho_{\min})]$ with $\rho_{\min} \in [1/2, 1]$ as in Definition 2.3,[3] there exists $C > 0$, $\nu \geq \log_{d_{\text{eff}}}(\log(d_{\text{eff}} / \delta))$, $\delta < m^{-3}$ such that for $m \geq C \rho_{\max} d_{\text{eff}}^{1+\nu}$, $(\frac{m}{m - d_{\text{eff}}} \mathbf{A}^\top \mathbf{S}^\top \mathbf{S} \mathbf{A} + \mathbf{C})^{-1}$ is an $(\epsilon, \delta)$-unbiased estimator of $(\mathbf{A}^\top \mathbf{A} + \mathbf{C})^{-1}$ with inversion bias $\epsilon = \max\{O(d_{\text{eff}}^{-3\nu/2}), O(\epsilon_\rho d_{\text{eff}}^{-\nu})\}$ and $\epsilon_\rho = \max\{\rho_{\min}^{-1} - 1, 1 - \rho_{\max}^{-1}\}$.*

*Remark* 3.5 (Inversion bias for exact versus approximate leverage score sampling). It follows from Corollary 3.4 that for exact and/or approximate leverage score sampling with $\rho_{\max} \geq \rho_{\min} / (2\rho_{\min} - 1) \geq 1$ and $\rho_{\min} > (1 + \Theta(d_{\text{eff}}^{-\nu/2}))^{-1} > 1/2$ (so that $\epsilon_\rho = 1 - \rho_{\max}^{-1}$), the inversion bias induced by the scalar debiasing $\frac{m}{m - d_{\text{eff}}}$ establishes the following *phase transition* behavior:

1. if the random sampling scheme is sufficiently close to exact leverage sampling, in that $\rho_{\max} \in [\rho_{\min} / (2\rho_{\min} - 1), 1 / (1 - \Theta(d_{\text{eff}}^{-\nu/2}))]$ (or equivalently the importance sampling probabilities satisfy $\pi_i \in [(1 \pm \Theta(d_{\text{eff}}^{-\nu/2})) \ell_i^{\mathbf{C}} / d_{\text{eff}}]$), then the inversion bias under scalar debiasing is the *same* as that (of the fine-grained matrix debiasing) in Proposition 3.2; but

2. if the random sampling scheme significantly deviates from exact leverage sampling with $\rho_{\max} > 1 / (1 - \Theta(d_{\text{eff}}^{-\nu/2}))$ (or equivalently $|\pi_i - \ell_i^{\mathbf{C}} / d_{\text{eff}}| > \Theta(d_{\text{eff}}^{-\nu/2}) \ell_i^{\mathbf{C}} / d_{\text{eff}}$), then the inversion bias under scalar

---

[3]Note that by Definition 2.3 we have $\rho_{\min} \leq \ell_i^{\mathbf{C}} / (\pi_i d_{\text{eff}}) \leq \rho_{\max}$ for all $i$, which, together with $\rho_{\min} \geq 1/2$ yields that $|\pi_i - \ell_i^{\mathbf{C}} / d_{\text{eff}}| \leq \epsilon_\rho \ell_i^{\mathbf{C}} / d_{\text{eff}} \leq \ell_i^{\mathbf{C}} / d_{\text{eff}}$.

debiasing becomes larger than that in Proposition 3.2, increases with $\rho_{\max}$, and saturates at $\rho_{\max} = \Theta(1)$.

This phase transition behavior is visualized in Figure 1. See also Figure 4 in Appendix G for the numerical comparison of inversion bias using scalar debiasing between exact and approximate leverage score sampling.

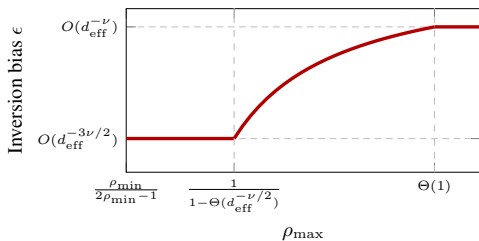

Figure 1: The phase transition behavior of inversion bias $\epsilon$ as a function of $\rho_{\max}$ discussed in Remark 3.5 with scalar debiasing.

As a side remark, it is known from Dereziński et al. (2021b, Theorem 10) that for *approximate* leverage sampling, and *any* scalar $\gamma > 0, m > 0$, $(\gamma \mathbf{A}^\top \mathbf{S}^\top \mathbf{S} \mathbf{A} + \mathbf{C})^{-1}$ with $\mathbf{C} = \mathbf{0}_d$ is not an $(\epsilon, \delta)$-unbiased estimator of $(\mathbf{A}^\top \mathbf{A} + \mathbf{C})^{-1}$, with any $\epsilon \leq c d_{\text{eff}}/m$ and $c > 0$ an absolute constant. Thus,

1. in the case of *approximate* leverage score sampling with $\rho_{\max} = 3/2$ and $\rho_{\min} = 1/2$, for any $m \geq C \rho_{\max} d_{\text{eff}} \log d_{\text{eff}}$, it follows from the proof of Corollary 3.4 that the inversion bias is upper bounded by $O(d_{\text{eff}}/m)$, and this coincides with the lower bound in Dereziński et al. (2021b, Theorem 10); and

2. in the case of *exact* leverage score sampling with $\rho_{\max} = \rho_{\min} = 1$, the inversion bias can be made smaller than $d_{\text{eff}}/m$ under scalar debiasing.[4]

As an important consequence, Proposition 3.2 also applies to effectively de-bias another commonly-used data-oblivious sketching scheme, the SRHT (Ailon & Chazelle, 2006).

**Definition 3.6** (Sub-sampled randomized Walsh–Hadamard transform, SRHT, Ailon & Chazelle (2006))**.** For a given matrix $\mathbf{A} \in \mathbb{R}^{n \times d}$ of rank $d$ with $n \geq d$, assume without loss of generality that $n = 2^p$ for some integer $p$. Then, the SRHT of $\mathbf{A}$ is given by

$$\tilde{\mathbf{A}}_{\text{SRHT}} = \mathbf{S} \mathbf{H}_n \mathbf{D}_n \mathbf{A}/\sqrt{n} \in \mathbb{R}^{m \times n}, \qquad (6)$$

for uniform random sampling matrix $\mathbf{S} \in \mathbb{R}^{m \times n}$, $\pi_i = 1/n$ as in Definition 2.1, $\mathbf{H}_n \in \mathbb{R}^{n \times n}$ the Walsh–Hadamard matrix of size $n$, and diagonal $\mathbf{D}_n \in \mathbb{R}^{n \times n}$ having i.i.d. Rademacher random variables on its diagonal.

The SRHT in Definition 3.6 enjoys the following properties: the randomized Walsh–Hadamard transform $\mathbf{H}_n \mathbf{D}_n \mathbf{A}$ of $\mathbf{A}$ is known to have approximately uniform leverages scores, that is $\ell_i^{\mathbf{C}}(\mathbf{H}_n \mathbf{D}_n \mathbf{A}/\sqrt{n}) \approx d_{\text{eff}}/n$, see Drineas et al. (2011) and Tropp (2011, Theorems 3.1 and 3.2), as well as Lemma E.4 in Appendix E.3 in our setting; and since $\mathbf{H}_n^\top \mathbf{H}_n/n = \mathbf{I}_n$ and $\mathbf{D}_n^2 = \mathbf{I}_n$, one has $\frac{1}{n} \mathbf{A}^\top \mathbf{D}_n \mathbf{H}_n^\top \mathbf{H}_n \mathbf{D}_n \mathbf{A} = \mathbf{A}^\top \mathbf{A}$, so that $\mathbf{H}_n \mathbf{D}_n \mathbf{A}/\sqrt{n}$ and $\mathbf{A}$ have the same effective dimension. These lead to the following fine-grained debiasing result for SRHT with scalar debiasing, proven in Appendix E.3.4.

**Corollary 3.7** (Fine-grained debiasing for SRHT using scalar debiasing)**.** *Under the setting and notations of Theorem 3.1, for $\tilde{\mathbf{A}}_{\text{SRHT}} \in \mathbb{R}^{m \times n}$ the SRHT of $\mathbf{A}$ as in Definition 3.6, then there exists $C > 0$, $\nu \geq 0$, $n \exp(-d_{\text{eff}}) < \delta < m^{-3}$ such that for $m \geq C \rho_{\max} d_{\text{eff}}^{1+\nu}$, $(\frac{m}{m-d_{\text{eff}}} \tilde{\mathbf{A}}_{\text{SRHT}}^\top \tilde{\mathbf{A}}_{\text{SRHT}} + \mathbf{C})^{-1}$ is an $(O(d_{\text{eff}}^{-3\nu/2}) + O(\rho_{\max}^{-1} \sqrt{\log(n/\delta)} d_{\text{eff}}^{-\nu-1/2}), \delta)$-unbiased estimator of $(\mathbf{A}^\top \mathbf{A} + \mathbf{C})^{-1}$.*[5]

## 4. Application to De-biased Sub-sampled Newton with Improved Convergence

In this section, we show that the precise characterizations of random sampling inversion bias and the debiasing techniques in Section 3 apply to establish *problem-independent* local convergence rates of SSN methods.

Consider the following optimization problem:

$$\boldsymbol{\beta}^* = \underset{\boldsymbol{\beta} \in \mathcal{C}}{\arg\min} \, F(\boldsymbol{\beta}) = \underset{\boldsymbol{\beta} \in \mathcal{C}}{\arg\min} \, f(\boldsymbol{\beta}) + \Phi(\boldsymbol{\beta}), \quad (7)$$

for some smooth function $F: \mathbb{R}^d \to \mathbb{R}$ that can be decomposed into $f$ and $\Phi$, and $\mathcal{C} \subseteq \mathbb{R}^d$ a convex set. This decomposition naturally arises in ML when, e.g., the loss function $F$ is the sum of the empirical risk $f$ over a set of $n$ training samples and some regularization penalty $\Phi$.

Newton's method solves (7) by performing iterative updates $\boldsymbol{\beta}_{t+1} = \boldsymbol{\beta}_t - \mu_t \mathbf{H}_t^{-1}(\boldsymbol{\beta}_t) \nabla F(\boldsymbol{\beta}_t)$, with $\mu_t$ the step size, $\nabla F(\boldsymbol{\beta}_t) \in \mathbb{R}^d$ the gradient, and $\mathbf{H}_t(\boldsymbol{\beta}_t) \in \mathbb{R}^{d \times d}$ the Hessian of $F$ at $\boldsymbol{\beta}_t$ that can be decomposed as

$$\mathbf{H}_t(\boldsymbol{\beta}_t) = \mathbf{A}(\boldsymbol{\beta}_t)^\top \mathbf{A}(\boldsymbol{\beta}_t) + \mathbf{C}(\boldsymbol{\beta}_t), \qquad (8)$$

with $\mathbf{A}(\boldsymbol{\beta}_t) \in \mathbb{R}^{n \times d}$ and some p.s.d. matrix $\mathbf{C}(\boldsymbol{\beta}_t) = \nabla^2 \Phi(\boldsymbol{\beta}_t) \in \mathbb{R}^{d \times d}$ that takes a simple form, e.g., $\mathbf{C}(\boldsymbol{\beta}_t) = 2\lambda \mathbf{I}_d$ in the case of $L_2$ regularization $\Phi(\boldsymbol{\beta}) = \lambda \|\boldsymbol{\beta}\|^2$.

---

[4]Notably, using *exact* (instead of approximate) leverage score sampling in the same setting of Dereziński et al. (2021b, Theorem 10), the inversion bias (conditioned on any event $\zeta$ that ensures invertibility) can be made *zero* by taking $\gamma = \frac{m}{d} \mathbb{E}_\zeta[1/b]$ for $b$ distributed as $\text{Binomial}(m, 1/d)$. This aligns with our conclusion of a possibly smaller inversion bias than $d_{\text{eff}}/m$. See Corollary E.2 in Appendix E for a proof of this fact.

[5]Recall from Definition 2.3 that here for SRHT, we have $\pi_i = 1/n$ and thus $\rho_{\max} = \max_{1 \leq i \leq n} n \ell_i^{\mathbf{C}}(\mathbf{H}_n \mathbf{D}_n \mathbf{A}/\sqrt{n})/d_{\text{eff}}$.

Despite having a locally super-linear convergence rate, Newton's method suffers from a heavy computational burden in forming the Hessian matrix $\mathbf{H}_t(\boldsymbol{\beta}_t)$, particularly when the training samples $n$ is large, e.g., $n \gg d$. In this case, the major computational bottleneck of Newton's method lies in the construction of $\mathbf{A}(\boldsymbol{\beta}_t)^\top \mathbf{A}(\boldsymbol{\beta}_t)$ for the *inverse* Hessian matrix, as this takes $O(nd^2)$ time.[6] Many randomized second-order methods have been proposed to replace the exact Hessian inverse by some computationally efficient estimate. Here, we consider SSN methods, that randomly sample the Hessian (Yao et al., 2018; Roosta-Khorasani & Mahoney, 2019; Xu et al., 2020) as follows.

**Definition 4.1** (Sub-sampled Newton, SSN). To solve the optimization problem in (7), the SSN method performs the following iteration:

$$\boldsymbol{\beta}_{t+1} = \boldsymbol{\beta}_t - \mu_t \left( \mathbf{A}(\boldsymbol{\beta}_t)^\top \mathbf{S}_t^\top \mathbf{S}_t \mathbf{A}(\boldsymbol{\beta}_t) + \mathbf{C}(\boldsymbol{\beta}_t) \right)^{-1} \mathbf{g}_t, \quad (9)$$

for $t = 0, 1, \dots T$, with $\mathbf{g}_t \equiv \nabla F(\boldsymbol{\beta}_t) \in \mathbb{R}^d$ the gradient of $F$ at $\boldsymbol{\beta}_t$, $\mu_t$ the step size at time $t$, and random sampling matrix $\mathbf{S}_t \in \mathbb{R}^{m \times n}$ as in Definition 2.1.

By randomly sampling the (computationally intense component of the) Hessian matrix, each SSN step in (9) takes only $O(md^2)$ time. The (local or global) convergence rates of SSN have been extensively studied in the literature of optimization, RandNLA, and ML; see, e.g., Xu et al. (2016); Bollapragada et al. (2019); Roosta-Khorasani & Mahoney (2019); Ye et al. (2021); Lacotte et al. (2021) and Section 1.2 above. In particular, Lacotte et al. (2021) proposed an adaptive SSN-type algorithm that achieves a quadratic convergence rate by dynamically adjusting the sketch size, improving upon the well-established linear–quadratic rates of traditional SSN methods. Incorporating their adaptive algorithm into our proposed de-biased approach could potentially accelerate convergence further; however, this exploration lies beyond the scope of the present work.

Due to the absence of precise characterizations of the sub-sampled Hessian inverse, as in Theorem 3.1 and Proposition 3.2 (that allow to, e.g., prove the *near-unbiasedness* of SSN iteration $\mathbb{E}[\boldsymbol{\beta}_{t+1}] \approx \boldsymbol{\beta}_t - \mu_t \mathbf{H}_t^{-1}(\boldsymbol{\beta}_t)\mathbf{g}_t)$, it is technically challenging to obtain *problem-independent* convergence rates for SSN. In the following, we fill this gap by showing how our inversion bias results in Section 3 apply to establish *problem-independent* local convergence rates for *de-biased* SSN. We position ourselves under the following standard assumption on the objective function $F$.

**Assumption 4.2** (Lipschitz Hessian). $F, f \colon \mathbb{R}^d \to \mathbb{R}$ in (7) have Lipschitz continuous Hessian with Lipschitz constant $L$, that is, for any $\boldsymbol{\beta}, \boldsymbol{\beta}' \in \mathbb{R}^d$, $\max\{\|\nabla^2 F(\boldsymbol{\beta}) - \nabla^2 F(\boldsymbol{\beta}')\|, \|\nabla^2 f(\boldsymbol{\beta}) - \nabla^2 f(\boldsymbol{\beta}')\|\} \leq L\|\boldsymbol{\beta} - \boldsymbol{\beta}'\|$.

---

[6]Of course, this does *not* need to be done explicitly.

Under Assumption 4.2, we evaluate the local convergence rate of the following *de-biased* SSN iterations:

$$\check{\boldsymbol{\beta}}_{t+1} = \check{\boldsymbol{\beta}}_t - \mu_t \left( \mathbf{A}(\check{\boldsymbol{\beta}}_t)^\top \check{\mathbf{S}}_t^\top \check{\mathbf{S}}_t \mathbf{A}(\check{\boldsymbol{\beta}}_t) + \mathbf{C}(\check{\boldsymbol{\beta}}_t) \right)^{-1} \mathbf{g}_t, \quad (10)$$

with de-biased $\check{\mathbf{S}}_t = \mathrm{diag}\left\{ \sqrt{m/(m - \ell_{i_s}^{\mathbf{C}}(\check{\boldsymbol{\beta}}_t)/\pi_{i_s})} \right\}_{s=1}^m \cdot$ $\mathbf{S}_t$ as in Proposition 3.2, with $\ell_{i_s}^{\mathbf{C}}(\check{\boldsymbol{\beta}}_t)$ the $i_s^{th}$ leverage score of $\mathbf{A}(\check{\boldsymbol{\beta}}_t)$ given $\mathbf{C}(\check{\boldsymbol{\beta}}_t)$. This leads to the following result.

**Theorem 4.3** (Local convergence of de-biased SSN). *Let Assumption 4.2 hold. For p.d. $\mathbf{A}(\boldsymbol{\beta}^*)^\top \mathbf{A}(\boldsymbol{\beta}^*) = \nabla^2 f(\boldsymbol{\beta}^*)$ and p.s.d. $\mathbf{C}(\boldsymbol{\beta}^*) = \nabla^2 \Phi(\boldsymbol{\beta}^*)$, there exists a neighborhood $U$ of $\boldsymbol{\beta}^*$ such that the de-biased SSN iteration in (10) starting from $\check{\boldsymbol{\beta}}_0 \in U$ satisfies, for $U = \{\boldsymbol{\beta} \colon \|\boldsymbol{\beta} - \boldsymbol{\beta}^*\|_{\mathbf{H}} < (\rho_{\max} d_{\mathrm{eff}} \sigma_{\min}/m)^{3/2}/L\}$, step size $\mu_t = 1 - \frac{\rho_{\max}}{m/d_{\mathrm{eff}} + \rho_{\max}}$, $m \geq C \rho_{\max} d_{\mathrm{eff}}^{1+\nu}$, and $\nu \geq \log_{d_{\mathrm{eff}}}(\log(d_{\mathrm{eff}} T/\delta))$ that*

$$\left( \mathbb{E}_\zeta \left[ \frac{\|\check{\boldsymbol{\beta}}_T - \boldsymbol{\beta}^*\|_{\mathbf{H}}}{\|\check{\boldsymbol{\beta}}_0 - \boldsymbol{\beta}^*\|_{\mathbf{H}}} \right] \right)^{1/T} \leq \frac{\rho_{\max} d_{\mathrm{eff}}}{m}(1 + \epsilon), \quad (11)$$

*holds for $\epsilon = O(d_{\mathrm{eff}}^{-\nu/2})$ and conditioned on an event $\zeta$ that happens with probability at least $1 - \delta$. Here, $\sigma_{\min}$ is the smallest singular value of $\mathbf{H} \equiv \mathbf{A}(\boldsymbol{\beta}^*)^\top \mathbf{A}(\boldsymbol{\beta}^*) + \mathbf{C}(\boldsymbol{\beta}^*)$, $\rho_{\max}$ is the max importance sampling approximation factor in Definition 2.3 for $\ell_i^{\mathbf{C}} = \max_{1 \leq t \leq T} \ell_i^{\mathbf{C}}(\check{\boldsymbol{\beta}}_t)$ and $d_{\mathrm{eff}} = \max_{1 \leq t \leq T} d_{\mathrm{eff}}(\check{\boldsymbol{\beta}}_t)$ with $\ell_i^{\mathbf{C}}(\check{\boldsymbol{\beta}}_t)$ and $d_{\mathrm{eff}}(\check{\boldsymbol{\beta}}_t)$ the leverage scores and effective dimension of $\mathbf{A}(\check{\boldsymbol{\beta}}_t)$ given $\mathbf{C}(\check{\boldsymbol{\beta}}_t)$, respectively.*

The proof of Theorem 4.3 can be found in Appendix F.2. The proof relies on a precise characterization of the second inverse moment of the randomly sampled Hessian matrix, extending beyond the first inverse moment result in Proposition 3.2. Due to space limitation, this technical result is presented in Proposition F.1 of Appendix F. For the sake of practical implementation, we also present, in Corollaries F.5 and F.6 of Appendix F, local convergence rates of SSN iterations *using scalar debiasing $m/(m - d_{\mathrm{eff}})$* under exact and approximate leverage score sampling as well as SRHT.

## 5. Numerical Experiments of De-biased SSN

In this section, we provide empirical evidence showing the improved convergence (and a better "complexity–convergence" trade-off as a consequence) of different de-biased SSN methods proposed in Section 4, with respect to several first- and second-order baseline optimization methods. We solve the following logistic regression problem

$$\min_{\boldsymbol{\beta} \in \mathbb{R}^d} \frac{1}{n} \sum_{i=1}^n \log\left(1 + \exp(-y_i \mathbf{a}_i^\top \boldsymbol{\beta})\right) + \frac{\lambda}{2}\|\boldsymbol{\beta}\|^2, \quad (12)$$

of the form (7), for regularization $\lambda > 0$, where $\mathbf{a}_i^\top \in \mathbb{R}^d$ is the $i^{th}$ row of data matrix $\mathbf{A} \in \mathbb{R}^{n \times d}$ sampled from both

MNIST (LeCun et al., 1998) and CIFAR-10 (Krizhevsky, 2009) datasets, and $\mathbf{y} \in \{\pm 1\}^n$ the response vector. Implementation details are provided in Appendix G. Note that the time reported in Figures 2 and 3 include both the input data pre-processing time ( e.g., the computation of exact or approximate leverage scores, and Walsh–Hadamard transform) and the computational overhead associated with the sketching process.

Figure 2 assesses the impact of the sketch size $m$ on both relative error ($\|\check{\boldsymbol{\beta}}_T - \boldsymbol{\beta}^*\|_{\mathbf{H}}^2 / \|\check{\boldsymbol{\beta}}_0 - \boldsymbol{\beta}^*\|_{\mathbf{H}}^2$) and running time of de-biased SSN employing approximate ridge leverage score sampling (SSN-ARLev), in comparison to the Newton-LESS method (Dereziński et al., 2021a) based on random projection. The results in Figure 2 demonstrate that the proposed de-biased SSN consistently outperforms Newton-LESS across all tested sketch size $m$, exhibiting a *superior convergence–complexity trade-off*. Notably, while the running time for Newton-LESS increases significantly with $m$, that of SSN-ARLev remains approximately constant.

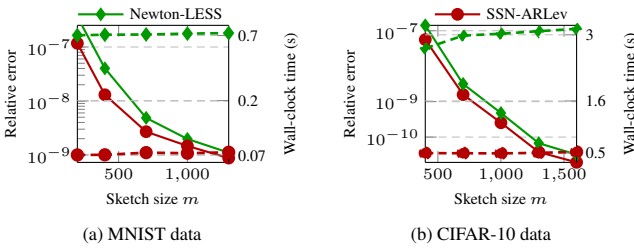

(a) MNIST data      (b) CIFAR-10 data

Figure 2: Relative errors (in solid lines) and wall-clock time (in dashed lines) as a function of the sketch size $m$, for **Newton-LESS** and the proposed de-biased **SSN-ARLev** methods, applied to logistic regression on both MNIST and CIFAR-10 data. Relative errors are obtained after a fixed number of iterations ($T = 5$ for MNIST data and $T = 7$ for CIFAR-10 data). Results are obtained by averaging over 30 independent runs.

Figure 3 compares the relative errors as a function of wall-clock time across various optimization methods:

1. **First-order baselines**: Gradient Descent (GD) and Stochastic GD (SGD).

2. **De-biased SSN methods** using different sampling schemes: Approximate $\lambda$-Ridge Leverage Score (AR-Lev), Approximate Leverage Score (ALev), Shrinkage Leverage Score (SLev) sampling (Ma et al., 2015), and SRHT (see Definition 3.6 above).

3. **Newton Sketch with LESS-uniform sketch** (LESS) (Dereziński et al., 2021a), which achieves significantly shorter running times compared to the original Newton Sketch with dense Gaussian sketches.

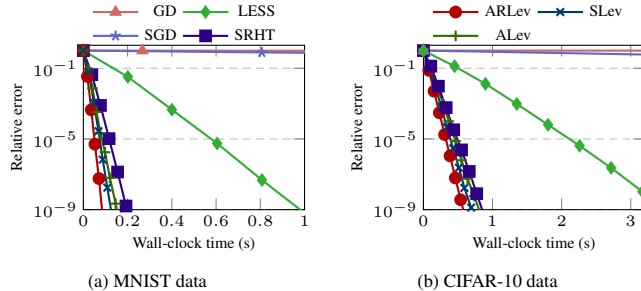

(a) MNIST data      (b) CIFAR-10 data

Figure 3: Convergence–complexity trade-off between various optimization methods on logistic regression for MNIST and CIFAR-10 data, with sketch size $m = 300$ for MNIST and $m = 400$ for CIFAR-10 data. Results are obtained by averaging over 10 independent runs (except for GD that is deterministic).

From Figure 3, we see that SSN methods, when properly de-biased, exhibit a significantly better *convergence–complexity* trade-off than both first-order methods *and* the Newton-LESS approach. The SRHT sampling outperforms Newton-LESS but still lags slightly behind all other SSN variants in speed. Among these, the SLev scheme edges out ALev yet remains slower than ARLev. Across both datasets tested, de-biased SSN with ARLev consistently delivers the *optimal* convergence–complexity trade-off among all methods evaluated.

## 6. Conclusions and Perspectives

In the work, we investigate the inversion bias inherent in various random sampling schemes, including uniform and non-uniform leverage-based sampling, as wel as structured random projections (e.g., the Hadamard transform-based SRHT). Leveraging recent advances in RMT and RandNLA, we provide a precise characterization of this inversion bias and propose corresponding de-biasing techniques. Notably, for approximate leverage sampling and SRHT, this de-biasing reduces to a simple scalar rescaling. We further show that our results enable an improved SSN method, achieving local convergence rates comparable to those of Newton Sketch with dense Gaussian projections. Our theoretical insights are complemented by numerical results on MNIST and CIFAR-10 datasets, underscoring the practical relevance of the proposed approach.

It would be of future interest to see whether the proposed debiasing technique, when wisely combined with adaptive sampling schemes, can achieve or even improve the quadratic convergence in Lacotte et al. (2021). In addition, it is also worthwhile to extend our debiasing framework to dependent sampling methods (Cortinovis & Kressner, 2024), such as volume sampling, which may further improve SSN's efficiency and convergence.

## Acknowledgements

Z. Liao would like to acknowledge the National Natural Science Foundation of China (via fund NSFC-62206101) and the Guangdong Provincial Key Laboratory of Mathematical Foundations for Artificial Intelligence (2023B1212010001) for providing partial support. Z. Ling is supported by the National Natural Science Foundation of China (via NSFC-62406119) and the Natural Science Foundation of Hubei Province (2024AFB074). M. W. Mahoney would like to acknowledge the DARPA, DOE, NSF, and ONR for providing partial support of this work.

## Impact Statement

This paper presents work whose goal is to advance the field of ML, RandNLA, and their connection to RMT. The work is primarily theoretical, and we do not see any immediate negative societal impact. If anything, providing methods that come with a smaller theory-practice gap will make it easier to identify and mitigate unintended negative impact.

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

**Supplementary Material of**
**Fundamental Bias in Inverting Random Sampling Matrices**
**with Application to Sub-sampled Newton**



The technical appendices of this paper are organized as follows.

- We recall a few technical lemmas that will be used in our proofs in Appendix A.

- The proof of Lemma 2.7 on the subspace embedding property for random sampling is given in Appendix B.

- The proof of Proposition 2.8 on coarse-grained debiasing of random sampling is given in Appendix C.

- The proof of Theorem 3.1 on fine-grained analysis of inversion bias for random sampling is given in Appendix D.

- The proof of Proposition 3.2 on fine-grained debiasing for random sampling is given in Appendix E.

- The proof concerning results on the application to de-biased SSN in Section 4 is given in Appendix F.

- Implementation details for the numerical results in Section 5 are given in Appendix G.

## A. Useful Lemmas

In this section, we introduce a few technical lemmas to be used in subsequent sections.

**Lemma A.1** (Singular value bounds of symmetric p.s.d. matrices, Zhan (2001, Theorem 2.1)). *For real symmetric p.s.d. matrices $\mathbf{J}, \mathbf{K} \in \mathbb{R}^{s \times s}$ having (ordered) singular values $\sigma_1(\mathbf{J}) \geq \sigma_2(\mathbf{J}) \geq \ldots \geq \sigma_s(\mathbf{J})$ and $\sigma_1(\mathbf{K}) \geq \sigma_2(\mathbf{K}) \geq \ldots \geq \sigma_s(\mathbf{K})$, the singular values of the difference $\mathbf{J} - \mathbf{K}$ are bounded as*

$$\sigma_j(\mathbf{J} - \mathbf{K}) \leq \sigma_j \begin{pmatrix} \mathbf{J} & 0 \\ 0 & \mathbf{K} \end{pmatrix}, \quad j = 1, \ldots, s.$$

*In particular, we obtain*

$$\|\mathbf{J} - \mathbf{K}\| \leq \max\{\|\mathbf{J}\|, \|\mathbf{K}\|\}.$$

**Lemma A.2** (Intrinsic matrix Bernstein, Tropp (2015, Theorem 7.3.1)). *Let $\mathbf{X}_1, \ldots, \mathbf{X}_n$ be $n$ independent real symmetric random matrices such that $\mathbb{E}[\mathbf{X}_i] = \mathbf{0}$, $\max_{1 \leq i \leq n} \|\mathbf{X}_i\| \leq \rho_1$, and $\sum_{i=1}^n \mathbb{E}[\mathbf{X}_i^2] \preceq \mathbf{B}$ for some symmetric p.s.d. matrix $\mathbf{B}$. Then, for any $\epsilon \geq \|\mathbf{B}\|^{1/2} + \rho_1/3$, we have*

$$\Pr\left(\left\|\sum_{i=1}^n \mathbf{X}_i\right\| \geq \epsilon\right) \leq \frac{4 \operatorname{tr} \mathbf{B}}{\|\mathbf{B}\|} \exp\left(-\frac{\epsilon^2/2}{\|\mathbf{B}\| + \rho_1 \epsilon/3}\right).$$

**Lemma A.3** (Sherman–Morrison formula). *For an invertible matrix $\mathbf{A} \in \mathbb{R}^{n \times n}$ and two vectors $\mathbf{u}, \mathbf{v} \in \mathbb{R}^n$, $\mathbf{A} + \mathbf{u}\mathbf{v}^\top$ is invertible if and only if $1 + \mathbf{v}^\top \mathbf{A}^{-1} \mathbf{u} \neq 0$ and*

$$(\mathbf{A} + \mathbf{u}\mathbf{v}^\top)^{-1} = \mathbf{A}^{-1} - \frac{\mathbf{A}^{-1}\mathbf{u}\mathbf{v}^\top \mathbf{A}^{-1}}{1 + \mathbf{v}^\top \mathbf{A}^{-1} \mathbf{u}}.$$

*Besides, it also follows that*

$$(\mathbf{A} + \mathbf{u}\mathbf{v}^\top)^{-1}\mathbf{u} = \frac{\mathbf{A}^{-1}\mathbf{u}}{1 + \mathbf{v}^\top \mathbf{A}^{-1} \mathbf{u}}.$$

**Lemma A.4** (Burkholder inequality, Burkholder (1973)). *For $\{y_i\}_{i=1}^n$ a real martingale difference sequence with respect to the increasing $\sigma$-field $\mathcal{F}_i$, we have*

$$\mathbb{E}\left[\left|\sum_{i=1}^n y_i\right|^T\right] \leq L_T \cdot \mathbb{E}\left[\left(\sum_{i=1}^n y_i^2\right)^{T/2}\right],$$

*for any integer $T \geq 2$ and some constant $L_T > 0$ independent of $n$.*

We will use Lemma A.4 to establish concentration results for resolvent/inverse matrices. In that context, the increasing $\sigma$-field $\mathcal{F}_i$ is defined with respect to the independent trials of random sampling (i.e., the rows of the random sampling matrix $\mathbf{S} \in \mathbb{R}^{m \times n}$ in Definition 2.1, see Lemma D.1 in Appendix D.3 and the discussion thereafter.

**Theorem A.5** (Hanson-Wright inequality, Vershynin (2018)). *Assume* $\mathbf{x} = (x_1, \ldots, x_n) \in \mathbb{R}^n$ *be a random vector with independent, mean-zero, sub-gaussian variables. For a matrix* $\mathbf{M} \in \mathbb{R}^{n \times n}$ *and any* $t \geq 0$, *then we have*

$$\Pr\left(\left|\mathbf{x}^\top \mathbf{M} \mathbf{x} - \mathbb{E}[\mathbf{x}^\top \mathbf{M} \mathbf{x}]\right| \geq t\right) \leq 2 \exp\left[-c \min\left(\frac{t^2}{K^4 \|\mathbf{M}\|_F^2}, \frac{t}{K^2 \|\mathbf{M}\|}\right)\right],$$

*where* $\|\cdot\|_F$ *denotes the Frobenius norm,* $K = \max_{1 \leq i \leq n} \inf\{s > 0 : \mathbb{E}[\exp(x_i^2/s^2)] \leq 2\}$, *and* $c$ *is a universal constant.*

## B. Proof of Lemma 2.7

In this section, we present the proof of Lemma 2.7. Recall that our objective is, for $\mathbf{A} \in \mathbb{R}^{n \times d}$ of rank $d$ with $n \geq d$, p.s.d. $\mathbf{C} \in \mathbb{R}^{d \times d}$, $\mathbf{S}$ some random sampling matrix with number of trials $m$ and importance sampling distribution $\{\pi_i\}_{i=1}^n$ as in Definition 2.1, $d_{\text{eff}} = \text{tr}(\mathbf{A}_{\mathbf{C}}^\top \mathbf{A}_{\mathbf{C}})$ the effective dimension of $\mathbf{A}$ given $\mathbf{C}$ with $\mathbf{A}_{\mathbf{C}} \equiv \mathbf{A}(\mathbf{A}^\top \mathbf{A} + \mathbf{C})^{-1/2}$ as in Definition 2.2, and max importance sampling approximation factor $\rho_{\max}$ as in Definition 2.3, to show that for $m \geq C \rho_{\max} d_{\text{eff}} \log(d_{\text{eff}}/\delta)/\epsilon^2$, $\mathbf{A}_{\mathbf{C}}^\top \mathbf{S}^\top \mathbf{S} \mathbf{A}_{\mathbf{C}}$ is an $(\epsilon, \delta)$-approximation of $\mathbf{A}_{\mathbf{C}}^\top \mathbf{A}_{\mathbf{C}}$. That is

$$(1 + \epsilon)^{-1} \mathbf{A}_{\mathbf{C}}^\top \mathbf{A}_{\mathbf{C}} \preceq \mathbf{A}_{\mathbf{C}}^\top \mathbf{S}^\top \mathbf{S} \mathbf{A}_{\mathbf{C}} \preceq (1 + \epsilon) \mathbf{A}_{\mathbf{C}}^\top \mathbf{A}_{\mathbf{C}}, \tag{13}$$

holds with probability at least $1 - \delta$.

First note that, for $\mathbf{A}_{\mathbf{C}} \equiv \mathbf{A}(\mathbf{A}^\top \mathbf{A} + \mathbf{C})^{-1/2} \in \mathbb{R}^{n \times d}$, we have

$$\mathbf{A}_{\mathbf{C}}^\top \mathbf{S}^\top \mathbf{S} \mathbf{A}_{\mathbf{C}} - \mathbf{A}_{\mathbf{C}}^\top \mathbf{A}_{\mathbf{C}} = \sum_{s=1}^m \left(\frac{\mathbf{a}_{\mathbf{C}i_s} \mathbf{a}_{\mathbf{C}i_s}^\top}{m \pi_{i_s}} - \frac{\mathbf{A}_{\mathbf{C}}^\top \mathbf{A}_{\mathbf{C}}}{m}\right) \equiv \sum_{s=1}^m \mathbf{F}_s,$$

where $\mathbf{a}_{\mathbf{C}i}^\top \in \mathbb{R}^d$ denotes the $i^{th}$ row of $\mathbf{A}_{\mathbf{C}}$. Given $\mathbf{A}_{\mathbf{C}}$, it follows from Definition 2.1 that the indices $i_s$ are i.i.d. drawn with replacement from the index set $\{1, \ldots, n\}$. We would like to apply the Intrinsic matrix Bernstein, Lemma A.2, to bound the sum of random matrices $\mathbf{A}_{\mathbf{C}}^\top \mathbf{S}^\top \mathbf{S} \mathbf{A}_{\mathbf{C}} - \mathbf{A}_{\mathbf{C}}^\top \mathbf{A}_{\mathbf{C}} = \sum_{s=1}^m \mathbf{F}_s$. To this end, note first that given $\mathbf{A}_{\mathbf{C}}$ we have

$$\mathbb{E}[\mathbf{F}_s] = \mathbf{0}_d. \tag{14}$$

It then remains to bound (i) the spectral norm $\|\mathbf{F}_s\|$ and (ii) the sum $\sum_{s=1}^m \mathbb{E}[\mathbf{F}_s^2]$ in the sense of p.s.d. matrices. By Lemma A.1 and the triangle inequality, for $s = 1, \ldots, m$, we get

$$\|\mathbf{F}_s\| \leq \max_{1 \leq i \leq n}\left\{\left\|\frac{\mathbf{a}_{\mathbf{C}i} \mathbf{a}_{\mathbf{C}i}^\top}{m \pi_i}\right\|, \frac{1}{m}\right\} = \frac{1}{m} \max_{1 \leq i \leq n}\left\{\frac{\|\mathbf{a}_{\mathbf{C}i}\|^2}{\pi_i}, 1\right\} = \frac{\rho_{\max} d_{\text{eff}}}{m} \equiv \rho_1,$$

where we recall $\|\mathbf{A}_{\mathbf{C}}^\top \mathbf{A}_{\mathbf{C}}\| \leq 1$ and $\rho_{\max} = \max_{1 \leq i \leq n} \|\mathbf{a}_{\mathbf{C}i}\|^2/(\pi_i d_{\text{eff}}) \geq 1$ is the max importance sampling approximation factor in Definition 2.3.

Further note that, per its definition $\mathbf{F}_s = \frac{\mathbf{a}_{\mathbf{C}i_s} \mathbf{a}_{\mathbf{C}i_s}^\top}{m \pi_{i_s}} - \frac{\mathbf{A}_{\mathbf{C}}^\top \mathbf{A}_{\mathbf{C}}}{m}$, we have

$$\left(\mathbf{F}_s + \frac{\mathbf{A}_{\mathbf{C}}^\top \mathbf{A}_{\mathbf{C}}}{m}\right)^2 = \frac{\|\mathbf{a}_{\mathbf{C}i_s}\|^2 \cdot \mathbf{a}_{\mathbf{C}i_s} \mathbf{a}_{\mathbf{C}i_s}^\top}{m^2 \pi_{i_s}^2} = \mathbf{F}_s^2 + \mathbf{F}_s \frac{\mathbf{A}_{\mathbf{C}}^\top \mathbf{A}_{\mathbf{C}}}{m} + \frac{\mathbf{A}_{\mathbf{C}}^\top \mathbf{A}_{\mathbf{C}}}{m} \mathbf{F}_s + \frac{(\mathbf{A}_{\mathbf{C}}^\top \mathbf{A}_{\mathbf{C}})^2}{m^2}. \tag{15}$$

Taking expectations on both sides of (15) and applying (14), we get

$$\mathbb{E}[\mathbf{F}_s^2] + \frac{(\mathbf{A}_{\mathbf{C}}^\top \mathbf{A}_{\mathbf{C}})^2}{m^2} = \sum_{i=1}^n \pi_i \frac{\|\mathbf{a}_{\mathbf{C}i}\|^2 \cdot \mathbf{a}_{\mathbf{C}i} \mathbf{a}_{\mathbf{C}i}^\top}{m^2 \pi_i^2} \preceq \frac{\rho_{\max} d_{\text{eff}}}{m^2} \mathbf{A}_{\mathbf{C}}^\top \mathbf{A}_{\mathbf{C}}.$$

Thus,

$$\sum_{s=1}^m \mathbb{E}[\mathbf{F}_s^2] \preceq \frac{\rho_{\max} d_{\text{eff}}}{m} \mathbf{A}_{\mathbf{C}}^\top \mathbf{A}_{\mathbf{C}} \equiv \mathbf{P},$$

for which we have $\|\mathbf{P}\| = \frac{\rho_{\max} d_{\text{eff}}}{m}\|\mathbf{A}_{\mathbf{C}}^\top \mathbf{A}_{\mathbf{C}}\|$ and $\text{tr}(\mathbf{P}) = \frac{\rho_{\max} d_{\text{eff}}^2}{m}$, so that $\frac{\text{tr}\,\mathbf{P}}{\|\mathbf{P}\|} = \frac{d_{\text{eff}}}{\|\mathbf{A}_{\mathbf{C}}^\top \mathbf{A}_{\mathbf{C}}\|}$.

Recall that $\|\mathbf{A}_{\mathbf{C}}^\top \mathbf{A}_{\mathbf{C}}\| \leq 1$, applying Lemma A.2, we obtain

$$1 - \Pr\left(\frac{1}{1+\epsilon}\mathbf{A}_{\mathbf{C}}^\top \mathbf{A}_{\mathbf{C}} \preceq \mathbf{A}_{\mathbf{C}}^\top \mathbf{S}^\top \mathbf{S} \mathbf{A}_{\mathbf{C}} \preceq (1+\epsilon)\mathbf{A}_{\mathbf{C}}^\top \mathbf{A}_{\mathbf{C}}\right)$$

$$\leq \Pr\left(\|\mathbf{A}_{\mathbf{C}}^\top \mathbf{S}^\top \mathbf{S} \mathbf{A}_{\mathbf{C}} - \mathbf{A}_{\mathbf{C}}^\top \mathbf{A}_{\mathbf{C}}\| > \epsilon\right) = \Pr\left(\left\|\sum_{s=1}^m \mathbf{F}_s\right\| > \epsilon\right)$$

$$\leq 4\|\mathbf{A}_{\mathbf{C}}^\top \mathbf{A}_{\mathbf{C}}\|^{-1} d_{\text{eff}} \exp\left(-\frac{\epsilon^2/2}{\frac{\rho_{\max} d_{\text{eff}}}{m}\|\mathbf{A}_{\mathbf{C}}^\top \mathbf{A}_{\mathbf{C}}\| + \frac{\rho_{\max} d_{\text{eff}}}{m}\epsilon/3}\right)$$

$$= 4\|\mathbf{A}_{\mathbf{C}}^\top \mathbf{A}_{\mathbf{C}}\|^{-1} d_{\text{eff}} \exp\left(-\frac{\epsilon^2}{\frac{\rho_{\max} d_{\text{eff}}}{m}(2\|\mathbf{A}_{\mathbf{C}}^\top \mathbf{A}_{\mathbf{C}}\| + 2\epsilon/3)}\right).$$

Clearly, $\Pr(\frac{1}{1+\epsilon}\mathbf{A}_{\mathbf{C}}^\top \mathbf{A}_{\mathbf{C}} \preceq \mathbf{A}_{\mathbf{C}}^\top \mathbf{S}^\top \mathbf{S} \mathbf{A}_{\mathbf{C}} \preceq (1+\epsilon)\mathbf{A}_{\mathbf{C}}^\top \mathbf{A}_{\mathbf{C}}) \leq 1 - \delta$ holds if

$$4\|\mathbf{A}_{\mathbf{C}}^\top \mathbf{A}_{\mathbf{C}}\|^{-1} d_{\text{eff}} \exp\left(-\frac{\epsilon^2}{\frac{\rho_{\max} d_{\text{eff}}}{m}(2\|\mathbf{A}_{\mathbf{C}}^\top \mathbf{A}_{\mathbf{C}}\| + 2\epsilon/3)}\right) \leq \delta, \tag{16}$$

that can be rewritten as

$$m \geq \frac{\rho_{\max} d_{\text{eff}}(2\|\mathbf{A}_{\mathbf{C}}^\top \mathbf{A}_{\mathbf{C}}\| + 2\epsilon/3)}{\epsilon^2} \log\left(\frac{4\|\mathbf{A}_{\mathbf{C}}^\top \mathbf{A}_{\mathbf{C}}\|^{-1} d_{\text{eff}}}{\delta}\right). \tag{17}$$

Since $\epsilon \leq 1$ and $\|\mathbf{A}_{\mathbf{C}}^\top \mathbf{A}_{\mathbf{C}}\| \leq 1$, (17) holds if $m \geq \frac{8\rho d_{\text{eff}}}{3\epsilon^2} \log(\frac{4\|\mathbf{A}_{\mathbf{C}}^\top \mathbf{A}_{\mathbf{C}}\|^{-1} d_{\text{eff}}}{\delta})$.

Lastly, let us check the condition $\epsilon \geq \|\mathbf{P}\|^{1/2} + \rho_1/3$ in Lemma A.2 is satisfied. Solving (16) for $\epsilon$, we get

$$4\|\mathbf{A}_{\mathbf{C}}^\top \mathbf{A}_{\mathbf{C}}\|^{-1} d_{\text{eff}} \exp\left(-\frac{\epsilon^2}{\frac{\rho_{\max} d_{\text{eff}}}{m}(2\|\mathbf{A}_{\mathbf{C}}^\top \mathbf{A}_{\mathbf{C}}\| + 2\epsilon/3)}\right) \leq \delta$$

$$\implies \epsilon^2 - \frac{2\rho_{\max} d_{\text{eff}}}{3m}\log(4\|\mathbf{A}_{\mathbf{C}}^\top \mathbf{A}_{\mathbf{C}}\|^{-1} d_{\text{eff}}/\delta)\epsilon - 6\frac{\rho_{\max} d_{\text{eff}}\|\mathbf{A}_{\mathbf{C}}^\top \mathbf{A}_{\mathbf{C}}\|}{3m}\log(4\|\mathbf{A}_{\mathbf{C}}^\top \mathbf{A}_{\mathbf{C}}\|^{-1} d_{\text{eff}}/\delta) \geq 0$$

$$\implies \epsilon^2 - 2v\epsilon - 6\|\mathbf{A}_{\mathbf{C}}^\top \mathbf{A}_{\mathbf{C}}\|v \geq 0$$

$$\implies \epsilon \geq v + \sqrt{v^2 + 6\|\mathbf{A}_{\mathbf{C}}^\top \mathbf{A}_{\mathbf{C}}\|v},$$

where $v = \frac{\rho_{\max} d_{\text{eff}}}{3m}\log(\frac{4\|\mathbf{A}_{\mathbf{C}}^\top \mathbf{A}_{\mathbf{C}}\|^{-1} d_{\text{eff}}}{\delta})$. As $\delta < 1$ and $4\|\mathbf{A}_{\mathbf{C}}^\top \mathbf{A}_{\mathbf{C}}\|^{-1} d_{\text{eff}} > e$, we have $\log(\frac{4\|\mathbf{A}_{\mathbf{C}}^\top \mathbf{A}_{\mathbf{C}}\|^{-1} d_{\text{eff}}}{\delta}) \geq 1$. As such, we get $v \geq \rho_1/3$ and $6\|\mathbf{A}_{\mathbf{C}}^\top \mathbf{A}_{\mathbf{C}}\|v \geq \|\mathbf{P}\|$, so that $\epsilon \geq \|\mathbf{P}\|^{1/2} + \rho_1/3$ holds. This concludes the proof of Lemma 2.7. $\square$

Given the subspace embedding result in Lemma 2.7, it can be shown that the inversion $(\mathbf{A}^\top \mathbf{S}^\top \mathbf{S} \mathbf{A} + \mathbf{C})^{-1}$ also satisfies a similar subspace embedding property and is close to the (population) inversion $(\mathbf{A}^\top \mathbf{A} + \mathbf{C})^{-1}$. This is given in the following result.

**Lemma B.1.** *Under the settings and notations of Lemma 2.7, we have that $(\mathbf{A}^\top \mathbf{S}^\top \mathbf{S} \mathbf{A} + \mathbf{C})^{-1}$ is an $(\epsilon, \delta)$-approximation of $(\mathbf{A}^\top \mathbf{A} + \mathbf{C})^{-1}$.*

*Proof of Lemma B.1.* Denote $\mathbf{H} = \mathbf{A}^\top \mathbf{A} + \mathbf{C}$ (so that $\|\mathbf{H}^{-\frac{1}{2}}\mathbf{C}\mathbf{H}^{-\frac{1}{2}}\| \leq 1$), it then follows from Lemma 2.7 that

$\mathbf{A}_\mathbf{C}^\top\mathbf{S}^\top\mathbf{S}\mathbf{A}_\mathbf{C}$ is an $(\epsilon,\delta)$-approximation of $\mathbf{A}_\mathbf{C}^\top\mathbf{A}_\mathbf{C}$, that is,

$$\frac{1}{1+\epsilon}\mathbf{A}_\mathbf{C}^\top\mathbf{A}_\mathbf{C} \preceq \mathbf{A}_\mathbf{C}^\top\mathbf{S}^\top\mathbf{S}\mathbf{A}_\mathbf{C} \preceq (1+\epsilon)\mathbf{A}_\mathbf{C}^\top\mathbf{A}_\mathbf{C}$$

$$\Rightarrow \frac{1}{1+\epsilon}(\mathbf{A}_\mathbf{C}^\top\mathbf{A}_\mathbf{C}+\mathbf{H}^{-\frac{1}{2}}\mathbf{C}\mathbf{H}^{-\frac{1}{2}}) \preceq \mathbf{A}_\mathbf{C}^\top\mathbf{S}^\top\mathbf{S}\mathbf{A}_\mathbf{C}+\mathbf{H}^{-\frac{1}{2}}\mathbf{C}\mathbf{H}^{-\frac{1}{2}} \preceq (1+\epsilon)(\mathbf{A}_\mathbf{C}^\top\mathbf{A}_\mathbf{C}+\mathbf{H}^{-\frac{1}{2}}\mathbf{C}\mathbf{H}^{-\frac{1}{2}})$$

$$\Rightarrow \frac{1}{1+\epsilon}\mathbf{H} \preceq \mathbf{A}^\top\mathbf{S}^\top\mathbf{S}\mathbf{A}+\mathbf{C} \preceq (1+\epsilon)\mathbf{H}.$$

$$\Rightarrow \frac{1}{1+\epsilon}\mathbf{H}^{-1} \preceq (\mathbf{A}^\top\mathbf{S}^\top\mathbf{S}\mathbf{A}+\mathbf{C})^{-1} \preceq (1+\epsilon)\mathbf{H}^{-1},$$

where we recall the definition $\mathbf{A}_\mathbf{C} = \mathbf{A}(\mathbf{A}^\top\mathbf{A}+\mathbf{C})^{-1/2}$. This concludes the proof of Lemma B.1. $\qquad\square$

## C. Proof of Proposition 2.8

The proof of Proposition 2.8 follows roughly the same line as that of Dereziński et al. (2021a, Theorem 6) and Dereziński et al. (2021b, Theorem 11). It is provided here for completeness.

Here, we focus on the differences from the proof in Dereziński et al. (2021a;b) by considering now $\mathbf{S}$ under study is a random sampling matrix as in Definition 2.1.

To start with, we introduce the following condition that is crucial to the proof in Dereziński et al. (2021a;b) and of our Proposition 2.8 here.

**Condition C.1** (Concentration property of random vector **s**). Given $\mathbf{V}\in\mathbb{R}^{n\times d}$, the $n$-dimensional random vector **s** satisfies $\mathrm{Var}[\mathbf{s}^\top\mathbf{V}\mathbf{B}\mathbf{V}^\top\mathbf{s}] \leq \alpha\cdot\mathrm{tr}(\mathbf{V}\mathbf{B}^2\mathbf{V}^\top)$ for all p.s.d. matrices $\mathbf{B}\in\mathbb{R}^{d\times d}$ and some $\alpha>0$.

The proof of Proposition 2.8 then comes in the following two steps:

1. construct a high probability event $\zeta$ (via subspace embedding-type results in Lemma 2.7), on which the inverse $(\mathbf{A}^\top\mathbf{S}^\top\mathbf{S}\mathbf{A}+\mathbf{C})^{-1}$ exists (in particular for $\mathbf{C}=\mathbf{0}_d$); and

2. conditioned on that event $\zeta$, bound the spectral norm $\|\mathbf{I}_d - \mathbb{E}_\zeta[\tilde{\mathbf{Q}}]\mathbf{H}\|$ via "leave-one-out" type analysis, for $\mathbf{H} = \mathbf{A}^\top\mathbf{A}+\mathbf{C}$ and $\tilde{\mathbf{Q}} = (\gamma\mathbf{A}^\top\mathbf{S}^\top\mathbf{S}\mathbf{A}+\mathbf{C})^{-1}$ with $\gamma = \frac{m}{m-d_{\mathrm{eff}}}$.

Let us start with the first step, for random sampling in Definition 2.1, denote $\mathbf{x}_s^\top = \mathbf{e}_{i_s}^\top/\sqrt{\pi_{i_s}}\mathbf{A}\in\mathbb{R}^d$ the $i^{th}$ row of the sketch $\tilde{\mathbf{A}}$, so that $\mathbb{E}[\mathbf{x}_s\mathbf{x}_s^\top] = \mathbf{A}^\top\mathbf{A}$. Denote

$$\mathbf{H} = \mathbf{A}^\top\mathbf{A}+\mathbf{C}, \quad \tilde{\mathbf{Q}} = (\gamma\mathbf{A}^\top\mathbf{S}^\top\mathbf{S}\mathbf{A}+\mathbf{C})^{-1} = \left(\sum_{s=1}^m \frac{\gamma}{m}\mathbf{x}_s\mathbf{x}_s^\top + \mathbf{C}\right)^{-1}, \quad \gamma = \frac{m}{m-d_{\mathrm{eff}}}, \tag{18}$$

and

$$\tilde{\mathbf{Q}}_{-s} = \left(\sum_{j\neq s}\frac{\gamma}{m}\mathbf{x}_j\mathbf{x}_j^\top + \mathbf{C}\right)^{-1},$$

that is *independent* of $\mathbf{x}_s$.

Without loss of generality, assume that $t = m/3$ is an integer, and define the following events:

$$\zeta_j: \sum_{s=t(j-1)+1}^{tj}\frac{1}{t}\mathbf{x}_s\mathbf{x}_s^\top \succeq \frac{1}{2}\mathbf{A}^\top\mathbf{A}, \quad j=1,2,3, \quad \zeta = \bigcap_{j=1}^3\zeta_j. \tag{19}$$

Recall that $\gamma = \frac{m}{m-d_{\mathrm{eff}}} > 1$, the events $\zeta_j$ imply

$$\sum_{s=t(j-1)+1}^{tj}\frac{\gamma}{t}\mathbf{x}_s\mathbf{x}_s^\top \succeq \frac{1}{2}\mathbf{A}^\top\mathbf{A}, \quad j=1,2,3.$$

Here, the event $\zeta_j$ presents that the (weighted) average of the rank-one matrices $\mathbf{x}_s\mathbf{x}_s^\top$ over the corresponding $j$-th third of indices $1,\ldots,n$ forms a sketch (of size $t = m/3$) of $\mathbf{A}^\top\mathbf{A}$ that is a "lower" 1/2-spectral-approximation of $\mathbf{A}^\top\mathbf{A}$ in the sense of Definition 2.4.

In the case of random sampling in Definition 2.1, the events $\zeta_1$, $\zeta_2$, and $\zeta_3$ are independent. As such, for each $s \in \{1,\ldots,m\}$, there exists an index $j = j(s) \in \{1,2,3\}$ such that

1. $\zeta_j$ is independent of $\mathbf{x}_s$; and

2. conditioned on $\zeta_j$ one has $\tilde{\mathbf{Q}} \preceq \tilde{\mathbf{Q}}_{-s} \preceq 6\mathbf{H}^{-1}$, which, for $m > 2d_{\mathrm{eff}}$, further leads to $\gamma\mathbf{H}^{1/2}\tilde{\mathbf{Q}}_{-s}\mathbf{H}^{1/2} \preceq 12\mathbf{I}_d$.

Following the decomposition of $\mathbb{E}_\zeta[\tilde{\mathbf{Q}}]$ in Dereziński et al. (2021a;b) with $\tilde{\gamma}_s = 1 + \frac{\gamma}{m}\mathbf{x}_s^\top\tilde{\mathbf{Q}}_{-s}\mathbf{x}_s$, we write

$$\mathbf{I}_d - \mathbb{E}_\zeta[\tilde{\mathbf{Q}}]\mathbf{H} = \underbrace{\mathbb{E}_\zeta[\tilde{\mathbf{Q}}_{-s}(\mathbf{x}_s\mathbf{x}_s^\top - \mathbf{A}^\top\mathbf{A})]}_{\tilde{\mathbf{Z}}_0} + \underbrace{\mathbb{E}_\zeta[\tilde{\mathbf{Q}}_{-s} - \tilde{\mathbf{Q}}]\mathbf{A}^\top\mathbf{A}}_{\tilde{\mathbf{Z}}_1} + \underbrace{\mathbb{E}_\zeta\left[\left(\frac{\gamma}{\tilde{\gamma}_s} - 1\right)\tilde{\mathbf{Q}}_{-s}\mathbf{x}_s\mathbf{x}_s^\top\right]}_{\tilde{\mathbf{Z}}_2}. \tag{20}$$

Recall that $\tilde{\mathbf{Q}}$ in (18) is p.s.d. symmetric, to establish the result in Proposition 2.8, it suffices to bound the spectral norm

$$\|\mathbf{I}_d - \mathbf{H}^{\frac{1}{2}}\mathbb{E}_\zeta[\tilde{\mathbf{Q}}]\mathbf{H}^{\frac{1}{2}}\| \leq \|\mathbf{H}^{\frac{1}{2}}\tilde{\mathbf{Z}}_0\mathbf{H}^{-\frac{1}{2}}\| + \|\mathbf{H}^{\frac{1}{2}}\tilde{\mathbf{Z}}_1\mathbf{H}^{-\frac{1}{2}}\| + \|\mathbf{H}^{\frac{1}{2}}\tilde{\mathbf{Z}}_2\mathbf{H}^{-\frac{1}{2}}\|, \tag{21}$$

for $\tilde{\mathbf{Z}}_0, \tilde{\mathbf{Z}}_1, \tilde{\mathbf{Z}}_2$ defined in (20).

Without loss of generality, assume that the events $\zeta_1$ and $\zeta_2$ are independent of $\mathbf{x}_s$, and set $\zeta' = \zeta_1 \bigcap \zeta_2$ as well as $\delta_3 = \Pr(\neg\zeta_3)$. To bound (21), we first recall the following results from the proof of Dereziński et al. (2021a;b).

**Lemma C.2.** *Under the settings and notations of Proposition 2.8, we have,*

1. *for a p.s.d. random matrix $\mathbf{M}$ (or a non-negative random variable) living in the probability space of $\mathbf{S}$,*

$$\mathbb{E}_\zeta[\mathbf{M}] = \frac{\mathbb{E}[(\prod_{j=1}^3 \mathbf{1}_{\zeta_j})\mathbf{M}]}{\Pr(\zeta)} \preceq \frac{1}{1-\delta}\mathbb{E}[\mathbf{1}_{\zeta'}\mathbf{M}] \preceq 2\mathbb{E}_{\zeta'}[\mathbf{M}], \tag{22}$$

   *where $\mathbf{1}_{\zeta_j}$ is the indicator of the event $\zeta_j$.*

2. $\|\mathbf{H}^{1/2}\tilde{\mathbf{Z}}_1\mathbf{H}^{-1/2}\| = O(\frac{1}{m})$.

Next, we bound the terms invoking $\tilde{\mathbf{Z}}_0$ and $\tilde{\mathbf{Z}}_2$ in (21), particularly by emphasizing the difference between our proof of Proposition 2.8 here from that of Dereziński et al. (2021a;b). By $\delta_3 \leq \frac{1}{2}$ and

$$\tilde{\mathbf{Z}}_0 = -\frac{1}{1-\delta_3}\mathbb{E}_{\zeta'}[\tilde{\mathbf{Q}}_{-s}(\mathbf{x}_s\mathbf{x}_s^\top - \mathbf{A}^\top\mathbf{A}) \cdot \mathbf{1}_{\neg\zeta_3}],$$

it follows that

$$\|\mathbf{H}^{\frac{1}{2}}\tilde{\mathbf{Z}}_0\mathbf{H}^{-\frac{1}{2}}\| \leq 12\mathbb{E}_{\zeta'}[(1 + \mathbf{x}_s^\top\mathbf{H}^{-1}\mathbf{x}_s) \cdot \mathbf{1}_{\neg\zeta_3}].$$

We now need to bound the (conditional) expectation $\mathbb{E}_{\zeta'}[(1 + \mathbf{x}_s^\top\mathbf{H}^{-1}\mathbf{x}_s) \cdot \mathbf{1}_{\neg\zeta_3}]$. To do this, we resort to Condition C.1 with an appropriate choice of $\alpha$. Note that

$$\mathbb{E}[\mathbf{x}_s^\top\mathbf{H}^{-1}\mathbf{x}_s] = \mathrm{tr}(\mathbf{H}^{-1}\mathbf{A}^\top\mathbf{A}) = d_{\mathrm{eff}},$$

per Definition 2.2, it follows that

$$\begin{aligned}
\mathrm{Var}[\mathbf{x}_s^\top\mathbf{H}^{-1}\mathbf{x}_s] &= \mathrm{Var}[\mathbf{e}_{i_s}^\top/\sqrt{\pi_{i_s}}\mathbf{A}\mathbf{H}^{-1}\mathbf{A}^\top\mathbf{e}_{i_s}/\sqrt{\pi_{i_s}}] \\
&= \mathbb{E}[(\mathbf{x}_s^\top\mathbf{H}^{-1}\mathbf{x}_s)^2] - (\mathbb{E}[\mathbf{x}_s^\top\mathbf{H}^{-1}\mathbf{x}_s])^2 = \sum_{i=1}^n \frac{(\mathbf{a}_i^\top\mathbf{H}^{-1}\mathbf{a}_i)^2}{\pi_i} - d_{\mathrm{eff}}^2 \\
&\leq \rho_{\max}d_{\mathrm{eff}}^2 - d_{\mathrm{eff}}^2 = O(\rho_{\max}d_{\mathrm{eff}}^2),
\end{aligned}$$

for max importance sampling approximation factor $\rho_{\max}$ in Definition 2.3, which implies that $\alpha$ in Condition C.1 satisfies

$$\alpha = O(\rho_{\max} d_{\text{eff}}).$$

Further, by Chebyshev's inequality, one has, for $x \geq 2d_{\text{eff}}$, that $\Pr(\mathbf{x}_s^\top \mathbf{H}^{-1} \mathbf{x}_s \geq x \mid \zeta') \leq \alpha d_{\text{eff}}/x^2$. This further leads to, for $\delta_3 \leq 1/m^3$,

$$\|\mathbf{H}^{\frac{1}{2}} \tilde{\mathbf{Z}}_0 \mathbf{H}^{-\frac{1}{2}}\| \leq 12 \int_0^\infty \Pr(\mathbf{x}_s^\top \mathbf{H}^{-1} \mathbf{x}_s \cdot \mathbf{1}_{\neg \zeta_3} \geq x \mid \zeta') dx + O\Big(\frac{1}{m^3}\Big)$$

$$\leq 24 m^2 \delta_3 + 12 \int_{2m^2}^\infty \Pr(\mathbf{x}_s^\top \mathbf{H}^{-1} \mathbf{x}_s \geq x \mid \zeta') dx + O\Big(\frac{1}{m^3}\Big)$$

$$\leq O\Big(\frac{1}{m}\Big) + \frac{\alpha d_{\text{eff}}}{m^2} = O\left(\frac{\alpha \sqrt{d_{\text{eff}}}}{m}\right).$$

We then move on to bound the last term concerning $\tilde{\mathbf{Z}}_2$ in (21). Applying Cauchy-Schwarz inequality twice, we get

$$\|\mathbf{H}^{\frac{1}{2}} \tilde{\mathbf{Z}}_2 \mathbf{H}^{-\frac{1}{2}}\| \leq \underbrace{\sqrt{\mathbb{E}_\zeta \left[ (\tilde{\gamma}_s - \gamma)^2 \right]}}_{\tilde{T}_1} \cdot \underbrace{\sup_{\|\mathbf{u}\|=1} \sqrt[4]{\mathbb{E}_\zeta \left[ (\mathbf{u}^\top \mathbf{H}^{\frac{1}{2}} \tilde{\mathbf{Q}}_{-s} \mathbf{x}_s)^4 \right]}}_{\tilde{T}_2} \cdot \underbrace{\sup_{\|\mathbf{u}\|=1} \sqrt[4]{\mathbb{E}_\zeta \left[ (\mathbf{u}^\top \mathbf{H}^{-\frac{1}{2}} \mathbf{x}_s)^4 \right]}}_{\tilde{T}_3}. \tag{23}$$

We start by bounding the term $\tilde{T}_3$ using Condition C.1 with $\mathbf{B} = \mathbf{A} \mathbf{H}^{-1/2} \mathbf{u} \mathbf{u}^\top \mathbf{H}^{-1/2} \mathbf{A}^\top$. Noting that $\text{tr}(\mathbf{B}) = \mathbf{u}^\top \mathbf{H}^{-1/2} \mathbf{A}^\top \mathbf{A} \mathbf{H}^{-1/2} \mathbf{u} \leq 1$, we get

$$\mathbb{E}_\zeta \left[ (\mathbf{u}^\top \mathbf{H}^{-\frac{1}{2}} \mathbf{x}_s)^4 \right] \leq 2 \mathbb{E}_{\zeta'} \left[ (\mathbf{u}^\top \mathbf{H}^{\frac{1}{2}} \mathbf{x}_s)^4 \right] = 2 \mathbb{E}[(\mathbf{x}_s^\top \mathbf{H}^{-\frac{1}{2}} \mathbf{u} \mathbf{u}^\top \mathbf{H}^{-\frac{1}{2}} \mathbf{x}_s)^2]$$

$$= 2 \text{Var}_{\zeta'}[(\mathbf{x}_s^\top \mathbf{H}^{-\frac{1}{2}} \mathbf{u} \mathbf{u}^\top \mathbf{H}^{-\frac{1}{2}} \mathbf{x}_s)^2] + 2(\mathbb{E}_{\zeta'}[\mathbf{x}_s^\top \mathbf{H}^{-\frac{1}{2}} \mathbf{u} \mathbf{u}^\top \mathbf{H}^{-\frac{1}{2}} \mathbf{x}_s])^2$$

$$\leq 2 \sum_{i=1}^n \frac{(\mathbf{a}_i^\top \mathbf{H}^{-\frac{1}{2}} \mathbf{u} \mathbf{u}^\top \mathbf{H}^{-\frac{1}{2}} \mathbf{a}_i)^2}{\pi_i} + 2(\text{tr}(\mathbf{B}))^2$$

$$\leq 2 \rho_{\max} d_{\text{eff}} \sum_{i=1}^n \mathbf{u}^\top \mathbf{H}^{-\frac{1}{2}} \mathbf{a}_i \mathbf{a}_i^\top \mathbf{H}^{-\frac{1}{2}} \mathbf{u} + 2(\text{tr}(\mathbf{B}))^2$$

$$\leq 2 \rho_{\max} d_{\text{eff}} \, \text{tr}(\mathbf{B}) + 2(\text{tr}(\mathbf{B}))^2 \leq 2(\rho d_{\text{eff}} + 1) = O(\alpha + 1),$$

for $\alpha = O(\rho_{\max} d_{\text{eff}})$ in Condition C.1. This results in $\tilde{T}_3 = O(\sqrt[4]{\alpha + 1})$. Similarly, we bound $\tilde{T}_2$ by taking $\mathbf{B} = \mathbf{A} \tilde{\mathbf{Q}}_{-s} \mathbf{H}^{1/2} \mathbf{u} \mathbf{u}^\top \mathbf{H}^{1/2} \tilde{\mathbf{Q}}_{-s} \mathbf{A}^\top$ in Condition C.1. It follows $\text{tr}(\mathbf{B}) \leq \mathbf{u}^\top (\mathbf{H}^{1/2} \tilde{\mathbf{Q}}_{-s} \mathbf{H}^{1/2})^2 \mathbf{u} \leq 6^2$ and $\text{tr}(\mathbf{B}^2) \leq 6^4$, so that $\tilde{T}_2 = O(\sqrt[4]{\alpha + 1})$.

It thus remains to bound the first term $\tilde{T}_1$ in (23). Noting $\bar{\gamma} = \mathbb{E}_{\zeta'}[\tilde{\gamma}_s] = 1 + \frac{\gamma}{m} \text{tr}(\mathbb{E}_{\zeta'}[\tilde{\mathbf{Q}}_{-s}] \mathbf{A}^\top \mathbf{A})$, we write

$$\mathbb{E}_\zeta \left[ (\tilde{\gamma}_s - \gamma)^2 \right] \leq 2(\gamma - \bar{\gamma})^2 + \frac{2\gamma^2}{m^2} \mathbb{E}_{\zeta'}[(\text{tr}(\tilde{\mathbf{Q}}_{-s} - \mathbb{E}_{\zeta'}[\tilde{\mathbf{Q}}_{-s}]) \mathbf{A}^\top \mathbf{A})^2]$$

$$+ \frac{2\gamma^2}{m^2} \mathbb{E}_{\zeta'}[(\text{tr}(\mathbf{A} \tilde{\mathbf{Q}}_{-s} \mathbf{A}^\top) - \mathbf{x}_s^\top \tilde{\mathbf{Q}}_{-s} \mathbf{x}_s)^2].$$

Analogously as above, letting $\mathbf{B} = \mathbf{A} \tilde{\mathbf{Q}}_{-s} \mathbf{A}^\top$ so that $\text{tr}(\mathbf{B}^2) \leq 36 d_{\text{eff}}$ in Condition C.1, we have, for $m \geq 2 d_{\text{eff}}$ that

$$\frac{2\gamma^2}{m^2} \mathbb{E}_{\zeta'} \left[ \left( \text{tr}(\mathbf{A} \tilde{\mathbf{Q}}_{-s} \mathbf{A}^\top) - \mathbf{x}_s^\top \tilde{\mathbf{Q}}_{-s} \mathbf{x}_s \right)^2 \right] = \frac{2\gamma^2}{m^2} \mathbb{E}_{\zeta'} \left[ \sum_{i=1}^n \pi_i (\text{tr}(\mathbf{A} \mathbf{Q}_{-s} \mathbf{A}^\top) - \frac{\mathbf{a}_i^\top \tilde{\mathbf{Q}}_{-s} \mathbf{a}_i}{\pi_i})^2 \right]$$

$$= \frac{2\gamma^2}{m^2} \mathbb{E}_{\zeta'} \left[ (\text{tr}(\mathbf{A} \mathbf{Q}_{-s} \mathbf{A}^\top))^2 \right] + \frac{2\gamma^2}{m^2} \mathbb{E}_{\zeta'} \left[ \sum_{i=1}^n \frac{(\mathbf{a}_i^\top \tilde{\mathbf{Q}}_{-s} \mathbf{a}_i)^2}{\pi_i} \right] - \frac{4\gamma^2}{m^2} \mathbb{E}_{\zeta'} \left[ (\text{tr}(\mathbf{A} \mathbf{Q}_{-s} \mathbf{A}^\top))^2 \right]$$

$$\leq \frac{72 \gamma^2 \rho_{\max} d_{\text{eff}}^2}{m^2} - \frac{2\gamma^2}{m^2} \mathbb{E}_{\zeta'} \left[ (\text{tr}(\mathbf{A} \mathbf{Q}_{-s} \mathbf{A}^\top))^2 \right]$$

$$\leq \frac{72 \gamma^2 \rho_{\max} d_{\text{eff}}^2}{m^2} \leq \frac{288 \rho_{\max} d_{\text{eff}}^2}{m^2} = O\left(\frac{\alpha d_{\text{eff}}}{m^2}\right),$$

with again $\alpha = O(\rho_{\max} d_{\text{eff}})$ in Condition C.1.

This, following the line of arguments in Dereziński et al. (2021a), further leads to

$$\mathbb{E}_\zeta \left[ (\tilde{\gamma}_s - \bar{\gamma})^2 \right] \leq O\left( \frac{\alpha d_{\text{eff}}}{m^2} \right),$$

which, together with $|\gamma - \bar{\gamma}| = O(\sqrt{\alpha d_{\text{eff}}}/m)$, yields that $\tilde{T}_1 = O(\sqrt{\alpha d_{\text{eff}}}/m)$.

This allows us to conclude that

$$\|\mathbf{H}^{\frac{1}{2}} \tilde{\mathbf{Z}}_2 \mathbf{H}^{-\frac{1}{2}}\| \leq \tilde{T}_1 \cdot \tilde{T}_2 \cdot \tilde{T}_3 = O\left( \frac{\alpha\sqrt{d_{\text{eff}}}}{m} \right).$$

Putting everything together, we conclude that

$$\|\mathbf{I}_d - \mathbf{H}^{\frac{1}{2}} \mathbb{E}_\zeta[\tilde{\mathbf{Q}}] \mathbf{H}^{\frac{1}{2}}\| = O\left( \frac{\alpha\sqrt{d_{\text{eff}}}}{m} \right),$$

with $\alpha = O(\rho_{\max} d_{\text{eff}})$ in Condition C.1. This concludes the proof of Proposition 2.8. $\qquad\square$

# D. Proof and Discussions of Theorem 3.1

In this section, we start by presenting in Appendix D.1 the intuition for the self-consistent equation in (4) of Theorem 3.1. The detailed proof of Theorem 3.1 is given in Appendix D.2. In Appendix D.3, we provide discussions and auxiliary results on Theorem 3.1.

### D.1. RMT Intuition on the Self-consistent Equation in Theorem 3.1

Here, we present a heuristic derivation of the self-consistent equation in (4) of Theorem 3.1. Let us recall some notations from Theorem 3.1. Let $\mathbf{x}_s^\top = \mathbf{e}_{i_s}^\top / \sqrt{\pi_{i_s}} \mathbf{A}$ as in Appendix C. For the ease of further use, we denote

$$\mathbf{Q} = (\mathbf{A}^\top \mathbf{S}^\top \mathbf{S} \mathbf{A} + \mathbf{C})^{-1} = \left( \frac{1}{m} \sum_{s=1}^m \mathbf{x}_s \mathbf{x}_s^\top + \mathbf{C} \right)^{-1}.$$

and $\mathbf{Q}_{-s} = (\sum_{j \neq s} \frac{1}{m} \mathbf{x}_j \mathbf{x}_j^\top + \mathbf{C})^{-1}$, for which we get

$$\sum_{s=1}^m \frac{1}{m} \mathbb{E}[\mathbf{x}_s \mathbf{x}_s^\top] = \sum_{i=1}^n \mathbf{a}_i \mathbf{a}_i^\top = \mathbf{A}^\top \mathbf{A}.$$

Then, we follow the *deterministic equivalent* framework (see Couillet & Liao (2022, Chapter 2) for an introduction) and show that $\|\mathbb{E}[\mathbf{Q}] - \widetilde{\mathbf{H}}^{-1}\| \simeq 0$, for $\widetilde{\mathbf{H}} = \mathbf{A}^\top \mathbf{D} \mathbf{A} + \mathbf{C}$, where $\mathbf{D} \in \mathbb{R}^{n \times n}$ is the diagonal matrix in Theorem 3.1. Note that

$$\|\mathbb{E}[\mathbf{Q}] - \widetilde{\mathbf{H}}^{-1}\| = \|\mathbb{E}[\mathbf{Q}] \mathbf{A}^\top \mathbf{D} \mathbf{A} \widetilde{\mathbf{H}}^{-1} - \mathbb{E}[\mathbf{Q} \mathbf{A}^\top \mathbf{S}^\top \mathbf{S} \mathbf{A}] \widetilde{\mathbf{H}}^{-1}\| \simeq 0,$$

and using Sherman-Morrison formula in Lemma A.3, we further ascertain that

$$\mathbb{E}[\mathbf{Q} \mathbf{A}^\top \mathbf{S}^\top \mathbf{S} \mathbf{A} \widetilde{\mathbf{H}}^{-1}] = \sum_{s=1}^m \mathbb{E}\left[ \frac{\frac{1}{m} \mathbf{Q}_{-s} \mathbf{x}_s \mathbf{x}_s^\top \widetilde{\mathbf{H}}^{-1}}{1 + \mathbf{x}_s^\top \mathbf{Q}_{-s} \mathbf{x}_s / m} \right] = \sum_{i=1}^n \mathbb{E}\left[ \frac{\mathbf{Q}_{-s} \mathbf{a}_i \mathbf{a}_i^\top \widetilde{\mathbf{H}}^{-1}}{1 + \mathbf{a}_i^\top \mathbf{Q}_{-s} \mathbf{a}_i / m\pi_i} \right]$$
$$\simeq \sum_{i=1}^n \mathbb{E}\left[ \frac{\mathbf{Q}_{-s} \mathbf{a}_i \mathbf{a}_i^\top \widetilde{\mathbf{H}}^{-1}}{1 + \mathbf{a}_i^\top \widetilde{\mathbf{H}}^{-1} \mathbf{a}_i / m\pi_i} \right].$$

Using the rank-one perturbation lemma in Silverstein & Bai (1995, Lemma 2.6), we obtain

$$\mathbb{E}[\mathbf{Q} \mathbf{A}^\top \mathbf{S}^\top \mathbf{S} \mathbf{A} \widetilde{\mathbf{H}}^{-1}] \simeq \sum_{i=1}^n \mathbb{E}\left[ \frac{\mathbf{Q} \mathbf{a}_i \mathbf{a}_i^\top \widetilde{\mathbf{H}}^{-1}}{1 + \mathbf{a}_i^\top \widetilde{\mathbf{H}}^{-1} \mathbf{a}_i / m\pi_i} \right] = \mathbb{E}\left[ \mathbf{Q} \sum_{i=1}^n \frac{\mathbf{a}_i \mathbf{a}_i^\top}{1 + \mathbf{a}_i^\top \widetilde{\mathbf{H}}^{-1} \mathbf{a}_i / m\pi_i} \widetilde{\mathbf{H}}^{-1} \right]$$
$$= \mathbb{E}[\mathbf{Q}] \mathbf{A}^\top \mathbf{D} \mathbf{A} \widetilde{\mathbf{H}}^{-1},$$

so that $\mathbf{D} = \text{diag}\left\{ \frac{m\pi_i}{m\pi_i + \mathbf{a}_i^\top \widetilde{\mathbf{H}}^{-1} \mathbf{a}_i} \right\}_{i=1}^n$. This leads to the self-consistent equation in (4) of Theorem 3.1.

### D.2. Detailed Proof of Theorem 3.1

As outlined in Appendix C, the proof of Theorem 3.1 also comes in the following two steps:

1. construct a high probability event $\zeta$ as in (19); and

2. conditioned on that event $\zeta$, bound the spectral norm $\|\mathbf{I}_d - \mathbb{E}_\zeta[\mathbf{Q}]\widetilde{\mathbf{H}}\|$ using "leave-one-out" analysis.

Furthermore, for each $s \in \{1, \ldots, m\}$, there also exists an index $j = j(s) \in \{1, 2, 3\}$ such that, conditioned on $\zeta_j$, we have $\mathbf{Q} \preceq \mathbf{Q}_{-s} \preceq 6\mathbf{H}^{-1}$.

To complete the proof of Theorem 3.1, we first rewrite

$$\|\mathbf{I}_d - \widetilde{\mathbf{H}}^{\frac{1}{2}}\mathbb{E}_\zeta[\mathbf{Q}]\widetilde{\mathbf{H}}^{\frac{1}{2}}\| = \|\widetilde{\mathbf{H}}^{\frac{1}{2}}(\mathbb{E}_\zeta[\mathbf{Q}]\widetilde{\mathbf{H}} - \mathbf{I}_d)\widetilde{\mathbf{H}}^{-\frac{1}{2}}\|.$$

Taking $\gamma_s = 1 + \frac{1}{m}\mathbf{x}_s^\top \mathbf{Q}_{-s}\mathbf{x}_s$, $s = 1, \ldots, m$, and $\tilde{D}_s = \frac{1}{1 + \frac{1}{m}\mathbf{x}_s^\top \widetilde{\mathbf{H}}^{-1}\mathbf{x}_s}$, we then get

$$\begin{aligned}
\mathbb{E}_\zeta[\mathbf{Q}]\widetilde{\mathbf{H}} - \mathbf{I}_d &= (\mathbb{E}_\zeta[\mathbf{Q}] - \widetilde{\mathbf{H}}^{-1})\widetilde{\mathbf{H}} = \mathbb{E}_\zeta\left[\mathbf{Q}\left(\mathbf{A}^\top \mathbf{DA} - \mathbf{A}^\top \mathbf{S}^\top \mathbf{SA}\right)\widetilde{\mathbf{H}}^{-1}\right]\widetilde{\mathbf{H}} \\
&= \mathbb{E}_\zeta\left[\mathbf{Q}\left(\mathbf{A}^\top \mathbf{DA} - \mathbf{A}^\top \mathbf{S}^\top \mathbf{SA}\right)\right] \\
&= \underbrace{\mathbb{E}_\zeta[\mathbf{Q} - \mathbf{Q}_{-s}]\mathbf{A}^\top \mathbf{DA}}_{\mathbf{Z}_1} + \underbrace{\mathbb{E}_\zeta[\mathbf{Q}_{-s}(\mathbf{A}^\top \mathbf{DA} - \tilde{D}_s \mathbf{x}_s \mathbf{x}_s^\top)]}_{\mathbf{Z}_2} + \underbrace{\mathbb{E}_\zeta[\mathbf{Q}_{-s}(\tilde{D}_s - \frac{1}{\gamma_s})\mathbf{x}_s \mathbf{x}_s^\top]}_{\mathbf{Z}_3},
\end{aligned}$$

which yields

$$\begin{aligned}
\|\mathbf{I}_d - \widetilde{\mathbf{H}}^{\frac{1}{2}}\mathbb{E}_\zeta[\mathbf{Q}]\widetilde{\mathbf{H}}^{\frac{1}{2}}\| &= \|\widetilde{\mathbf{H}}^{\frac{1}{2}}(\mathbf{Z}_1 + \mathbf{Z}_2 + \mathbf{Z}_3)\widetilde{\mathbf{H}}^{-\frac{1}{2}}\| \\
&\leq \|\widetilde{\mathbf{H}}^{\frac{1}{2}}\mathbf{Z}_1\widetilde{\mathbf{H}}^{-\frac{1}{2}}\| + \|\widetilde{\mathbf{H}}^{\frac{1}{2}}\mathbf{Z}_2\widetilde{\mathbf{H}}^{-\frac{1}{2}}\| + \|\widetilde{\mathbf{H}}^{\frac{1}{2}}\mathbf{Z}_3\widetilde{\mathbf{H}}^{-\frac{1}{2}}\|. \quad (24)
\end{aligned}$$

Then, we bound the first term $\|\widetilde{\mathbf{H}}^{1/2}\mathbf{Z}_1\widetilde{\mathbf{H}}^{-1/2}\|$. Together with the fact that the event $\zeta'$ is independent of $\mathbf{x}_s$, and using the Sherman-Morrison formula in Lemma A.3 and (22), we get

$$\begin{aligned}
\mathbb{E}_\zeta[\mathbf{Q}_{-s} - \mathbf{Q}] &= \mathbb{E}_\zeta[\frac{1}{m\gamma_s}\mathbf{Q}_{-s}\mathbf{x}_s\mathbf{x}_s^\top \mathbf{Q}_{-s}] \preceq 2\mathbb{E}_{\zeta'}[\frac{1}{m\gamma_s}\mathbf{Q}_{-s}\mathbf{x}_s\mathbf{x}_s^\top \mathbf{Q}_{-s}] \\
&= \frac{2}{m}\mathbb{E}_{\zeta'}[\mathbf{Q}_{-s}\mathbf{x}_s\mathbf{x}_s^\top \mathbf{Q}_{-s}] \preceq \frac{2}{m}\mathbb{E}_{\zeta'}[\mathbf{Q}_{-s}\sum_{j=1}^n \mathbf{a}_j\mathbf{a}_j^\top \mathbf{Q}_{-s}] \\
&= \frac{2}{m}\mathbb{E}_{\zeta'}[\mathbf{Q}_{-s}\mathbf{A}^\top \mathbf{A}\mathbf{Q}_{-s}] \preceq \frac{2}{m}\mathbb{E}_{\zeta'}[\mathbf{Q}_{-s}\mathbf{H}\mathbf{Q}_{-s}],
\end{aligned}$$

which, by incorporating the fact that $\widetilde{\mathbf{H}} \preceq \mathbf{H}$ and the event $\zeta$ indicates $\mathbf{H}^{1/2}\mathbf{Q}_{-s}\mathbf{H}^{1/2} \preceq 6\mathbf{I}_d$, leads to

$$\begin{aligned}
\|\widetilde{\mathbf{H}}^{\frac{1}{2}}\mathbf{Z}_1\widetilde{\mathbf{H}}^{-\frac{1}{2}}\| &\leq \|\widetilde{\mathbf{H}}^{\frac{1}{2}}\mathbb{E}_\zeta[\mathbf{Q} - \mathbf{Q}_{-s}]\widetilde{\mathbf{H}}^{\frac{1}{2}}\widetilde{\mathbf{H}}^{-\frac{1}{2}}\mathbf{A}^\top \mathbf{DA}\widetilde{\mathbf{H}}^{-\frac{1}{2}}\| \\
&\leq \|\widetilde{\mathbf{H}}^{\frac{1}{2}}\mathbf{H}^{-\frac{1}{2}}\|\|\mathbf{H}^{\frac{1}{2}}\mathbb{E}_\zeta[\mathbf{Q} - \mathbf{Q}_{-s}]\mathbf{H}^{\frac{1}{2}}\|\|\mathbf{H}^{-\frac{1}{2}}\widetilde{\mathbf{H}}^{\frac{1}{2}}\|\|\widetilde{\mathbf{H}}^{-\frac{1}{2}}\mathbf{A}^\top \mathbf{DA}\widetilde{\mathbf{H}}^{-\frac{1}{2}}\| \\
&\leq \|\widetilde{\mathbf{H}}^{\frac{1}{2}}\mathbf{H}^{-\frac{1}{2}}\|^2\|\mathbf{H}^{\frac{1}{2}}\mathbb{E}_\zeta[\mathbf{Q} - \mathbf{Q}_{-s}]\mathbf{H}^{\frac{1}{2}}\| \leq \frac{2}{m}\|\mathbb{E}_{\zeta'}[\mathbf{H}^{\frac{1}{2}}\mathbf{Q}_{-s}\mathbf{H}^{\frac{1}{2}}\mathbf{H}^{\frac{1}{2}}\mathbf{Q}_{-s}\mathbf{H}^{\frac{1}{2}}]\| \\
&\leq \frac{72}{m}. \quad (25)
\end{aligned}$$

Further, we move on to bound the second term in (24). Recalling the assumption regarding $\zeta_1, \zeta_2, \zeta'$ and $\delta_3$ from Appendix C, we have

$$\begin{aligned}
\mathbb{E}_\zeta[\mathbf{Q}_{-s}(\mathbf{A}^\top \mathbf{DA} - \tilde{D}_s\mathbf{x}_s\mathbf{x}_s^\top)] &= \frac{1}{1 - \delta_3}(\mathbb{E}_{\zeta'}[\mathbf{Q}_{-s}(\mathbf{A}^\top \mathbf{DA} - \tilde{D}_s\mathbf{x}_s\mathbf{x}_s^\top)] - \mathbb{E}_{\zeta'}[\mathbf{Q}_{-s}(\mathbf{A}^\top \mathbf{DA} - \tilde{D}_s\mathbf{x}_s\mathbf{x}_s^\top) \cdot \mathbf{1}_{\neg\zeta_3}]) \\
&= -\frac{1}{1 - \delta_3}\mathbb{E}_{\zeta'}[\mathbf{Q}_{-s}(\mathbf{A}^\top \mathbf{DA} - \tilde{D}_s\mathbf{x}_s\mathbf{x}_s^\top) \cdot \mathbf{1}_{\neg\zeta_3}].
\end{aligned}$$

Then, it follows that

$$
\begin{aligned}
\|\widetilde{\mathbf{H}}^{\frac{1}{2}}\mathbf{Z}_2\widetilde{\mathbf{H}}^{-\frac{1}{2}}\| &\leq 2\|\widetilde{\mathbf{H}}^{\frac{1}{2}}\mathbb{E}_{\zeta'}[\mathbf{Q}_{-s}(\mathbf{A}^\top\mathbf{D}\mathbf{A} - \tilde{D}_s\mathbf{x}_s\mathbf{x}_s^\top)\cdot\mathbf{1}_{\neg\zeta_3}]\widetilde{\mathbf{H}}^{-\frac{1}{2}}\| \\
&\leq 2\|\mathbb{E}_{\zeta'}[\widetilde{\mathbf{H}}^{\frac{1}{2}}\mathbf{Q}_{-s}\widetilde{\mathbf{H}}^{\frac{1}{2}}\widetilde{\mathbf{H}}^{-\frac{1}{2}}(\mathbf{A}^\top\mathbf{D}\mathbf{A} - \tilde{D}_s\mathbf{x}_s\mathbf{x}_s^\top)\cdot\mathbf{1}_{\neg\zeta_3}]\widetilde{\mathbf{H}}^{-\frac{1}{2}}\| \\
&\leq 12\|\widetilde{\mathbf{H}}^{\frac{1}{2}}\mathbf{H}^{-1}\widetilde{\mathbf{H}}^{\frac{1}{2}}\|\mathbb{E}_{\zeta'}[\|\widetilde{\mathbf{H}}^{-\frac{1}{2}}(\mathbf{A}^\top\mathbf{D}\mathbf{A} - \tilde{D}_s\mathbf{x}_s\mathbf{x}_s^\top)\cdot\mathbf{1}_{\neg\zeta_3}\widetilde{\mathbf{H}}^{-\frac{1}{2}}\|] \\
&\leq 12\mathbb{E}_{\zeta'}[\|\widetilde{\mathbf{H}}^{-\frac{1}{2}}(\mathbf{A}^\top\mathbf{D}\mathbf{A} - \tilde{D}_s\mathbf{x}_s\mathbf{x}_s^\top)\cdot\mathbf{1}_{\neg\zeta_3}\widetilde{\mathbf{H}}^{-\frac{1}{2}}\|] \\
&\leq 12\mathbb{E}_{\zeta'}[(1 + \|\widetilde{\mathbf{H}}^{-\frac{1}{2}}\tilde{D}_s\mathbf{x}_s\mathbf{x}_s^\top\widetilde{\mathbf{H}}^{-\frac{1}{2}}\|)\cdot\mathbf{1}_{\neg\zeta_3}] \\
&\leq 12\mathbb{E}_{\zeta'}[(1 + \tilde{D}_s\mathbf{x}_s^\top\widetilde{\mathbf{H}}^{-1}\mathbf{x}_s)\cdot\mathbf{1}_{\neg\zeta_3}] = 12\delta_3 + 12\mathbb{E}_{\zeta'}[\tilde{D}_s\mathbf{x}_s^\top\widetilde{\mathbf{H}}^{-1}\mathbf{x}_s\cdot\mathbf{1}_{\neg\zeta_3}].
\end{aligned}
$$

It follows from Lemma D.3 and $m \geq 2\rho_{\max}d_{\text{eff}}$ that

$$
\|\mathbf{H}^{\frac{1}{2}}\widetilde{\mathbf{H}}^{-1}\mathbf{H}^{\frac{1}{2}}\| \leq \frac{m + 2\rho_{\max}d_{\text{eff}}}{m}\|\mathbf{H}^{\frac{1}{2}}\mathbf{H}^{-1}\mathbf{H}^{\frac{1}{2}}\| = \frac{m + 2\rho_{\max}d_{\text{eff}}}{m} \leq 2, \tag{26}
$$

so that

$$
\begin{aligned}
\mathrm{Var}_{\zeta'}[\tilde{D}_s\mathbf{x}_s^\top\widetilde{\mathbf{H}}^{-1}\mathbf{x}_s] &\leq \mathbb{E}_{\zeta'}[(\tilde{D}_s\mathbf{x}_s^\top\widetilde{\mathbf{H}}^{-1}\mathbf{x}_s)^2] = \sum_{j=1}^n \pi_j\left(\frac{m\mathbf{a}_j^\top\widetilde{\mathbf{H}}^{-1}\mathbf{a}_j}{m\pi_j + \mathbf{a}_j^\top\widetilde{\mathbf{H}}^{-1}\mathbf{a}_j}\right)^2 \\
&= \sum_{j=1}^n \frac{D_{jj}^2(\mathbf{a}_j^\top\widetilde{\mathbf{H}}^{-1}\mathbf{a}_j)^2}{\pi_j} \leq \sum_{j=1}^n \frac{\|\mathbf{H}^{\frac{1}{2}}\widetilde{\mathbf{H}}^{-1}\mathbf{H}^{\frac{1}{2}}\|^2(\mathbf{a}_j^\top\mathbf{H}^{-1}\mathbf{a}_j)^2}{\pi_j} \\
&\leq 4\rho_{\max}d_{\text{eff}}\sum_{j=1}^n \mathbf{a}_j^\top\mathbf{H}^{-1}\mathbf{a}_j \leq 4\rho_{\max}d_{\text{eff}}^2,
\end{aligned}
$$

and

$$
\mathbb{E}_{\zeta'}[\tilde{D}_s\mathbf{x}_s^\top\widetilde{\mathbf{H}}^{-1}\mathbf{x}_s] = \sum_{j=1}^n \frac{m\pi_j}{m\pi_j + \mathbf{a}_j^\top\widetilde{\mathbf{H}}^{-1}\mathbf{a}_j}\mathbf{a}_j^\top\widetilde{\mathbf{H}}^{-1}\mathbf{a}_j \leq \sum_{j=1}^n \|\mathbf{H}^{\frac{1}{2}}\widetilde{\mathbf{H}}^{-1}\mathbf{H}^{\frac{1}{2}}\|\mathbf{a}_j^\top\mathbf{H}^{-1}\mathbf{a}_j \leq 2d_{\text{eff}}.
$$

Using Chebyshev's inequality, we get, for $x > 2d_{\text{eff}}$,

$$
\Pr(\tilde{D}_s\mathbf{x}_s^\top\widetilde{\mathbf{H}}^{-1}\mathbf{x}_s \geq x \mid \zeta') \leq \frac{4\rho_{\max}d_{\text{eff}}^2}{x^2}.
$$

This, together with $\delta_3 \leq \delta \leq \frac{1}{m^3}$ and $m > \rho_{\max}d_{\text{eff}}$ further leads to

$$
\begin{aligned}
\|\widetilde{\mathbf{H}}^{\frac{1}{2}}\mathbf{Z}_2\widetilde{\mathbf{H}}^{-\frac{1}{2}}\| &\leq O\left(\frac{1}{m^3}\right) + 12\int_0^\infty \Pr(\tilde{D}_s\mathbf{x}_s^\top\widetilde{\mathbf{H}}^{-1}\mathbf{x}_s\cdot\mathbf{1}_{\neg\zeta_3} \geq x \mid \zeta')dx \\
&\leq O\left(\frac{1}{m^3}\right) + 24m^2\delta_3 + 12\int_{2m^2}^\infty \Pr(\tilde{D}_s\mathbf{x}_s^\top\widetilde{\mathbf{H}}^{-1}\mathbf{x}_s \geq x \mid \zeta')dx \\
&\leq O\left(\frac{1}{m^3}\right) + \frac{24}{m} + 48\rho_{\max}d_{\text{eff}}^2\int_{2m^2}^\infty \frac{1}{x^2}dx \leq O\left(\frac{1}{m}\right) + \frac{24\rho_{\max}d_{\text{eff}}^2}{m^2} \\
&= O\left(\sqrt{\frac{\rho_{\max}^3 d_{\text{eff}}^3}{m^3}}\right).
\end{aligned}
$$

Using (22) and (26), we bound the last term in (24) as follows:

$$
\begin{aligned}
\|\widetilde{\mathbf{H}}^{\frac{1}{2}}\mathbf{Z}_3\widetilde{\mathbf{H}}^{-\frac{1}{2}}\| &\leq \sup_{\|\mathbf{u}\|=1,\|\mathbf{v}\|=1} \mathbb{E}_\zeta\left[|\tilde{D}_s - \gamma_s^{-1}||\mathbf{u}^\top\widetilde{\mathbf{H}}^{\frac{1}{2}}\mathbf{Q}_{-s}\mathbf{x}_s\mathbf{x}_s^\top\widetilde{\mathbf{H}}^{-\frac{1}{2}}\mathbf{v}|\right] \\
&\leq \sup_{\|\mathbf{u}\|=1,\|\mathbf{v}\|=1} 2\mathbb{E}_{\zeta'}\left[|\tilde{D}_s - \gamma_s^{-1}||\mathbf{u}^\top\widetilde{\mathbf{H}}^{\frac{1}{2}}\mathbf{Q}_{-s}\mathbf{x}_s\mathbf{x}_s^\top\widetilde{\mathbf{H}}^{-\frac{1}{2}}\mathbf{v}|\right] \\
&= \sup_{\|\mathbf{u}\|=1,\|\mathbf{v}\|=1} 2\mathbb{E}_{\zeta'}\left[\left|\frac{\frac{1}{m}\mathbf{x}_s^\top\mathbf{Q}_{-s}\mathbf{x}_s - \frac{1}{m}\mathbf{x}_s^\top\widetilde{\mathbf{H}}^{-1}\mathbf{x}_s}{\tilde{D}_s^{-1}\gamma_s}\right|\left|\mathbf{u}^\top\widetilde{\mathbf{H}}^{\frac{1}{2}}\mathbf{Q}_{-s}\mathbf{x}_s\mathbf{x}_s^\top\widetilde{\mathbf{H}}^{-\frac{1}{2}}\mathbf{v}\right|\right] \\
&\leq \sup_{\|\mathbf{u}\|=1,\|\mathbf{v}\|=1} \mathbb{E}_{\zeta'}\left[\left|\frac{\mathbf{a}_{i_s}^\top\mathbf{Q}_{-s}\mathbf{a}_{i_s} - \mathbf{a}_{i_s}^\top\widetilde{\mathbf{H}}^{-1}\mathbf{a}_{i_s}}{m\pi_{i_s}}\right|(\mathbf{u}^\top\widetilde{\mathbf{H}}^{\frac{1}{2}}\mathbf{Q}_{-s}\mathbf{x}_s\mathbf{x}_s\mathbf{Q}_{-s}\widetilde{\mathbf{H}}^{\frac{1}{2}}\mathbf{u} + \mathbf{v}^\top\widetilde{\mathbf{H}}^{-\frac{1}{2}}\mathbf{x}_s\mathbf{x}_s^\top\widetilde{\mathbf{H}}^{-\frac{1}{2}}\mathbf{v})\right] \\
&= \sup_{\|\mathbf{u}\|=1} \mathbb{E}_{\zeta'}\left[\sum_{j=1}^n\left|\frac{\mathbf{a}_j^\top\mathbf{Q}_{-s}\mathbf{a}_j - \mathbf{a}_j^\top\widetilde{\mathbf{H}}^{-1}\mathbf{a}_j}{m\pi_j}\right|\mathbf{u}^\top\widetilde{\mathbf{H}}^{\frac{1}{2}}\mathbf{Q}_{-s}\mathbf{a}_j\mathbf{a}_j^\top\mathbf{Q}_{-s}\widetilde{\mathbf{H}}^{\frac{1}{2}}\mathbf{u}\right] \\
&\quad + \sup_{\|\mathbf{v}\|=1} \mathbb{E}_{\zeta'}\left[\sum_{j=1}^n\left|\frac{\mathbf{a}_j^\top\mathbf{Q}_{-s}\mathbf{a}_j - \mathbf{a}_j^\top\widetilde{\mathbf{H}}^{-1}\mathbf{a}_j}{m\pi_j}\right|\mathbf{v}^\top\widetilde{\mathbf{H}}^{-\frac{1}{2}}\mathbf{a}_j\mathbf{a}_j^\top\widetilde{\mathbf{H}}^{-\frac{1}{2}}\mathbf{v}\right] \\
&\leq \sup_{\|\mathbf{u}\|=1} \mathbb{E}_{\zeta'}\left[\max_{1\leq j\leq n}\left|\frac{\mathbf{a}_j^\top\mathbf{Q}_{-s}\mathbf{a}_j - \mathbf{a}_j^\top\widetilde{\mathbf{H}}^{-1}\mathbf{a}_j}{m\pi_j}\right|\sum_{j=1}^n\mathbf{u}^\top\widetilde{\mathbf{H}}^{\frac{1}{2}}\mathbf{Q}_{-s}\mathbf{a}_j\mathbf{a}_j^\top\mathbf{Q}_{-s}\widetilde{\mathbf{H}}^{\frac{1}{2}}\mathbf{u}\right] \\
&\quad + \sup_{\|\mathbf{v}\|=1} \mathbb{E}_{\zeta'}\left[\max_{1\leq j\leq n}\left|\frac{\mathbf{a}_j^\top\mathbf{Q}_{-s}\mathbf{a}_j - \mathbf{a}_j^\top\widetilde{\mathbf{H}}^{-1}\mathbf{a}_j}{m\pi_j}\right|\sum_{j=1}^n\mathbf{v}^\top\widetilde{\mathbf{H}}^{-\frac{1}{2}}\mathbf{a}_j\mathbf{a}_j^\top\widetilde{\mathbf{H}}^{-\frac{1}{2}}\mathbf{v}\right] \\
&\leq \sup_{\|\mathbf{u}\|=1} \mathbb{E}_{\zeta'}\left[\max_{1\leq j\leq n}\left|\frac{\mathbf{a}_j^\top\mathbf{Q}_{-s}\mathbf{a}_j - \mathbf{a}_j^\top\widetilde{\mathbf{H}}^{-1}\mathbf{a}_j}{m\pi_j}\right|\mathbf{u}^\top\widetilde{\mathbf{H}}^{\frac{1}{2}}\mathbf{Q}_{-s}\mathbf{A}^\top\mathbf{A}\mathbf{Q}_{-s}\widetilde{\mathbf{H}}^{\frac{1}{2}}\mathbf{u}\right] \\
&\quad + \sup_{\|\mathbf{v}\|=1} \mathbb{E}_{\zeta'}\left[\max_{1\leq j\leq n}\left|\frac{\mathbf{a}_j^\top\mathbf{Q}_{-s}\mathbf{a}_j - \mathbf{a}_j^\top\widetilde{\mathbf{H}}^{-1}\mathbf{a}_j}{m\pi_j}\right|\mathbf{v}^\top\widetilde{\mathbf{H}}^{-\frac{1}{2}}\mathbf{A}^\top\mathbf{A}\widetilde{\mathbf{H}}^{-\frac{1}{2}}\mathbf{v}\right] \\
&\leq \mathbb{E}_{\zeta'}\left[\max_{1\leq j\leq n}\left|\frac{\mathbf{a}_j^\top\mathbf{Q}_{-s}\mathbf{a}_j - \mathbf{a}_j^\top\widetilde{\mathbf{H}}^{-1}\mathbf{a}_j}{m\pi_j}\right|\|\widetilde{\mathbf{H}}^{\frac{1}{2}}\mathbf{H}^{-\frac{1}{2}}\|^2\|\mathbf{H}^{\frac{1}{2}}\mathbf{Q}_{-s}\mathbf{H}^{\frac{1}{2}}\|^2\|\mathbf{H}^{-\frac{1}{2}}\mathbf{A}^\top\mathbf{A}\mathbf{H}^{-\frac{1}{2}}\|\right] \\
&\quad + \mathbb{E}_{\zeta'}\left[\max_{1\leq j\leq n}\left|\frac{\mathbf{a}_j^\top\mathbf{Q}_{-s}\mathbf{a}_j - \mathbf{a}_j^\top\widetilde{\mathbf{H}}^{-1}\mathbf{a}_j}{m\pi_j}\right|\|\widetilde{\mathbf{H}}^{-\frac{1}{2}}\mathbf{H}^{\frac{1}{2}}\|^2\|\mathbf{H}^{-\frac{1}{2}}\mathbf{A}^\top\mathbf{A}\mathbf{H}^{-\frac{1}{2}}\|\right] \\
&\overset{(a)}{\leq} \max_{1\leq j\leq n} 38\mathbb{E}_{\zeta'}\left[\left|\frac{\mathbf{a}_j^\top\mathbf{Q}_{-s}\mathbf{a}_j - \mathbf{a}_j^\top\widetilde{\mathbf{H}}^{-1}\mathbf{a}_j}{m\pi_j}\right|\right] \overset{(b)}{\leq} \max_{1\leq j\leq n} 38\sqrt{\frac{\mathbb{E}_{\zeta'}\left[(\mathbf{a}_j^\top\mathbf{Q}_{-s}\mathbf{a}_j - \mathbf{a}_j^\top\widetilde{\mathbf{H}}^{-1}\mathbf{a}_j)^2\right]}{(m\pi_j)^2}} \\
&= \underbrace{\max_{1\leq j\leq n} 38\sqrt{\frac{\mathbb{E}_{\zeta'}\left[(\mathbf{a}_j^\top\mathbf{Q}_{-s}\mathbf{a}_j - \mathbb{E}_{\zeta'}[\mathbf{a}_j^\top\mathbf{Q}_{-s}\mathbf{a}_j])^2\right]}{(m\pi_j)^2}}}_{M_1} + \underbrace{\max_{1\leq j\leq n} 38\frac{|\mathbb{E}_{\zeta'}[\mathbf{a}_j^\top\mathbf{Q}_{-s}\mathbf{a}_j] - \mathbf{a}_j^\top\widetilde{\mathbf{H}}^{-1}\mathbf{a}_j|}{m\pi_j}}_{M_2}, \quad (27)
\end{aligned}
$$

where in $(a)$ we use the fact that $\mathbf{A}^\top\mathbf{A} \preceq \mathbf{H}$, $\widetilde{\mathbf{H}} \preceq \mathbf{H}$ and the event $\zeta$ indicates $\mathbf{H}^{1/2}\mathbf{Q}_{-s}\mathbf{H}^{1/2} \preceq 6\mathbf{I}_d$, along with (26), and in $(b)$ we use Cauchy-Schwarz inequality. Then, by Lemma D.1, we bound the term $M_1$ in (27) as

$$
M_1 \leq \max_{1\leq j\leq n} 2736L_2^{\frac{1}{2}}\sqrt{\frac{\rho_{\max}d_{\text{eff}}}{m}\frac{(\mathbf{a}_j^\top\mathbf{H}^{-1}\mathbf{a}_j)^2}{(m\pi_j)^2}} + \max_{1\leq j\leq n} 645L_2^{\frac{1}{2}}\sqrt{\frac{1}{m}\frac{(\mathbf{a}_j^\top\mathbf{H}^{-1}\mathbf{a}_j)^2}{(m\pi_j)^2}} = O\left(\sqrt{\frac{\rho_{\max}^3d_{\text{eff}}^3}{m^3}}\right).
$$

Now, it remains to bound the second term $M_2$ in (27). Using again (26), we get

$$
\begin{aligned}
|\mathbb{E}_{\zeta'}[\mathbf{a}_j^\top \mathbf{Q}_{-s}\mathbf{a}_j] - \mathbf{a}_j^\top \widetilde{\mathbf{H}}^{-1}\mathbf{a}_j| &= |\mathbf{a}_j^\top \widetilde{\mathbf{H}}^{-\frac{1}{2}}\widetilde{\mathbf{H}}^{\frac{1}{2}}\mathbb{E}_{\zeta'}[\mathbf{Q}_{-s} - \widetilde{\mathbf{H}}^{-1}]\widetilde{\mathbf{H}}^{\frac{1}{2}}\widetilde{\mathbf{H}}^{-\frac{1}{2}}\mathbf{a}_j| \\
&\leq \mathbf{a}_j^\top \widetilde{\mathbf{H}}^{-1}\mathbf{a}_j \|\widetilde{\mathbf{H}}^{\frac{1}{2}}(\mathbb{E}_{\zeta'}[\mathbf{Q}_{-s}] - \widetilde{\mathbf{H}}^{-1})\widetilde{\mathbf{H}}^{\frac{1}{2}}\| \leq 2\mathbf{a}_j^\top \mathbf{H}^{-1}\mathbf{a}_j \|\widetilde{\mathbf{H}}^{\frac{1}{2}}(\mathbb{E}_{\zeta'}[\mathbf{Q}_{-s}] - \widetilde{\mathbf{H}}^{-1})\widetilde{\mathbf{H}}^{\frac{1}{2}}\| \\
&\leq 2\mathbf{a}_j^\top \mathbf{H}^{-1}\mathbf{a}_j \|\widetilde{\mathbf{H}}^{\frac{1}{2}}((\mathbb{E}_{\zeta'} - \mathbb{E}_{\zeta})[\mathbf{Q}_{-s}] + \mathbb{E}_{\zeta}[\mathbf{Q}_{-s} - \mathbf{Q}] + \mathbb{E}_{\zeta}[\mathbf{Q}] - \widetilde{\mathbf{H}}^{-1})\widetilde{\mathbf{H}}^{\frac{1}{2}}\| \\
&\leq 2\mathbf{a}_j^\top \mathbf{H}^{-1}\mathbf{a}_j \left( \underbrace{\|\widetilde{\mathbf{H}}^{\frac{1}{2}}(\mathbb{E}_{\zeta'} - \mathbb{E}_{\zeta})[\mathbf{Q}_{-s}]\widetilde{\mathbf{H}}^{\frac{1}{2}}\|}_{G_1} + \underbrace{\|\widetilde{\mathbf{H}}^{\frac{1}{2}}\mathbb{E}_{\zeta}[\mathbf{Q}_{-s} - \mathbf{Q}]\widetilde{\mathbf{H}}^{\frac{1}{2}}\|}_{G_2} + \|\widetilde{\mathbf{H}}^{\frac{1}{2}}(\mathbb{E}_{\zeta}[\mathbf{Q}]\widetilde{\mathbf{H}} - \mathbf{I}_d)\widetilde{\mathbf{H}}^{-\frac{1}{2}}\| \right).
\end{aligned}
\tag{28}
$$

Noting $\delta_3 < \frac{1}{m^3}$ and

$$
\mathbb{E}_{\zeta}[\mathbf{Q}_{-s}] = \frac{1}{1 - \delta_3}(\mathbb{E}_{\zeta'}[\mathbf{Q}_{-s}] - \delta_3 \mathbb{E}_{\zeta'}[\mathbf{Q}_{-s}|\neg\delta_3]),
$$

we have, for the term $G_1$ defined in (28), that

$$
\begin{aligned}
G_1 &= \frac{\delta_3}{1 - \delta_3}\|\widetilde{\mathbf{H}}^{\frac{1}{2}}(\mathbb{E}_{\zeta'}[\mathbf{Q}_{-s}] - \mathbb{E}_{\zeta'}[\mathbf{Q}_{-s}|\neg\delta_3])\widetilde{\mathbf{H}}^{\frac{1}{2}}\| \\
&\leq 2\delta_3(\|\widetilde{\mathbf{H}}^{\frac{1}{2}}\mathbb{E}_{\zeta'}[\mathbf{Q}_{-s}]\widetilde{\mathbf{H}}^{\frac{1}{2}}\| + \|\widetilde{\mathbf{H}}^{\frac{1}{2}}\mathbb{E}_{\zeta'}[\mathbf{Q}_{-s}|\neg\delta_3]\widetilde{\mathbf{H}}^{\frac{1}{2}}\|) \leq 24\delta_3 \leq \frac{24}{m^3}.
\end{aligned}
$$

For the term $G_2$ in (28), it follows from (25) that $G_2 = O(\frac{1}{m})$. We thus have

$$
\begin{aligned}
M_2 &\leq 76 \max_{1 \leq j \leq n} \frac{\mathbf{a}_j^\top \mathbf{H}^{-1}\mathbf{a}_j}{m\pi_j} \left( O\left(\frac{1}{m^3}\right) + O\left(\frac{1}{m}\right) + \|\widetilde{\mathbf{H}}^{\frac{1}{2}}\mathbf{Z}_1\widetilde{\mathbf{H}}^{-\frac{1}{2}}\| + \|\widetilde{\mathbf{H}}^{\frac{1}{2}}\mathbf{Z}_2\widetilde{\mathbf{H}}^{-\frac{1}{2}}\| + \|\widetilde{\mathbf{H}}^{\frac{1}{2}}\mathbf{Z}_3\widetilde{\mathbf{H}}^{-\frac{1}{2}}\| \right) \\
&\leq \frac{76\rho_{\max}d_{\text{eff}}}{m}\left( O\left(\frac{1}{m^3}\right) + O\left(\frac{1}{m}\right) + \|\widetilde{\mathbf{H}}^{\frac{1}{2}}\mathbf{Z}_1\widetilde{\mathbf{H}}^{-\frac{1}{2}}\| + \|\widetilde{\mathbf{H}}^{\frac{1}{2}}\mathbf{Z}_2\widetilde{\mathbf{H}}^{-\frac{1}{2}}\| + \|\widetilde{\mathbf{H}}^{\frac{1}{2}}\mathbf{Z}_3\widetilde{\mathbf{H}}^{-\frac{1}{2}}\| \right) \\
&\leq \frac{76\rho_{\max}d_{\text{eff}}}{m}\left( O\left(\frac{1}{m^3}\right) + O\left(\frac{1}{m}\right) + O\left(\frac{1}{m}\right) + O\left(\sqrt{\frac{\rho_{\max}^3 d_{\text{eff}}^3}{m^3}}\right) + O\left(\sqrt{\frac{\rho_{\max}^3 d_{\text{eff}}^3}{m^3}}\right) + M_2 \right),
\end{aligned}
$$

which, for $m > 76\rho_{\max}d_{\text{eff}}$, yields that

$$
M_2 = O\left( \sqrt{\frac{\rho_{\max}^3 d_{\text{eff}}^3}{m^3}\frac{76^2\rho_{\max}^2 d_{\text{eff}}^2}{(m - 76\rho_{\max}d_{\text{eff}})^2}} \right) = O\left( \sqrt{\frac{\rho_{\max}^3 d_{\text{eff}}^3}{m^3}} \right).
$$

Putting everything together, we conclude that $\|\widetilde{\mathbf{H}}^{1/2}\mathbb{E}_{\zeta}[\mathbf{Q}]\widetilde{\mathbf{H}}^{1/2} - \mathbf{I}_d\| = O\left( \sqrt{\rho_{\max}^3 d_{\text{eff}}^3/m^3} \right)$. This concludes the proof of Theorem 3.1. $\square$

### D.3. Discussions on Theorem 3.1 and Auxiliary Results

**Lemma D.1.** *Under the settings and notations of Theorem 3.1 and let* $\text{Var}_{\zeta'}[\cdot]$ *denote the variance conditioned on the event* $\zeta'$ *with* $\mathbf{x}_s$ *independent of* $\zeta'$. *Then, we have*

$$
\text{Var}_{\zeta'}[\mathbf{a}_j^\top \mathbf{Q}_{-s}\mathbf{a}_j] \leq \frac{5184L_2\rho_{\max}d_{\text{eff}}(\mathbf{a}_j^\top \mathbf{H}^{-1}\mathbf{a}_j)^2}{m} + \frac{288L_2(\mathbf{a}_j^\top \mathbf{H}^{-1}\mathbf{a}_j)^2}{m}.
$$

*Proof of Lemma D.1.* Use $\mathbf{Q}_{-sl}$ to denote the matrix $(\mathbf{A}^\top \mathbf{S}_{-sl}^\top \mathbf{S}_{-sl}\mathbf{A} + \mathbf{C})^{-1}$ where $\mathbf{S}_{-sl}$ is the matrix $\mathbf{S}$ without the $s^{th}$ and $l^{th}$ rows; let that, for each pair $s$, $l$, one of $\zeta_1$, $\zeta_2$ is independent of both $\mathbf{x}_s$ and $\mathbf{x}_l$. Without loss of generality, we assume that it is $\zeta_1$. Also, let $\mathbb{E}_{\zeta',l}[\cdot]$ be the expectation conditioned on $\zeta'$ and the $\sigma$-field $\mathcal{F}_l$ generated by the basic events $\mathbf{x}_1, \ldots, \mathbf{x}_l$. To apply the Burkholder inequality in Lemma A.4, we rewrite $\mathbf{a}_j^\top \mathbf{Q}_{-s}\mathbf{a}_j - \mathbb{E}_{\zeta'}[\mathbf{a}_j^\top \mathbf{Q}_{-s}\mathbf{a}_j]$ as the martingale

difference sequence

$$\mathbf{a}_j^\top \mathbf{Q}_{-s}\mathbf{a}_j - \mathbb{E}_{\zeta'}[\mathbf{a}_j^\top \mathbf{Q}_{-s}\mathbf{a}_j] = \mathbb{E}_{\zeta',m}[\mathbf{a}_j^\top \mathbf{Q}_{-s}\mathbf{a}_j] - \mathbb{E}_{\zeta',0}[\mathbf{a}_j^\top \mathbf{Q}_{-s}\mathbf{a}_j]$$

$$= \sum_{l=1}^m (\mathbb{E}_{\zeta',l} - \mathbb{E}_{\zeta',l-1})[\mathbf{a}_j^\top \mathbf{Q}_{-s}\mathbf{a}_j - \mathbf{a}_j^\top \mathbf{Q}_{-sl}\mathbf{a}_j] + \sum_{l=1}^m (\mathbb{E}_{\zeta',l} - \mathbb{E}_{\zeta',l-1})[\mathbf{a}_j^\top \mathbf{Q}_{-sl}\mathbf{a}_j]$$

$$= -\sum_{l=1}^m (\phi_l + \varphi_l),$$

where $\phi_l = (\mathbb{E}_{\zeta',l} - \mathbb{E}_{\zeta',l-1})[\mathbf{a}_j^\top \mathbf{Q}_{-sl}\mathbf{a}_j - \mathbf{a}_j^\top \mathbf{Q}_{-s}\mathbf{a}_j]$, and $\varphi_l = -(\mathbb{E}_{\zeta',l} - \mathbb{E}_{\zeta',l-1})[\mathbf{a}_j^\top \mathbf{Q}_{-sl}\mathbf{a}_j]$. Analogous to (22), we have the following result, the proof of which is straightforward.

**Lemma D.2.** *Under the settings and notations of Theorem 3.1, we have, for a p.s.d. random matrix* $\mathbf{M}$ *(or a non-negative random variable) living in the probability space of* $\mathbf{S}$,

$$\mathbb{E}_{\zeta'}[\mathbf{M}] = \frac{\mathbb{E}[(\prod_{j=1}^2 \mathbf{1}_{\zeta_j})\mathbf{M}]}{\Pr(\zeta')} \preceq \frac{1}{1-\delta}\mathbb{E}[\mathbf{1}_{\zeta_1}\mathbf{M}] \preceq 2\mathbb{E}_{\zeta_1}[\mathbf{M}], \tag{29}$$

*where* $\mathbf{1}_{\zeta_j}$ *is the indicator of the event* $\zeta_j$.

Conditioned on $\zeta'$, using (29) and the fact that

$$\mathbf{a}_j^\top \mathbf{Q}_{-sl}\mathbf{a}_j - \mathbf{a}_j^\top \mathbf{Q}_{-s}\mathbf{a}_j = \frac{1}{m}\frac{\mathbf{a}_j^\top \mathbf{Q}_{-sl}\mathbf{x}_l\mathbf{x}_l^\top \mathbf{Q}_{-sl}\mathbf{a}_j}{1 + \frac{1}{m}\mathbf{x}_l^\top \mathbf{Q}_{-sl}\mathbf{x}_l},$$

we obtain the following bound on the second moment of $\phi_l$ as

$$\mathbb{E}_{\zeta'}[\phi_l^2] \leq 2\mathbb{E}_{\zeta_1}[\phi_l^2] \leq 2\mathbb{E}_{\zeta_1}\left[\left(\frac{1}{m}\frac{\mathbf{a}_j^\top \mathbf{Q}_{-sl}\mathbf{x}_l\mathbf{x}_l^\top \mathbf{Q}_{-sl}\mathbf{a}_j}{1 + \frac{1}{m}\mathbf{x}_l^\top \mathbf{Q}_{-sl}\mathbf{x}_l}\right)^2\right] \leq 2\mathbb{E}_{\zeta_1}\left[\sum_{i=1}^n \pi_i\left(\frac{1}{m\pi_i}\frac{\mathbf{a}_j^\top \mathbf{Q}_{-sl}\mathbf{a}_i\mathbf{a}_i^\top \mathbf{Q}_{-sl}\mathbf{a}_j}{1 + \frac{1}{m\pi_i}\mathbf{a}_i^\top \mathbf{Q}_{-sl}\mathbf{a}_i}\right)^2\right]$$

$$\leq 2\mathbb{E}_{\zeta_1}\left[\sum_{i=1}^n \frac{1}{m^2\pi_i}\left(\mathbf{a}_j^\top \mathbf{Q}_{-sl}\mathbf{a}_i\mathbf{a}_i^\top \mathbf{Q}_{-sl}\mathbf{a}_j\right)^2\right]$$

$$\leq \frac{2}{m^2}\mathbb{E}_{\zeta_1}\left[\sum_{i=1}^n \mathbf{a}_j^\top \mathbf{Q}_{-sl}\mathbf{a}_i\mathbf{a}_i^\top \mathbf{Q}_{-sl}\mathbf{a}_j\frac{\mathbf{a}_j^\top \mathbf{Q}_{-sl}\mathbf{H}\mathbf{Q}_{-sl}\mathbf{a}_j\mathbf{a}_i^\top \mathbf{H}^{-1}\mathbf{a}_i}{\pi_i}\right]$$

$$\leq \frac{2\rho_{\max}d_{\mathrm{eff}}}{m^2}\mathbb{E}_{\zeta_1}\left[\sum_{i=1}^n \mathbf{a}_j^\top \mathbf{Q}_{-sl}\mathbf{a}_i\mathbf{a}_i^\top \mathbf{Q}_{-sl}\mathbf{a}_j\mathbf{a}_j^\top \mathbf{Q}_{-sl}\mathbf{H}\mathbf{Q}_{-sl}\mathbf{a}_j\right]$$

$$\leq \frac{2\rho_{\max}d_{\mathrm{eff}}}{m^2}\mathbb{E}_{\zeta_1}[\mathbf{a}_j^\top \mathbf{Q}_{-sl}\mathbf{A}^\top \mathbf{A}\mathbf{Q}_{-sl}\mathbf{a}_j\mathbf{a}_j^\top \mathbf{Q}_{-sl}\mathbf{H}\mathbf{Q}_{-sl}\mathbf{a}_j]$$

$$\leq \frac{2\rho_{\max}d_{\mathrm{eff}}}{m^2}\mathbb{E}_{\zeta_1}[(\mathbf{a}_j^\top \mathbf{Q}_{-sl}\mathbf{H}\mathbf{Q}_{-sl}\mathbf{a}_j)^2].$$

This, combined with the fact that $\mathbf{H}^{1/2}\mathbf{Q}_{-sl}\mathbf{H}^{1/2} \preceq 6\mathbf{I}_d$, results in

$$\mathbb{E}_{\zeta'}[\phi_l^2] \leq \frac{2592\rho_{\max}d_{\mathrm{eff}}}{m^2}(\mathbf{a}_j^\top \mathbf{H}^{-1}\mathbf{a}_j)^2.$$

Subsequently, we proceed to bound $|\varphi_l|$. Considering that $\zeta_1$ is independent of $\mathbf{x}_l$, it follows that $\mathbb{E}_{\zeta_1,l}[\mathbf{a}_j^\top \mathbf{Q}_{-sl}\mathbf{a}_j] = \mathbb{E}_{\zeta_1,l-1}[\mathbf{a}_j^\top \mathbf{Q}_{-sl}\mathbf{a}_j]$. Then, noting that, for $\delta_2 = \Pr(\neg\zeta_2) < \frac{1}{m}$,

$$\mathbb{E}_{\zeta',l}[\mathbf{a}_j^\top \mathbf{Q}_{-sl}\mathbf{a}_j] = \frac{1}{1-\delta_2}\mathbb{E}_{\zeta_1,l}[\mathbf{a}_j^\top \mathbf{Q}_{-sl}\mathbf{a}_j] - \frac{\delta_2}{1-\delta_2}\mathbb{E}_{\zeta_1,l}[\mathbf{a}_j^\top \mathbf{Q}_{-sl}\mathbf{a}_j|\neg\zeta_2],$$

and

$$\mathbb{E}_{\zeta',l-1}[\mathbf{a}_j^\top \mathbf{Q}_{-sl}\mathbf{a}_j] = \frac{1}{1-\delta_2}\mathbb{E}_{\zeta_1,l-1}[\mathbf{a}_j^\top \mathbf{Q}_{-sl}\mathbf{a}_j] - \frac{\delta_2}{1-\delta_2}\mathbb{E}_{\zeta_1,l-1}[\mathbf{a}_j^\top \mathbf{Q}_{-sl}\mathbf{a}_j|\neg\zeta_2],$$

so that

$$|\varphi_l| = |(\mathbb{E}_{\zeta',l} - \mathbb{E}_{\zeta',l-1})[\mathbf{a}_j^\top \mathbf{Q}_{-sl}\mathbf{a}_j]| \le \frac{\delta_2}{1-\delta_2}|(\mathbb{E}_{\zeta_1,l} - \mathbb{E}_{\zeta_1,l-1})[\mathbf{a}_j^\top \mathbf{Q}_{-sl}\mathbf{a}_j|\neg\zeta_2]|$$

$$\le 2\delta_2\mathbb{E}_{\zeta_1}[\mathbf{a}_j^\top \mathbf{Q}_{-sl}\mathbf{a}_j|\neg\zeta_2] \le 12\delta_2\mathbf{a}_j^\top \mathbf{H}^{-1}\mathbf{a}_j < \frac{12\mathbf{a}_j^\top \mathbf{H}^{-1}\mathbf{a}_j}{m}.$$

Consequently, let $y_l = -(\phi_l + \varphi_l)$ such that $\sum_{l=1}^m y_l = \mathbf{a}_j^\top \mathbf{Q}_{-s}\mathbf{a}_j - \mathbb{E}_{\zeta'}[\mathbf{a}_j^\top \mathbf{Q}_{-s}\mathbf{a}_j]$, it follows from the Burkholder inequality in Lemma A.4 (for $T = 2$) that

$$\mathrm{Var}_{\zeta'}[\mathbf{a}_j^\top \mathbf{Q}_{-s}\mathbf{a}_j] = \mathbb{E}_{\zeta'}\left[\left|\sum_{l=1}^m y_l\right|^2\right] \le L_2\sum_{l=1}^m \mathbb{E}_{\zeta'}[(\phi_l + \varphi_l)^2] \le 2L_2\sum_{l=1}^m \mathbb{E}_{\zeta'}[\phi_l^2] + 2L_2\sum_{l=1}^m \mathbb{E}_{\zeta'}[\varphi_l^2]$$

$$\le \frac{5184L_2\rho_{\max}d_{\mathrm{eff}}(\mathbf{a}_j^\top \mathbf{H}^{-1}\mathbf{a}_j)^2}{m} + \frac{288L_2(\mathbf{a}_j^\top \mathbf{H}^{-1}\mathbf{a}_j)^2}{m}.$$

This concludes the proof of Lemma D.1. □

Below is the proof of the result in Footnote 2.

**Lemma D.3** (On the self-consistent **D**). *For a given matrix $\mathbf{A} \in \mathbb{R}^{n\times d}$, let $\mathbf{S}$ be a random sampling matrix with number of trials $m$ and importance sampling distribution $\{\pi_i\}_{i=1}^n$ as in Definition 2.1, and let $\mathbf{C} \in \mathbb{R}^{d\times d}$ be a p.s.d. matrix and $d_{\mathrm{eff}} = \sum_{i=1}^n \ell_i^{\mathbf{C}}$ with leverage score $\ell_i^{\mathbf{C}}$ as in Definition 2.2. We have*

$$\frac{m}{m + 2\rho_{\max}d_{\mathrm{eff}}}\mathbf{I}_n \preceq \mathbf{D} \preceq \frac{m}{m + \rho_{\min}d_{\mathrm{eff}}}\mathbf{I}_n,$$

*in the sense of p.s.d. matrices, where we recall $\rho_{\max} = \max_{1\le i\le n} \ell_i^{\mathbf{C}}/(\pi_i d_{\mathrm{eff}})$ and $\rho_{\min} = \min_{1\le i\le n} \ell_i^{\mathbf{C}}/(\pi_i d_{\mathrm{eff}})$.*

*Proof of Lemma D.3.* For each $D_{ii}$, $i = 1, \ldots, n$, it follows from its definition in Theorem 3.1 that

$$D_{ii} = \frac{m\pi_i}{m\pi_i + \mathbf{a}_i^\top (\mathbf{A}^\top \mathbf{DA} + \mathbf{C})^{-1}\mathbf{a}_i} \ge \frac{m\pi_i}{m\pi_i + D_{\min}^{-1}\mathbf{a}_i^\top (\mathbf{A}^\top \mathbf{A} + \mathbf{C})^{-1}\mathbf{a}_i},$$

where $D_{\min} = \min_{1\le i\le n} D_{ii} \le 1$. Without loss of generality, let $D_{\min} = D_{nn}$ and use the fact that $m \ge C\rho_{\max}d_{\mathrm{eff}} \ge C\ell_n^{\mathbf{C}}/\pi_n = C\mathbf{a}_n^\top (\mathbf{A}^\top \mathbf{A} + \mathbf{C})^{-1}\mathbf{a}_n/\pi_n$, we obtain a lower bound on $D_{nn}$ as

$$D_{nn} = \frac{m\pi_n}{m\pi_n + \mathbf{a}_n^\top (\mathbf{A}^\top \mathbf{DA} + \mathbf{C})^{-1}\mathbf{a}_n} \ge \frac{m\pi_n}{m\pi_n + D_{nn}^{-1}\mathbf{a}_n^\top (\mathbf{A}^\top \mathbf{A} + \mathbf{C})^{-1}\mathbf{a}_n}$$

$$\ge \frac{m\pi_n}{m\pi_n + D_{nn}^{-1}C^{-1}m\pi_n} = \frac{C}{C + D_{nn}^{-1}},$$

so that $D_{nn} \ge \frac{C-1}{C} \equiv \Delta > 1/2$, for some $C > 2$. Then, it follows that, for $i = 1, \ldots, n$,

$$D_{ii} \ge \frac{m\pi_i}{m\pi_i + \Delta^{-1}\mathbf{a}_i^\top (\mathbf{A}^\top \mathbf{A} + \mathbf{C})^{-1}\mathbf{a}_i} \ge \frac{m}{m + \Delta^{-1}\mathbf{a}_i^\top (\mathbf{A}^\top \mathbf{A} + \mathbf{C})^{-1}\mathbf{a}_i/\pi_i} \ge \frac{m}{m + \Delta^{-1}\rho_{\max}d_{\mathrm{eff}}}.$$

This together with $C > 2$ results in

$$D_{ii} \ge \frac{m}{m + 2\rho_{\max}d_{\mathrm{eff}}}.$$

On the other hand, we have

$$D_{ii} = \frac{m}{m + \mathbf{a}_i^\top (\mathbf{A}^\top \mathbf{DA} + \mathbf{C})^{-1}\mathbf{a}_i/\pi_i} \le \frac{m}{m + \mathbf{a}_i^\top (\mathbf{A}^\top \mathbf{A} + \mathbf{C})^{-1}\mathbf{a}_i/\pi_i} \le \frac{m}{m + \rho_{\min}d_{\mathrm{eff}}}.$$

Consequently, we have that $\frac{m}{m+2\rho_{\max}d_{\mathrm{eff}}}\mathbf{I}_n \preceq \mathbf{D} \preceq \frac{m}{m+\rho_{\min}d_{\mathrm{eff}}}\mathbf{I}_n$, this concludes the proof of Lemma D.3. □

# E. Proof of Proposition 3.2

In this section, we start by presenting the RMT intuition for Proposition 3.2, and in particular how the *exact* leverage scores $\ell_i^{\mathbf{C}}$ comes into play in the analysis and debiasing, in Appendix E.1. The detailed proof of Proposition 3.2 is then given in Appendix E.2; and finally we provide discussions and auxiliary results on Proposition 3.2 in Appendix E.3.

## E.1. RMT Intuition on Proposition 3.2

Here, we present the heuristic derivation of Proposition 3.2. First, let us recall some notations from Proposition 3.2: let $\mathbf{x}_s^\top = \mathbf{e}_{i_s}^\top/\sqrt{\pi_{i_s}}\mathbf{A}$ as in Appendix C, for the ease of further use, let

$$\check{\mathbf{S}}^\top \check{\mathbf{S}} = \sum_{s=1}^{m} F_{i_s i_s} \cdot \frac{\mathbf{e}_{i_s}\mathbf{e}_{i_s}^\top}{m\pi_{i_s}},$$

for some *deterministic* $F_{ii}$ to be specified, and

$$\check{\mathbf{Q}} = (\mathbf{A}^\top \check{\mathbf{S}}^\top \check{\mathbf{S}}\mathbf{A} + \mathbf{C})^{-1} = \left(\frac{1}{m}\sum_{s=1}^{m} F_{i_s i_s}\mathbf{x}_s\mathbf{x}_s^\top + \mathbf{C}\right)^{-1},$$

and similarly $\check{\mathbf{Q}}_{-s} = (\frac{1}{m}\sum_{l\neq s} F_{i_l i_l}\mathbf{x}_l\mathbf{x}_l^\top + \mathbf{C})^{-1}$, for which we have

$$\frac{1}{m}\sum_{s=1}^{m} \mathbb{E}[F_{i_s i_s}\mathbf{x}_s\mathbf{x}_s^\top] = \sum_{i=1}^{n} F_{ii}\mathbf{a}_i\mathbf{a}_i^\top.$$

Our objective here is to find $\check{\mathbf{Q}}$ (and $F_{ii}$) such that $\|\mathbb{E}[\check{\mathbf{Q}}] - \mathbf{H}^{-1}\| \simeq 0$, for $\mathbf{H}^{-1} = (\mathbf{A}^\top\mathbf{A} + \mathbf{C})^{-1}$. This is

$$\mathbb{E}[\check{\mathbf{Q}}] - \mathbf{H}^{-1} = \mathbb{E}[\check{\mathbf{Q}}]\mathbf{A}^\top\mathbf{A}\mathbf{H}^{-1} - \mathbb{E}[\check{\mathbf{Q}}\mathbf{A}^\top\check{\mathbf{S}}^\top\check{\mathbf{S}}\mathbf{A}]\mathbf{H}^{-1} \simeq 0.$$

Using the Sherman-Morrison formula in Lemma A.3, we further ascertain that

$$\mathbb{E}[\check{\mathbf{Q}}\mathbf{A}^\top\check{\mathbf{S}}^\top\check{\mathbf{S}}\mathbf{A}\mathbf{H}^{-1}] = \mathbb{E}\left[\frac{\check{\mathbf{Q}}_{-s}F_{i_s i_s}\mathbf{A}^\top\mathbf{e}_{i_s}\mathbf{e}_{i_s}^\top/\pi_{i_s}\mathbf{A}\mathbf{H}^{-1}}{1 + F_{i_s i_s}\mathbf{e}_{i_s}^\top\mathbf{A}\check{\mathbf{Q}}_{-s}\mathbf{A}^\top\mathbf{e}_{i_s}/m\pi_{i_s}}\right] = \sum_{i=1}^{n}\mathbb{E}\left[\frac{\check{\mathbf{Q}}_{-s}F_{ii}\mathbf{A}^\top\mathbf{e}_i\mathbf{e}_i^\top\mathbf{A}\mathbf{H}^{-1}}{1 + F_{ii}\mathbf{e}_i^\top\mathbf{A}\check{\mathbf{Q}}_{-s}\mathbf{A}^\top\mathbf{e}_i/m\pi_i}\right]$$

$$\simeq \sum_{i=1}^{n}\mathbb{E}\left[\frac{\check{\mathbf{Q}}_{-s}F_{ii}\mathbf{A}^\top\mathbf{e}_i\mathbf{e}_i^\top\mathbf{A}\mathbf{H}^{-1}}{1 + F_{ii}\mathbf{e}_i^\top\mathbf{A}\mathbf{H}^{-1}\mathbf{A}^\top\mathbf{e}_i/m\pi_i}\right],$$

where we observe that the *exact* leverage score $\mathbf{e}_i^\top\mathbf{A}\mathbf{H}^{-1}\mathbf{A}^\top\mathbf{e}_i = \mathbf{a}_i^\top(\mathbf{A}^\top\mathbf{A} + \mathbf{C})^{-1}\mathbf{a}_i = \ell_i^{\mathbf{C}}$ given $\mathbf{C}$ as in Definition 2.2 naturally appears in the denominator from this leave-one-out derivation.

This, together with the rank-one perturbation lemma in Silverstein & Bai (1995, Lemma 2.6), gives that

$$\mathbb{E}[\check{\mathbf{Q}}\mathbf{A}^\top\check{\mathbf{S}}^\top\check{\mathbf{S}}\mathbf{A}\mathbf{H}^{-1}] \simeq \mathbb{E}[\check{\mathbf{Q}}]\mathbf{A}^\top\sum_{i=1}^{n}\frac{F_{ii}\mathbf{e}_i\mathbf{e}_i^\top}{1 + F_{ii}\ell_i^{\mathbf{C}}/m\pi_i}\mathbf{A}\mathbf{H}^{-1} = \mathbb{E}[\check{\mathbf{Q}}]\mathbf{A}^\top\mathbf{A}\mathbf{H}^{-1},$$

where we take the *debiasing factor* $F_{ii} = m\pi_i/(m\pi_i - \ell_i^{\mathbf{C}})$ such that $F_{ii}/(1 + F_{ii}\ell_i^{\mathbf{C}}/m\pi_i) = 1$. This thus leads to

$$\check{\mathbf{S}} = \text{diag}\left\{\sqrt{m/(m - \ell_{i_s}^{\mathbf{C}}/\pi_{i_s})}\right\}_{s=1}^{m} \cdot \mathbf{S}, \quad i_s \in \{1,\dots,n\}.$$

## E.2. Detailed Proof of Proposition 3.2

The proof of Proposition 3.2 follows the same line of arguments as in that of Theorem 3.1. In the following, we presented the proof for completeness. We specifically highlight the difference in the proof of Proposition 3.2 from that of Theorem 3.1, where $\mathbf{Q}$ is replaced by $\check{\mathbf{Q}}$.

The proof of Proposition 3.2 also comes in the following two steps:

1. construct a high-probability event $\zeta$, as in (19); and

2. conditional on the event $\zeta$, derive a bound for the spectral norm $\|\mathbf{I}_d - \mathbb{E}_\zeta[\check{\mathbf{Q}}]\mathbf{H}\|$ using again the "leave-one-out" type analysis.

For the second term above, we have, for $m > 2\rho_{\max} d_{\mathrm{eff}}$ that

$$F_{i_s i_s} < \frac{m\pi_{i_s}}{m\pi_{i_s} - 2^{-1}m\pi_{i_s}} < 2, \ s = 1, \dots, m. \tag{30}$$

This yields that, for given $s \in \{1, \dots, m\}$, there exists an index $j = j(s) \in \{1, 2, 3\}$ such that $\check{\mathbf{Q}} \preceq \check{\mathbf{Q}}_{-s} \preceq 6\mathbf{H}^{-1}$ holds on the event $\zeta_j$.

Denote $\check{\gamma}_s = 1 + \frac{1}{m}F_{i_s i_s}\mathbf{x}_s^\top \check{\mathbf{Q}}_{-s}\mathbf{x}_s$, we then obtain

$$\mathbb{E}_\zeta[\check{\mathbf{Q}}]\mathbf{H} - \mathbf{I}_d = (\mathbb{E}_\zeta[\check{\mathbf{Q}}] - \mathbf{H}^{-1})\mathbf{H} = \mathbb{E}_\zeta\left[\check{\mathbf{Q}}\left(\mathbf{A}^\top\mathbf{A} - \sum_{s=1}^m \frac{1}{m}F_{i_s i_s}\mathbf{x}_s\mathbf{x}_s^\top\right)\mathbf{H}^{-1}\right]\mathbf{H}$$

$$= \mathbb{E}_\zeta\left[\check{\mathbf{Q}}\left(\mathbf{A}^\top\mathbf{A} - \sum_{s=1}^m \frac{1}{m}F_{i_s i_s}\mathbf{x}_s\mathbf{x}_s^\top\right)\right] = \mathbb{E}_\zeta[\check{\mathbf{Q}}\mathbf{A}^\top\mathbf{A}] - \mathbb{E}_\zeta\left[\frac{1}{\check{\gamma}_s}\check{\mathbf{Q}}_{-s}F_{i_s i_s}\mathbf{x}_s\mathbf{x}_s^\top\right]$$

$$= \underbrace{\mathbb{E}_\zeta[\check{\mathbf{Q}} - \check{\mathbf{Q}}_{-s}]\mathbf{A}^\top\mathbf{A}}_{\check{\mathbf{Z}}_1} + \underbrace{\mathbb{E}_\zeta[\check{\mathbf{Q}}_{-s}(\mathbf{A}^\top\mathbf{A} - \mathbf{x}_s\mathbf{x}_s^\top)]}_{\check{\mathbf{Z}}_2} + \underbrace{\mathbb{E}_\zeta\left[\left(1 - \frac{F_{i_s i_s}}{\check{\gamma}_s}\right)\check{\mathbf{Q}}_{-s}\mathbf{x}_s\mathbf{x}_s^\top\right]}_{\check{\mathbf{Z}}_3},$$

which leads to

$$\|\mathbf{I}_d - \mathbf{H}^{\frac{1}{2}}\mathbb{E}_\zeta[\check{\mathbf{Q}}]\mathbf{H}^{\frac{1}{2}}\| = \|\mathbf{H}^{\frac{1}{2}}(\mathbb{E}_\zeta[\check{\mathbf{Q}}]\mathbf{H} - \mathbf{I}_d)\mathbf{H}^{-\frac{1}{2}}\| \le \|\mathbf{H}^{\frac{1}{2}}\check{\mathbf{Z}}_1\mathbf{H}^{-\frac{1}{2}}\| + \|\mathbf{H}^{\frac{1}{2}}\check{\mathbf{Z}}_2\mathbf{H}^{-\frac{1}{2}}\| + \|\mathbf{H}^{\frac{1}{2}}\check{\mathbf{Z}}_3\mathbf{H}^{-\frac{1}{2}}\|. \tag{31}$$

We first bound the first term $\|\mathbf{H}^{1/2}\check{\mathbf{Z}}_1\mathbf{H}^{-1/2}\|$ in (31). Adapting the bound for $\mathbb{E}_\zeta[\mathbf{Q}_{-s} - \mathbf{Q}]$ and $\|\widetilde{\mathbf{H}}^{1/2}\mathbf{Z}_1\widetilde{\mathbf{H}}^{-1/2}\|$ in the proof of Theorem 3.1 in Appendix D.2, and using (30) along with the fact that $\mathbf{H}^{1/2}\check{\mathbf{Q}}_{-s}\mathbf{H}^{1/2} \preceq 6\mathbf{I}_d$ when conditioned on $\zeta'$, we get

$$\mathbb{E}_\zeta[\check{\mathbf{Q}}_{-s} - \check{\mathbf{Q}}] \preceq \mathbb{E}_\zeta\left[\frac{F_{i_s i_s}}{\check{\gamma}_s m}\check{\mathbf{Q}}_{-s}\mathbf{x}_s\mathbf{x}_s^\top\check{\mathbf{Q}}_{-s}\right] \preceq \frac{4}{m}\mathbb{E}_{\zeta'}[\check{\mathbf{Q}}_{-s}\mathbf{H}\check{\mathbf{Q}}_{-s}], \tag{32}$$

and

$$\|\mathbf{H}^{\frac{1}{2}}\check{\mathbf{Z}}_1\mathbf{H}^{-\frac{1}{2}}\| = \|\mathbf{H}^{\frac{1}{2}}\mathbb{E}_\zeta[\check{\mathbf{Q}}_{-s} - \check{\mathbf{Q}}]\mathbf{H}^{\frac{1}{2}}\mathbf{H}^{-\frac{1}{2}}\mathbf{A}^\top\mathbf{A}\mathbf{H}^{-\frac{1}{2}}\| = O\left(\frac{1}{m}\right).$$

Next, we bound the second term $\|\mathbf{H}^{1/2}\check{\mathbf{Z}}_2\mathbf{H}^{-1/2}\|$ in (31). Note that $\mathbb{E}_{\zeta'}[\mathbf{x}_s^\top\mathbf{H}^{-1}\mathbf{x}_s] = d_{\mathrm{eff}}$ and

$$\mathrm{Var}_{\zeta'}[\mathbf{x}_s^\top\mathbf{H}^{-1}\mathbf{x}_s] \le \mathbb{E}_{\zeta'}[(\mathbf{x}_s^\top\mathbf{H}^{-1}\mathbf{x}_s)^2] = \sum_{j=1}^n \pi_j \frac{(\mathbf{a}_j^\top\mathbf{H}^{-1}\mathbf{a}_j)^2}{\pi_j^2} = \sum_{j=1}^n \frac{(\mathbf{a}_j^\top\mathbf{H}^{-1}\mathbf{a}_j)^2}{\pi_j} \le \rho_{\max}d_{\mathrm{eff}}^2.$$

Then, using Chebyshev's inequality, we have, for $x \ge 2d_{\mathrm{eff}}$ that

$$\Pr(\mathbf{x}_s^\top\mathbf{H}^{-1}\mathbf{x}_s \ge x \mid \zeta') \le \frac{\rho_{\max}d_{\mathrm{eff}}^2}{x^2}.$$

Analogous to the bound on $\mathbb{E}_\zeta[\mathbf{Q}_{-s}(\mathbf{A}^\top\mathbf{D}\mathbf{A} - \tilde{D}_s\mathbf{x}_s\mathbf{x}_s^\top)]$ and $\|\widetilde{\mathbf{H}}^{1/2}\mathbf{Z}_2\widetilde{\mathbf{H}}^{-1/2}\|$ in Appendix D.2, we get

$$\mathbb{E}_\zeta[\check{\mathbf{Q}}_{-s}(\mathbf{A}^\top\mathbf{A} - \mathbf{x}_s\mathbf{x}_s^\top)] = -\frac{1}{1-\delta_3}\mathbb{E}_{\zeta'}[\check{\mathbf{Q}}_{-s}(\mathbf{A}^\top\mathbf{A} - \mathbf{x}_s\mathbf{x}_s^\top) \cdot \mathbf{1}_{\neg\zeta_3}],$$

and

$$\|\mathbf{H}^{\frac{1}{2}}\check{\mathbf{Z}}_2\mathbf{H}^{-\frac{1}{2}}\| \leq 12\delta_3 + 12\mathbb{E}_{\zeta'}[\mathbf{x}_s^\top\mathbf{H}^{-1}\mathbf{x}_s \cdot \mathbf{1}\neg\zeta_3] = 12\delta_3 + 12\int_0^\infty \Pr(\mathbf{x}_s^\top\mathbf{H}^{-1}\mathbf{x}_s \cdot \mathbf{1}_{\neg\zeta_3} \geq x \mid \zeta')dx$$

$$= O\left(\sqrt{\frac{\rho_{\max}^3 d_{\mathrm{eff}}^3}{m^3}}\right).$$

Now, we move on to bound the last term $\|\mathbf{H}^{1/2}\check{\mathbf{Z}}_3\mathbf{H}^{-1/2}\|$ in (31). Considering (30), it follows that

$$\left|1 - \frac{F_{i_s i_s}}{\check{\gamma}_s}\right| = \left|\frac{\mathbf{a}_{i_s}^\top\check{\mathbf{Q}}_{-s}\mathbf{a}_{i_s} - \ell_{i_s}^{\mathbf{C}}}{m\pi_{i_s} - \ell_{i_s}^{\mathbf{C}} + \mathbf{a}_{i_s}^\top\check{\mathbf{Q}}_{-s}\mathbf{a}_{i_s}}\right| \leq \left|\frac{\mathbf{a}_{i_s}^\top\check{\mathbf{Q}}_{-s}\mathbf{a}_{i_s} - \ell_{i_s}^{\mathbf{C}}}{m\pi_{i_s} - \ell_{i_s}^{\mathbf{C}}}\right| = F_{i_s i_s}\left|\frac{\mathbf{a}_{i_s}^\top\check{\mathbf{Q}}_{-s}\mathbf{a}_{i_s} - \ell_{i_s}^{\mathbf{C}}}{m\pi_{i_s}}\right|$$

$$\leq 2\left|\frac{\mathbf{a}_{i_s}^\top\check{\mathbf{Q}}_{-s}\mathbf{a}_{i_s} - \ell_{i_s}^{\mathbf{C}}}{m\pi_{i_s}}\right| = 2\left|\frac{\mathbf{a}_{i_s}^\top\check{\mathbf{Q}}_{-s}\mathbf{a}_{i_s} - \mathbf{a}_{i_s}^\top\mathbf{H}^{-1}\mathbf{a}_{i_s}}{m\pi_{i_s}}\right|. \tag{33}$$

Following the bound on $\|\widetilde{\mathbf{H}}^{1/2}\mathbf{Z}_3\widetilde{\mathbf{H}}^{-1/2}\|$ in Appendix D.2, and recalling (30), we further have

$$\|\mathbf{H}^{\frac{1}{2}}\check{\mathbf{Z}}_3\mathbf{H}^{-\frac{1}{2}}\| \leq \sup_{\|\mathbf{u}\|=1,\|\mathbf{v}\|=1} \mathbb{E}_\zeta\left[\left|1 - \frac{F_{i_s i_s}}{\check{\gamma}_s}\right|\left|\mathbf{u}^\top\mathbf{H}^{\frac{1}{2}}\check{\mathbf{Q}}_{-s}\mathbf{x}_s\mathbf{x}_s^\top\mathbf{H}^{-\frac{1}{2}}\mathbf{v}\right|\right]$$

$$\leq \max_{1\leq j\leq n} 74\sqrt{\frac{\mathbb{E}_{\zeta'}\left[(\mathbf{a}_j^\top\check{\mathbf{Q}}_{-s}\mathbf{a}_j - \mathbf{a}_j^\top\mathbf{H}^{-1}\mathbf{a}_j)^2\right]}{(m\pi_j)^2}}$$

$$= \underbrace{\max_{1\leq j\leq n} 74\sqrt{\frac{\mathbb{E}_{\zeta'}\left[(\mathbf{a}_j^\top\check{\mathbf{Q}}_{-s}\mathbf{a}_j - \mathbb{E}_{\zeta'}[\mathbf{a}_j^\top\check{\mathbf{Q}}_{-s}\mathbf{a}_j])^2\right]}{(m\pi_j)^2}}}_{\check{M}_1} + \underbrace{\max_{1\leq j\leq n} 74\frac{\left|\mathbb{E}_{\zeta'}[\mathbf{a}_j^\top\check{\mathbf{Q}}_{-s}\mathbf{a}_j] - \mathbf{a}_j^\top\mathbf{H}^{-1}\mathbf{a}_j\right|}{m\pi_j}}_{\check{M}_2}, \tag{34}$$

Subsequently, like the bounds for $M_1$ and $M_2$ in Appendix D.2, and using (30), we obtain the following bounds for $\check{M}_1$ and $\check{M}_2$:

$$\check{M}_1 = O\left(\sqrt{\frac{\rho_{\max}^3 d_{\mathrm{eff}}^3}{m^3}}\right),$$

and

$$\check{M}_2 \leq \max_{1\leq j\leq n} \frac{74\mathbf{a}_j^\top\mathbf{H}^{-1}\mathbf{a}_j}{m\pi_j}(\|\mathbf{H}^{\frac{1}{2}}(\mathbb{E}_{\zeta'} - \mathbb{E}_\zeta)[\check{\mathbf{Q}}_{-s}]\mathbf{H}^{\frac{1}{2}}\| + \|\mathbf{H}^{\frac{1}{2}}\mathbb{E}_\zeta[\check{\mathbf{Q}}_{-s} - \check{\mathbf{Q}}]\mathbf{H}^{\frac{1}{2}}\| + \|\mathbf{H}^{\frac{1}{2}}(\mathbb{E}_\zeta[\check{\mathbf{Q}}]\mathbf{H} - \mathbf{I}_d)\mathbf{H}^{-\frac{1}{2}}\|)$$

$$= O\left(\sqrt{\frac{\rho_{\max}^3 d_{\mathrm{eff}}^3}{m^3}}\right).$$

Combining the above, we conclude that $\|\mathbf{I}_d - \mathbf{H}^{1/2}\mathbb{E}_\zeta[\check{\mathbf{Q}}]\mathbf{H}^{1/2}\| = O\left(\sqrt{\rho_{\max}^3 d_{\mathrm{eff}}^3/m^3}\right)$, and thus complete the proof of Proposition 3.2. □

### E.3. Discussions on Proposition 3.2 and Auxiliary Results

Here, we present some auxiliary results in addition to the fine-grained debiasing results in Proposition 3.2. Precisely, we show in Appendix E.3.1 that by substituting exact leverage scores with some (computationally efficient) approximations leads to a controlled inversion bias. We then provide the proof of Corollary 3.4 in Appendix E.3.2. A "counter-example" to Dereziński et al. (2021b, Theorem 10) that uses scalar debiasing to achieve zero inversion bias in the case of *exact* leverage score sampling is given in Appendix E.3.3. Finally, we establish the proof of Corollary 3.7 in Appendix E.3.4.

E.3.1. DISCUSSIONS ON DEBIASING $\check{\mathbf{S}}$ USING APPROXIMATE LEVERAGE SCORES

**Corollary E.1** (Fine-grained debiasing $\check{\mathbf{S}}$ using approximate leverage scores). Under the setting and notations of Theorem 3.1, if one uses approximate leverage scores $\check{\ell}_i^{\mathbf{C}}$ in the debiasing matrix, instead of the exact leverage scores $\ell_i^{\mathbf{C}}$ in (5) in Proposition 3.2, that is

$$\check{\mathbf{S}} = \mathrm{diag}\left\{ \sqrt{m/(m - \check{\ell}_{i_s}^{\mathbf{C}}/\pi_{i_s})} \right\}_{s=1}^m \cdot \mathbf{S}, \quad (1-\omega)\ell_i^{\mathbf{C}} \le \check{\ell}_i^{\mathbf{C}} \le (1+\omega)\ell_i^{\mathbf{C}}, \quad \omega \in [0,1].$$

Then, there exists $C > 0$ independent of $n, d_{\mathrm{eff}}$ such that for $m \ge C\rho_{\max}d_{\mathrm{eff}}(\log(d_{\mathrm{eff}}/\delta) + \max\{\omega/\epsilon, 1/\epsilon^{2/3}\})$ with $\delta \le m^{-3}$, $(\mathbf{A}^\top \check{\mathbf{S}}^\top \check{\mathbf{S}}\mathbf{A} + \mathbf{C})^{-1}$ is an $(\epsilon, \delta)$-unbiased estimator of $(\mathbf{A}^\top \mathbf{A} + \mathbf{C})^{-1}$.

*Proof of Corollary E.1.* The proof of Corollary E.1 largely follows the proof of Proposition 3.2, and is presented here for completeness. In the following, we emphasize the difference from the proof of Proposition 3.2, particularly regarding the matrix $\check{\mathbf{S}}$, which is constructed using the approximate leverage scores denoted by $\check{\ell}_i^{\mathbf{C}}$ as

$$\check{\mathbf{S}} = \mathrm{diag}\left\{ \sqrt{m/(m - \check{\ell}_{i_s}^{\mathbf{C}}/\pi_{i_s})} \right\}_{s=1}^m \cdot \mathbf{S}, \quad i_s \in \{1, \dots, n\}.$$

This implies, in the proof of Proposition 3.2 that

$$F_{i_s i_s} = \frac{m\pi_{i_s}}{m\pi_{i_s} - \check{\ell}_{i_s}^{\mathbf{C}}}. \tag{35}$$

Note that in this setting, we have, for $\omega \in [0,1]$ that $|\check{\ell}_i^{\mathbf{C}} - \ell_i^{\mathbf{C}}| \le \omega \ell_i^{\mathbf{C}}$ and $\check{\ell}_i^{\mathbf{C}} \le (1+\omega)\ell_i^{\mathbf{C}} \le 2\ell_i^{\mathbf{C}}$. As such, for $m \ge C\rho_{\max}d_{\mathrm{eff}} \ge C\ell_i^{\mathbf{C}}/\pi_{i_s}$ with $C > 4$, we have

$$F_{i_s i_s} \le \frac{m\pi_{i_s}}{m\pi_{i_s} - 2\ell_i^{\mathbf{C}}} < 2.$$

Thus, adapting the proof of Proposition 3.2, we get the same bounds on $\|\mathbf{H}^{1/2}\check{\mathbf{Z}}_1\mathbf{H}^{-1/2}\|$ and $\|\mathbf{H}^{1/2}\check{\mathbf{Z}}_2\mathbf{H}^{-1/2}\|$ as in Appendix E.2, that is

$$\|\mathbf{H}^{\frac{1}{2}}\check{\mathbf{Z}}_1\mathbf{H}^{-\frac{1}{2}}\| = O\left(\frac{1}{m}\right), \quad \|\mathbf{H}^{\frac{1}{2}}\check{\mathbf{Z}}_2\mathbf{H}^{-\frac{1}{2}}\| = O\left(\sqrt{\frac{\rho_{\max}^3 d_{\mathrm{eff}}^3}{m^3}}\right).$$

Next, we establish a bound for the term $\|\mathbf{H}^{1/2}\check{\mathbf{Z}}_3\mathbf{H}^{-1/2}\|$ in the setting of Corollary E.1. Using (35), we obtain

$$\left|1 - \frac{F_{i_s i_s}}{\check{\gamma}_s}\right| = \left|\frac{\mathbf{a}_{i_s}^\top \check{\mathbf{Q}}_{-s}\mathbf{a}_{i_s} - \check{\ell}_{i_s}^{\mathbf{C}}}{m\pi_{i_s} - \check{\ell}_{i_s}^{\mathbf{C}} + \mathbf{a}_{i_s}^\top \check{\mathbf{Q}}_{-s}\mathbf{a}_{i_s}}\right| \le \left|\frac{\mathbf{a}_{i_s}^\top \check{\mathbf{Q}}_{-s}\mathbf{a}_{i_s} - \check{\ell}_{i_s}^{\mathbf{C}}}{(1 - C^{-1})m\pi_{i_s}}\right| \le 2\left|\frac{\mathbf{a}_{i_s}^\top \check{\mathbf{Q}}_{-s}\mathbf{a}_{i_s} - \check{\ell}_{i_s}^{\mathbf{C}}}{m\pi_{i_s}}\right|$$

$$\le 2\left|\frac{\mathbf{a}_{i_s}^\top \check{\mathbf{Q}}_{-s}\mathbf{a}_{i_s} - \ell_{i_s}^{\mathbf{C}}}{m\pi_{i_s}}\right| + 2\left|\frac{\ell_{i_s}^{\mathbf{C}} - \check{\ell}_{i_s}^{\mathbf{C}}}{m\pi_{i_s}}\right| \le 2\left|\frac{\mathbf{a}_{i_s}^\top \check{\mathbf{Q}}_{-s}\mathbf{a}_{i_s} - \ell_{i_s}^{\mathbf{C}}}{m\pi_{i_s}}\right| + \frac{2\omega\rho_{\max}d_{\mathrm{eff}}}{m}.$$

This, together with (34) yields that

$$\|\mathbf{H}^{\frac{1}{2}}\check{\mathbf{Z}}_3\mathbf{H}^{-\frac{1}{2}}\| = O\left(\frac{\omega\rho_{\max}d_{\mathrm{eff}}}{m} + \sqrt{\frac{\rho_{\max}^3 d_{\mathrm{eff}}^3}{m^3}}\right).$$

Putting these bounds together, we conclude the proof of Corollary E.1. $\square$

E.3.2. PROOF OF COROLLARY 3.4

The proof Corollary 3.4 largely mirrors that of Proposition 3.2, and is included here for thoroughness. For $m > Cd_{\mathrm{eff}}$ with $C > 2$, we have

$$F_{i_s i_s} = \frac{m}{m - d_{\mathrm{eff}}} = \frac{m\pi_{i_s}}{m\pi_{i_s} - d_{\mathrm{eff}}\pi_{i_s}} \le \frac{m\pi_{i_s}}{(1 - C^{-1})m\pi_{i_s}} < 2.$$

Then, recalling $\pi_i \in [\ell_i^{\mathbf{C}}/(d_{\text{eff}}\rho_{\max}), \ell_i^{\mathbf{C}}/(d_{\text{eff}}\rho_{\min})]$ such that $|\pi_i - \ell_i^{\mathbf{C}}/d_{\text{eff}}| \leq \epsilon_\rho \ell_i^{\mathbf{C}}/d_{\text{eff}}$ with $\epsilon_\rho = \max\{(\rho_{\min}^{-1} - 1), (1 - \rho_{\max}^{-1})\}$, and (33) can be rewritten as

$$\left|1 - \frac{F_{i_s i_s}}{\check{\gamma}_s}\right| = \left|\frac{\mathbf{a}_{i_s}^\top \check{\mathbf{Q}}_{-s}\mathbf{a}_{i_s} - d_{\text{eff}}\pi_{i_s}}{m\pi_{i_s} - d_{\text{eff}}\pi_{i_s} + \mathbf{a}_{i_s}^\top\check{\mathbf{Q}}_{-s}\mathbf{a}_{i_s}}\right| \leq \left|\frac{\mathbf{a}_{i_s}^\top\check{\mathbf{Q}}_{-s}\mathbf{a}_{i_s} - d_{\text{eff}}\pi_{i_s}}{(1 - C^{-1})m\pi_{i_s}}\right| \leq 2\left|\frac{\mathbf{a}_{i_s}^\top\check{\mathbf{Q}}_{-s}\mathbf{a}_{i_s} - \ell_{i_s}^{\mathbf{C}}}{m\pi_{i_s}}\right| + 2\left|\frac{\ell_{i_s}^{\mathbf{C}} - d_{\text{eff}}\pi_{i_s}}{m\pi_{i_s}}\right|$$

$$\leq 2\left|\frac{\mathbf{a}_{i_s}^\top\check{\mathbf{Q}}_{-s}\mathbf{a}_{i_s} - \ell_{i_s}^{\mathbf{C}}}{m\pi_{i_s}}\right| + 2d_{\text{eff}}\left|\frac{\ell_{i_s}^{\mathbf{C}}/d_{\text{eff}} - \pi_{i_s}}{m\pi_{i_s}}\right| \leq 2\left|\frac{\mathbf{a}_{i_s}^\top\check{\mathbf{Q}}_{-s}\mathbf{a}_{i_s} - \ell_{i_s}^{\mathbf{C}}}{m\pi_{i_s}}\right| + 2\epsilon_\rho d_{\text{eff}}\frac{\ell_{i_s}^{\mathbf{C}}/d_{\text{eff}}}{m\pi_{i_s}}$$

$$\leq 2\left|\frac{\mathbf{a}_{i_s}^\top\check{\mathbf{Q}}_{-s}\mathbf{a}_{i_s} - \ell_{i_s}^{\mathbf{C}}}{m\pi_{i_s}}\right| + \frac{2\epsilon_\rho\rho_{\max}d_{\text{eff}}}{m}.$$

This, combined with the bound in (34), gives that

$$\|\mathbf{H}^{\frac{1}{2}}\check{\mathbf{Z}}_3\mathbf{H}^{-\frac{1}{2}}\| = O\left(\frac{\epsilon_\rho\rho_{\max}d_{\text{eff}}}{m} + \sqrt{\frac{\rho_{\max}^3 d_{\text{eff}}^3}{m^3}}\right).$$

Recall the proof of Proposition 3.2 that

$$\|\mathbf{H}^{\frac{1}{2}}\check{\mathbf{Z}}_1\mathbf{H}^{-\frac{1}{2}}\| = O\left(\frac{1}{m}\right), \quad \|\mathbf{H}^{\frac{1}{2}}\check{\mathbf{Z}}_2\mathbf{H}^{-\frac{1}{2}}\| = O\left(\sqrt{\frac{\rho_{\max}^3 d_{\text{eff}}^3}{m^3}}\right).$$

Putting these together, we conclude the proof of Corollary 3.4.

In particular, taking $\rho_{\max} = 3/2$ and $\rho_{\min} = 1/2$, we have $\epsilon_\rho = 1$ and thus an inversion bias of order $O(d_{\text{eff}}/m)$. □

### E.3.3. A "COUNTEREXAMPLE" TO DEREZIŃSKI ET AL. (2021B, THEOREM 10)

**Corollary E.2** ("Counterexample" to the lower bound in Dereziński et al. (2021b, Theorem 10) using exact leverage sampling). *For any $n \geq 2d \geq 4$, there exists $\mathbf{A} \in \mathbb{R}^{n \times d}$, an exact leverage score sampling matrix $\mathbf{S} \in \mathbb{R}^{m \times n}$ in Definition 2.2 and a high probability event $\zeta$ (that ensures the invertibility of $\mathbf{A}^\top\mathbf{S}^\top\mathbf{S}\mathbf{A}$), such that when conditioned on $\zeta$, $(\gamma\mathbf{A}^\top\mathbf{S}^\top\mathbf{S}\mathbf{A} + \mathbf{C})^{-1}$ is an unbiased estimator of $(\mathbf{A}^\top\mathbf{A} + \mathbf{C})^{-1}$ for $\gamma = \frac{m}{d}\mathbb{E}_\zeta[1/b]$, $b$ distributed as Binomial$(m, 1/d)$, and $\mathbf{C} = \mathbf{0}_d$.*

*Proof of Corollary E.2.* We begin by recalling the setup of the matrix $\mathbf{A}$ in Dereziński et al. (2021b, Theorem 10). Without loss of generality, assume $n = 2d$ (otherwise, we pad $\mathbf{A}$ with zeros). The matrix $\mathbf{A}$ consists of $n = 2d$ scaled standard basis vectors, where consecutive rows are defined as $\mathbf{a}_{2(i-1)+1}^\top = \mathbf{a}_{2(i-1)+2}^\top = \frac{1}{\sqrt{2}}\mathbf{e}_i^\top$ for $i \geq 2$, and the first two rows are $\mathbf{a}_1^\top = \frac{1}{\sqrt{4}}\mathbf{e}_1^\top$, $\mathbf{a}_2^\top = \frac{3}{\sqrt{4}}\mathbf{e}_1^\top$. This is

$$\mathbf{A} = \begin{bmatrix} \frac{1}{\sqrt{4}} & 0 & \dots & 0 \\ \frac{3}{\sqrt{4}} & 0 & \dots & 0 \\ 0 & \frac{1}{\sqrt{2}} & \dots & 0 \\ 0 & \frac{1}{\sqrt{2}} & \dots & 0 \\ \vdots & \vdots & \ddots & \vdots \\ 0 & 0 & \dots & \frac{1}{\sqrt{2}} \\ 0 & 0 & \dots & \frac{1}{\sqrt{2}} \end{bmatrix}.$$

In this case we have $\mathbf{A}^\top\mathbf{A} = \mathbf{I}_d$ and $\mathbf{C} = \mathbf{0}_d$, so that the leverage scores satisfy $\ell_1^{\mathbf{C}} = \frac{1}{4}$, $\ell_2^{\mathbf{C}} = \frac{3}{4}$, and $\ell_{2(i-1)+1}^{\mathbf{C}} = \ell_{2(i-1)+2}^{\mathbf{C}} = \frac{1}{2}$ for $i \geq 2$. The leverage score sampling distribution $\{\pi_i\}_{i=1}^n$ as in Definition 2.2, used to construct the sampling matrix $\mathbf{S}$ of size $m \geq d$, is thus given by

$$\pi_i = \begin{cases} \frac{1}{4d}, & \text{for } i = 1, \\ \frac{3}{4d}, & \text{for } i = 2, \\ \frac{1}{2d}, & \text{otherwise.} \end{cases}$$

For any $\gamma > 0$, the matrix $\gamma \mathbf{A}^\top \mathbf{S}^\top \mathbf{S} \mathbf{A}$ is diagonal, with diagonal entries given by

$$\left[\gamma \mathbf{A}^\top \mathbf{S}^\top \mathbf{S} \mathbf{A}\right]_{ii} = \frac{\gamma d b_i}{m}, \text{ for } i = 1, \ldots, d,$$

where $b_i$'s are all identically (but not independently) distributed as $\mathrm{Binomial}(m, 1/d)$. Then, conditioning on *any* high probability event $\zeta$ that ensures the invertibility of $\mathbf{A}^\top \mathbf{S}^\top \mathbf{S} \mathbf{A}$, it follows that

$$\mathbb{E}_\zeta\left[\left[\left(\gamma \mathbf{A}^\top \mathbf{S}^\top \mathbf{S} \mathbf{A}\right)^{-1}\right]_{ii}\right] = \frac{m}{\gamma d}\mathbb{E}_\zeta\left[\frac{1}{b_i}\right], \text{ for } i = 1, \ldots, d,$$

so that

$$\mathbb{E}_\zeta\left[\left(\gamma \mathbf{A}^\top \mathbf{S}^\top \mathbf{S} \mathbf{A}\right)^{-1}\right] = \left(\mathbf{A}^\top \mathbf{A}\right)^{-1}$$

with $\gamma = \frac{m}{d}\mathbb{E}_\zeta[1/b]$, where $b$ is any one of the aforementioned $\{b_i\}_{i=1}^d$. This completes the proof of Corollary E.2. $\square$

### E.3.4. PROOF OF COROLLARY 3.7

The proof of Corollary 3.7 comes in the following two steps:

1. construct two *independent* high probability events: $\zeta_\omega$ on which the randomized Walsh–Hadamard transform $\mathbf{H}_n \mathbf{D}_n \mathbf{A}$ of $\mathbf{A}$ as in Definition 3.6 has approximately uniform leverages scores; and $\zeta$ as in (19) to subspace embedding, but for $\mathbf{x}_s^\top = \mathbf{e}_{i_s}^\top \mathbf{H}_n \mathbf{D}_n \mathbf{A}/\sqrt{n\pi_{i_s}} \in \mathbb{R}^d$ the $i_s^{th}$ row of $\tilde{\mathbf{A}}_{\mathrm{SRHT}}$; and

2. conditioned on the event $\zeta \bigcap \zeta_\omega$, apply the bounds in Appendix E.3.2 to obtain inversion bias for $\left(\frac{m}{m-d_{\mathrm{eff}}}\tilde{\mathbf{A}}_{\mathrm{SRHT}}^\top \tilde{\mathbf{A}}_{\mathrm{SRHT}} + \mathbf{C}\right)^{-1}$, where $d_{\mathrm{eff}} = \sum_{i=1}^n \ell_i^{\mathbf{C}}$ is the effective dimension of $\mathbf{A}$ given $\mathbf{C}$.

We start by recalling some notations from (the proof of) Proposition 3.2. Denote $\mathbf{A}_{\mathbf{C}} = \mathbf{A}\mathbf{H}^{-1/2}$ with $\mathbf{H} = \mathbf{A}^\top \mathbf{A} + \mathbf{C}$ and let $\ell_i^{\mathbf{C}}(\mathbf{H}_n \mathbf{D}_n \mathbf{A}/\sqrt{n}) = \|\mathbf{e}_i^\top \mathbf{H}_n \mathbf{D}_n \mathbf{A}_{\mathbf{C}}\|^2/n$ be the $i^{th}$ leverage score of $\mathbf{H}_n \mathbf{D}_n \mathbf{A}/\sqrt{n}$. Define the following event $\zeta_\omega$:

$$\zeta_\omega : \left|\ell_i^{\mathbf{C}}(\mathbf{H}_n \mathbf{D}_n \mathbf{A}/\sqrt{n}) - \frac{d_{\mathrm{eff}}}{n}\right| \leq \omega, \quad \forall 1 \leq i \leq n.$$

Note that this is equivalent to, for uniform sampling with $\pi_i = 1/n$ that

$$\left|\frac{\ell_i^{\mathbf{C}}(\mathbf{H}_n \mathbf{D}_n \mathbf{A}/\sqrt{n}) - d_{\mathrm{eff}}\pi_i}{m\pi_i}\right| \leq \frac{n}{m}\omega, \quad \forall 1 \leq i \leq n.$$

Next, we verify that the event $\zeta_\omega$ holds with a controlled probability. Precisely, some $C > 0$, it follows from Lemma E.4 below and $\|\mathbf{A}_{\mathbf{C}}\|_F^2 = d_{\mathrm{eff}}$ that

$$\Pr\left(\left|\ell_i^{\mathbf{C}}(\mathbf{H}_n \mathbf{D}_n \mathbf{A}/\sqrt{n}) - \frac{d_{\mathrm{eff}}}{n}\right| \geq \omega, 1 \leq i \leq n\right) \leq \frac{\delta}{2}, \tag{36}$$

with $\omega \geq \max\{C\sqrt{d_{\mathrm{eff}}\log(n/\delta)}/n, C\log(n/\delta)/n\}$, which is equivalent to

$$\left|\frac{\ell_i^{\mathbf{C}}(\mathbf{H}_n \mathbf{D}_n \mathbf{A}/\sqrt{n}) - d_{\mathrm{eff}}\pi_i}{m\pi_i}\right| \leq \max\left\{\frac{C\sqrt{d_{\mathrm{eff}}\log(n/\delta)}}{m}, \frac{C\log(n/\delta)}{m}\right\}, \text{ for } 1 \leq i \leq n.$$

Further recall from Lacotte et al. (2021, Lemma 1) that a sampling size $m \geq C\rho_{\max}(d_{\mathrm{eff}} + \log(1/(\epsilon\delta)))\log(d_{\mathrm{eff}}/\delta)/\epsilon^2)$, $\rho_{\max} \equiv \max_{1 \leq i \leq n} n\ell_i^{\mathbf{C}}(\mathbf{H}_n \mathbf{D}_n \mathbf{A}/\sqrt{n})/d_{\mathrm{eff}}$ and $\epsilon \in (0, 1/2]$ suffices to ensure that $\mathbf{H}^{-1/2}\tilde{\mathbf{A}}_{\mathrm{SRHT}}^\top \tilde{\mathbf{A}}_{\mathrm{SRHT}}\mathbf{H}^{-1/2}$ is an $(\epsilon, \delta)$-approximation of $\mathbf{A}_{\mathbf{C}}^\top \mathbf{A}_{\mathbf{C}}$. This leads to $\Pr(\zeta) \leq \delta/2$. Combining the above results, we obtain $\Pr(\zeta \bigcap \zeta_\omega) \leq \delta$. This, together with

$$\|\mathbf{H}^{\frac{1}{2}}\check{\mathbf{Z}}_3\mathbf{H}^{-\frac{1}{2}}\| = O\left(\sqrt{\frac{\rho_{\max}^3 d_{\mathrm{eff}}^3}{m^3}}\right) + O\left(\max_{1 \leq i \leq n}\left|\frac{\ell_i^{\mathbf{C}}(\mathbf{H}_n \mathbf{D}_n \mathbf{A}/\sqrt{n}) - d_{\mathrm{eff}}\pi_i}{m\pi_i}\right|\right),$$

$$\|\mathbf{H}^{\frac{1}{2}}\check{\mathbf{Z}}_1\mathbf{H}^{-\frac{1}{2}}\| = O\left(\frac{1}{m}\right), \ \|\mathbf{H}^{\frac{1}{2}}\check{\mathbf{Z}}_2\mathbf{H}^{-\frac{1}{2}}\| = O\left(\sqrt{\frac{\rho_{\max}^3 d_{\mathrm{eff}}^3}{m^3}}\right).$$

in Appendix E.3.2, concludes the proof of Corollary 3.7. $\square$

*Remark* E.3 (SRHT inversion bias with different sampling sizes). Recalling (36), we observe that

$$\rho_{\max} = \max_{1 \leq i \leq n} n\ell_i^{\mathbf{C}}(\mathbf{H}_n \mathbf{D}_n \mathbf{A}/\sqrt{n})/d_{\mathrm{eff}} \leq 2$$

holds with the probability at least $1 - \delta/2$. Consequently, choosing $m \geq C(d_{\mathrm{eff}} + \log(1/(\epsilon\delta))) \log(d_{\mathrm{eff}}/\delta)/\epsilon^2)$ suffices to ensure that $\mathbf{H}^{-1/2}\tilde{\mathbf{A}}_{\mathrm{SRHT}}^\top \tilde{\mathbf{A}}_{\mathrm{SRHT}} \mathbf{H}^{-1/2}$ is an $(\epsilon, \delta)$-approximation (or subspace embedding) of $\mathbf{A}_{\mathbf{C}}^\top \mathbf{A}_{\mathbf{C}}$. On the other hand, note from Corollary 3.7 that

1. by taking $\nu = 0$, a number of $m = \Theta(d_{\mathrm{eff}})$ samples with $n \exp(-d_{\mathrm{eff}}) < \delta < m^{-3}$, we have that $(\frac{m}{m-d_{\mathrm{eff}}}\tilde{\mathbf{A}}_{\mathrm{SRHT}}^\top \tilde{\mathbf{A}}_{\mathrm{SRHT}} + \mathbf{C})^{-1}$ is an $(O(1), \delta)$-unbiased estimator of $(\mathbf{A}^\top \mathbf{A} + \mathbf{C})^{-1}$: This, in particular, agrees with the above subspace embedding condition up to logarithmic factor.

2. taking $\nu > 1$ in Corollary 3.7 allows one to (further) reduce the inversion bias (to a level that is significantly smaller than $O(1)$) by increasing the sample size $m = \Theta(d_{\mathrm{eff}}^{1+\nu})$.

**Lemma E.4** (Row norms). *Let $\mathbf{H}_n \in \mathbb{R}^{n \times n}$ be the Walsh–Hadamard matrix of size $n \geq 4$ as in Definition 3.6 and $\mathbf{D}_n = \mathrm{diag}(\boldsymbol{v}) \in \mathbb{R}^{n \times n}$ with $\boldsymbol{v} \in \mathbb{R}^n$ a Rademacher random vector. Then, we have, for a matrix $\mathbf{X} \in \mathbb{R}^{n \times d}$ with $n \geq d$ and $t \geq \max\{\|\mathbf{X}\mathbf{X}^\top\|_F \sqrt{\log(2n/\delta)/(cn^2)}, \|\mathbf{X}\|^2 \log(2n/\delta)/(cn)\}$ that*

$$\Pr\left(\left|\frac{\|\mathbf{e}_i^\top \mathbf{H}_n \mathbf{D}_n \mathbf{X}\|^2}{n} - \frac{\|\mathbf{X}\|_F^2}{n}\right| \geq t, 1 \leq i \leq n\right) \leq \delta,$$

*where $\|\cdot\|_F$ denotes the Frobenius norm, and $c > 0$ is a universal constant.*

*Proof of Lemma E.4.* Recall from Definition 3.6 that both $\mathbf{H}_n$ and $\mathbf{D}_n$ are orthogonal matrices such that $\mathbf{H}_n^\top \mathbf{D}_n^2 \mathbf{H}_n/n = \mathbf{I}_n$. Fix a row index $i \in \{1, \ldots, n\}$ and consider

$$\frac{\|\mathbf{e}_i^\top \mathbf{H}_n \mathbf{D}_n \mathbf{X}\|^2}{n} = \|\boldsymbol{v}^\top \mathbf{E} \mathbf{X}\|^2,$$

where $\mathbf{E} = \mathrm{diag}(\mathbf{e}_i^\top \mathbf{H}_n/\sqrt{n})$ is a diagonal matrix formed from the $i^{th}$ row of $\mathbf{H}_n/\sqrt{n}$. Observe that $\mathbf{E}^2 = \frac{1}{n}\mathbf{I}_n$, we further have

$$\mathbb{E}[\boldsymbol{v}^\top \mathbf{E} \mathbf{X} \mathbf{X}^\top \mathbf{E} \boldsymbol{v}] = \mathrm{tr}(\mathbb{E}[\mathbf{X}^\top \mathbf{E} \boldsymbol{v} \boldsymbol{v}^\top \mathbf{E} \mathbf{X}]) = \mathrm{tr}(\mathbf{X}^\top \mathbf{E}^2 \mathbf{X}) = \frac{\|\mathbf{X}\|_F^2}{n}.$$

Note that

$$\|\mathbf{E} \mathbf{X} \mathbf{X}^\top \mathbf{E}\| = \|\mathbf{X}^\top \mathbf{E}^2 \mathbf{X}\| = \frac{\|\mathbf{X}\|^2}{n},$$

$$\|\mathbf{E} \mathbf{X} \mathbf{X}^\top \mathbf{E}\|_F^2 = \mathrm{tr}(\mathbf{E} \mathbf{X} \mathbf{X}^\top \mathbf{E}^2 \mathbf{X} \mathbf{X}^\top \mathbf{E}) = \mathrm{tr}(\mathbf{X} \mathbf{X}^\top \mathbf{E}^2 \mathbf{X} \mathbf{X}^\top \mathbf{E}^2) = \frac{\|\mathbf{X}\mathbf{X}^\top\|_F^2}{n^2},$$

and

$$K = \max_{1 \leq i \leq n} \inf\{s > 0 : \mathbb{E}[\exp(v_i^2/s^2)] \leq 2\} = (\log(2))^{-1/2} > 1,$$

where $v_i$ is the $i^{th}$ variable of $\boldsymbol{v}$. Apply Hanson-Wright inequality in Theorem A.5 with $t \geq \max\{\log(2)\|\mathbf{X}\mathbf{X}^\top\|_F \sqrt{\log(2n/\delta)/(cn^2)}, \log(2)\|\mathbf{X}\|^2 \log(2n/\delta)/(cn)\}$, for each $i = 1, \ldots, n$, we have

$$\Pr\left(\left|\frac{\|\mathbf{e}_i^\top \mathbf{H}_n \mathbf{D}_n \mathbf{X}\|^2}{n} - \frac{\|\mathbf{X}\|_F^2}{n}\right| \geq t\right) \leq \frac{\delta}{n}.$$

Taking a union bound over these $n$ events, we conclude the proof of Lemma E.4. $\square$

## F. Proof of the Results in Section 4

In this section, we present proof of the results in Section 4. Precisely, to establish convergence guarantees for the bias-corrected SSN iteration in (9), we first introduce in Proposition F.1 a fine-grained non-asymptotic bound on second moment of the normalized sub-sampled inverse matrix in Proposition F.1 and present its proof in Appendix F.1. Stemming from this second inverse moment analysis, we also provide non-asymptotic bounds on the second inverse moment for the exact and/or approximate leverage sampling and SRHT in Corollary F.2 and Corollary F.3, respectively, under scalar debiasing. Then, in Appendix F.2, we provide the detailed proof of Theorem 4.3. Finally, further discussions and additional results related to Theorem 4.3 are provided in Appendix F.3.

**Proposition F.1** (Fine-grained analysis of second inverse moment). *For a given matrix $\mathbf{A} \in \mathbb{R}^{n \times d}$, let $\mathbf{S}$ be a random sampling matrix with number of trials $m$ and importance sampling distribution $\{\pi_i\}_{i=1}^n$ as in Definition 2.1, and let $\mathbf{C} \in \mathbb{R}^{d \times d}$ be a p.s.d. matrix and $\mathbf{C_A} = (\mathbf{A}^\top \mathbf{A} + \mathbf{C})^{-1/2} \mathbf{C} (\mathbf{A}^\top \mathbf{A} + \mathbf{C})^{-1/2} \in \mathbb{R}^{d \times d}$. Define $d_{\mathrm{eff}} = \mathrm{tr}(\mathbf{A_C^\top A_C})$ with $\mathbf{A_C} \equiv \mathbf{A}(\mathbf{A}^\top \mathbf{A} + \mathbf{C})^{-1/2} \in \mathbb{R}^{n \times d}$ and the fine-grained de-biased sampling matrix $\check{\mathbf{S}} = \mathrm{diag}\left\{ \sqrt{m/(m - \ell_{i_s}^{\mathbf{C}}/\pi_{i_s})} \right\}_{s=1}^m \cdot \mathbf{S}$ as in Proposition 3.2. Then, for diagonal matrix $\bar{\mathbf{F}} = \mathrm{diag}\{\bar{F}_{ii}\}_{i=1}^n$ with*

$$\bar{F}_{ii} = \frac{\mathbf{a_C}_i^\top \mathbb{E}_\zeta[(\mathbf{A_C^\top}\check{\mathbf{S}}^\top \check{\mathbf{S}} \mathbf{A_C} + \mathbf{C_A})^{-2}]\mathbf{a_C}_i}{m\pi_i},$$

*and $\mathbf{a_C}_i^\top \in \mathbb{R}^d$ the $i^{th}$ row of $\mathbf{A_C}$, there exists universal constant $C > 0$ independent of $n, d_{\mathrm{eff}}$ such that for $m \geq C\rho_{\max} d_{\mathrm{eff}}(\log(d_{\mathrm{eff}}/\delta) + 1/\epsilon^{2/3})$ with $\delta \leq m^{-3}$ and max factor $\rho_{\max} = \max_{1 \leq i \leq n} \ell_i^{\mathbf{C}}/(\pi_i d_{\mathrm{eff}})$ in Definition 2.3, $(\mathbf{A_C^\top}\check{\mathbf{S}}^\top \check{\mathbf{S}}\mathbf{A_C} + \mathbf{C_A})^{-2}$ is an $(\epsilon, \delta)$-unbiased estimator of $\mathbf{I}_d + \mathbf{A_C^\top}\bar{\mathbf{F}}\mathbf{A_C}$.*

Note that Proposition F.1 can be seen as a second-order extension of (the first-order inversion bias in) Proposition 3.2. As we shall see below in Appendix F.2, this result is instrumental in establishing the convergence rate for SSN in Theorem 4.3.

### F.1. Proof of Proposition F.1

Following the methodology of the proofs of Theorem 3.1 and Proposition 3.2, the proof of Proposition F.1 also comes in the following two steps:

1. construct an high probability event $\zeta$ as in (19); and

2. conditional on the event $\zeta$, derive a bound for the spectral norm $\|\mathbb{E}_\zeta[(\mathbf{A_C^\top}\check{\mathbf{S}}^\top\check{\mathbf{S}}\mathbf{A_C} + \mathbf{C_A})^{-2}] - (\mathbf{I}_d + \mathbf{A_C^\top}\bar{\mathbf{F}}\mathbf{A_C})\|$, using again "leave-one-out" type analysis.

First, let us recall some notations from the proofs of Theorem 3.1 and Proposition 3.2. For the ease of further use, denote

$$\check{\mathbf{S}} = \mathrm{diag}\left\{F_{i_s i_s}\right\}_{s=1}^m \cdot \mathbf{S}, \quad i_s \in \{1, \ldots, n\}, \quad F_{i_s i_s} = \sqrt{m/(m - \ell_{i_s}^{\mathbf{C}}/\pi_{i_s})},$$

$\mathbf{H} = \mathbf{A}^\top \mathbf{A} + \mathbf{C}$, $\mathbf{A_C} = \mathbf{A}\mathbf{H}^{-1/2}$, and $\hat{\mathbf{x}}_s^\top = \mathbf{e}_{i_s}^\top \mathbf{A_C}/\sqrt{\pi_{i_s}}$ such that $\mathbb{E}[\hat{\mathbf{x}}_s\hat{\mathbf{x}}_s^\top] = \mathbf{A_C^\top A_C}$. Further let

$$\hat{\mathbf{Q}} = (\mathbf{A_C^\top}\check{\mathbf{S}}^\top\check{\mathbf{S}}\mathbf{A_C} + \mathbf{C_A})^{-1} = \left(\sum_{s=1}^m \frac{1}{m}F_{i_s i_s}\hat{\mathbf{x}}_s\hat{\mathbf{x}}_s^\top + \mathbf{C_A}\right)^{-1}, \quad \text{and } \hat{\mathbf{Q}}_{-s} = \left(\sum_{j \neq s}^m \frac{1}{m}F_{jj}\hat{\mathbf{x}}_j\hat{\mathbf{x}}_j^\top + \mathbf{C_A}\right)^{-1}.$$

To prove Proposition F.1, we first rewrite

$$\mathbb{E}_\zeta[\hat{\mathbf{Q}}^2] - (\mathbf{I}_d + \mathbf{A_C^\top}\bar{\mathbf{F}}\mathbf{A_C}) = \underbrace{\mathbb{E}_\zeta[\hat{\mathbf{Q}} - \mathbf{I}_d]}_{\mathbf{T}_1} + \mathbb{E}_\zeta[\hat{\mathbf{Q}}(\hat{\mathbf{Q}} - \mathbf{I}_d)] - \mathbf{A_C^\top}\bar{\mathbf{F}}\mathbf{A_C}. \tag{37}$$

Then, along with $\mathbf{A}_{\mathbf{C}}^\top \mathbf{A}_{\mathbf{C}} + \mathbf{C}_{\mathbf{A}} = \mathbf{I}_d$ and let $\hat{\gamma}_s = 1 + \frac{1}{m} F_{i_s i_s} \hat{\mathbf{x}}_s^\top \hat{\mathbf{Q}}_{-s} \hat{\mathbf{x}}_s = \check{\gamma}_s$, $s = 1, \ldots, m$, we get

$$
\mathbb{E}_\zeta[\hat{\mathbf{Q}}(\hat{\mathbf{Q}} - \mathbf{I}_d)] = \mathbb{E}_\zeta[\hat{\mathbf{Q}}(\hat{\mathbf{Q}}(\mathbf{A}_{\mathbf{C}}^\top \mathbf{A}_{\mathbf{C}} - \mathbf{A}_{\mathbf{C}}^\top \check{\mathbf{S}}^\top \check{\mathbf{S}} \mathbf{A}_{\mathbf{C}}))] = \mathbb{E}_\zeta[\hat{\mathbf{Q}}^2 \mathbf{A}_{\mathbf{C}}^\top \mathbf{A}_{\mathbf{C}}] - \mathbb{E}_\zeta \left[ \sum_{s=1}^m \hat{\mathbf{Q}} \frac{F_{i_s i_s} \hat{\mathbf{Q}}_{-s} \hat{\mathbf{x}}_s \hat{\mathbf{x}}_s^\top}{m \hat{\gamma}_s} \right]
$$

$$
= \mathbb{E}_\zeta[\hat{\mathbf{Q}}^2 \mathbf{A}_{\mathbf{C}}^\top \mathbf{A}_{\mathbf{C}}] - \mathbb{E}_\zeta \left[ \sum_{s=1}^m \frac{F_{i_s i_s} \hat{\mathbf{Q}}_{-s}^2 \hat{\mathbf{x}}_s \hat{\mathbf{x}}_s^\top}{m \check{\gamma}_s} \right] + \mathbb{E}_\zeta \left[ \sum_{s=1}^m \frac{F_{i_s i_s}^2 \hat{\mathbf{Q}}_{-s} \hat{\mathbf{x}}_s \hat{\mathbf{x}}_s^\top \hat{\mathbf{Q}}_{-s}^2 \hat{\mathbf{x}}_s \hat{\mathbf{x}}_s^\top}{m^2 \check{\gamma}_s^2} \right]
$$

$$
= \mathbb{E}_\zeta[\hat{\mathbf{Q}}^2 \mathbf{A}_{\mathbf{C}}^\top \mathbf{A}_{\mathbf{C}}] - \mathbb{E}_\zeta \left[ \frac{F_{i_s i_s} \hat{\mathbf{Q}}_{-s}^2 \hat{\mathbf{x}}_s \hat{\mathbf{x}}_s^\top}{\check{\gamma}_s} \right] + \mathbb{E}_\zeta \left[ \frac{F_{i_s i_s}^2 \hat{\mathbf{Q}}_{-s} \hat{\mathbf{x}}_s \hat{\mathbf{x}}_s^\top \hat{\mathbf{Q}}_{-s}^2 \hat{\mathbf{x}}_s \hat{\mathbf{x}}_s^\top}{m \check{\gamma}_s^2} \right]
$$

$$
= \underbrace{\mathbb{E}_\zeta \left[ \hat{\mathbf{Q}}^2 \mathbf{A}_{\mathbf{C}}^\top \mathbf{A}_{\mathbf{C}} - \hat{\mathbf{Q}}_{-s}^2 \hat{\mathbf{x}}_s \hat{\mathbf{x}}_s^\top \right]}_{\mathbf{T}_2} + \underbrace{\mathbb{E}_\zeta \left[ \hat{\mathbf{Q}}_{-s}^2 \left( 1 - \frac{F_{i_s i_s}}{\check{\gamma}_s} \right) \hat{\mathbf{x}}_s \hat{\mathbf{x}}_s^\top \right]}_{\mathbf{T}_3} + \mathbb{E}_\zeta \left[ \frac{F_{i_s i_s}^2 \hat{\mathbf{Q}}_{-s} \hat{\mathbf{x}}_s \hat{\mathbf{x}}_s^\top \hat{\mathbf{Q}}_{-s}^2 \hat{\mathbf{x}}_s \hat{\mathbf{x}}_s^\top}{m \check{\gamma}_s^2} \right], \tag{38}
$$

where the second and third equalities follow from the Sherman–Morrison formula. Further define $\bar{\mathbf{F}}' = \text{diag}\{\bar{F}_{ii}'\}_{i=1}^n$ with $\bar{F}_{ii}' = \mathbf{a}_{\mathbf{C}i}^\top \mathbb{E}_\zeta[\hat{\mathbf{Q}}_{-s}^2] \mathbf{a}_{\mathbf{C}i}/m\pi_i$. Using (37) and (38), we get

$$
\mathbb{E}_\zeta[\hat{\mathbf{Q}}^2] - (\mathbf{I}_d + \mathbf{A}_{\mathbf{C}}^\top \bar{\mathbf{F}} \mathbf{A}_{\mathbf{C}}) = \mathbf{T}_1 + \mathbf{T}_2 + \mathbf{T}_3 + \mathbb{E}_\zeta \left[ \frac{F_{i_s i_s}^2 \hat{\mathbf{Q}}_{-s} \hat{\mathbf{x}}_s \hat{\mathbf{x}}_s^\top \hat{\mathbf{Q}}_{-s}^2 \hat{\mathbf{x}}_s \hat{\mathbf{x}}_s^\top}{m \check{\gamma}_s^2} \right] - \mathbf{A}_{\mathbf{C}}^\top \bar{\mathbf{F}} \mathbf{A}_{\mathbf{C}}
$$

$$
= \mathbf{T}_1 + \mathbf{T}_2 + \mathbf{T}_3 + \underbrace{\mathbb{E}_\zeta[\hat{\mathbf{Q}} - \mathbf{I}_d] \mathbf{A}_{\mathbf{C}}^\top \bar{\mathbf{F}} \mathbf{A}_{\mathbf{C}}}_{\mathbf{T}_4} + \underbrace{\mathbb{E}_\zeta[(\hat{\mathbf{Q}}_{-s} - \hat{\mathbf{Q}}) \mathbf{A}_{\mathbf{C}}^\top \bar{\mathbf{F}} \mathbf{A}_{\mathbf{C}}]}_{\mathbf{T}_5}
$$

$$
+ \underbrace{\mathbb{E}_\zeta[\hat{\mathbf{Q}}_{-s}(\mathbf{A}_{\mathbf{C}}^\top \bar{\mathbf{F}}' \mathbf{A}_{\mathbf{C}} - \mathbf{A}_{\mathbf{C}}^\top \bar{\mathbf{F}} \mathbf{A}_{\mathbf{C}})]}_{\mathbf{T}_6}
$$

$$
+ \underbrace{\mathbb{E}_\zeta \left[ \frac{\hat{\mathbf{x}}_s^\top \hat{\mathbf{Q}}_{-s}^2 \hat{\mathbf{x}}_s \hat{\mathbf{Q}}_{-s} \hat{\mathbf{x}}_s \hat{\mathbf{x}}_s^\top}{m} \right] - \mathbb{E}_\zeta[\hat{\mathbf{Q}}_{-s}(\mathbf{A}_{\mathbf{C}}^\top \bar{\mathbf{F}}' \mathbf{A}_{\mathbf{C}})]}_{\mathbf{T}_7}
$$

$$
+ \underbrace{\mathbb{E}_\zeta \left[ \left( \frac{F_{i_s i_s}^2}{\check{\gamma}_s^2} - 1 \right) \frac{\hat{\mathbf{x}}_s^\top \hat{\mathbf{Q}}_{-s}^2 \hat{\mathbf{x}}_s \hat{\mathbf{Q}}_{-s} \hat{\mathbf{x}}_s \hat{\mathbf{x}}_s^\top}{m} \right]}_{\mathbf{T}_8}.
$$

At this point, note from Proposition 3.2 that $\|\mathbf{T}_1\| = O\left(\sqrt{\rho_{\max}^3 d_{\text{eff}}^3 / m^3}\right)$. Using $\hat{\mathbf{Q}} = \mathbf{H}^{1/2} \check{\mathbf{Q}} \mathbf{H}^{1/2}$, and noting the fact that $\zeta'$ implies that

$$
\check{\mathbf{Q}} \preceq \check{\mathbf{Q}}_{-s} \preceq 6 \mathbf{H}^{-1}, \tag{39}
$$

we have $\bar{F}_{ii} \leq \mathbf{a}_{\mathbf{C}i}^\top \mathbb{E}_\zeta[\hat{\mathbf{Q}}^2] \mathbf{a}_{\mathbf{C}i}/m\pi_i \leq 36 \rho_{\max} d_{\text{eff}}/m < 1$, which, together with Proposition 3.2 yields that $\|\mathbf{T}_4\| = O\left(\sqrt{\rho_{\max}^3 d_{\text{eff}}^3 / m^3}\right)$.

Now, we proceed to bound $\mathbf{T}_2$. We first write

$$
\|\mathbf{T}_2\| = \|\mathbb{E}_\zeta[\hat{\mathbf{Q}}^2 \mathbf{A}_{\mathbf{C}}^\top \mathbf{A}_{\mathbf{C}}] - \mathbb{E}_\zeta[\hat{\mathbf{Q}}_{-s}^2 \mathbf{A}_{\mathbf{C}}^\top \mathbf{A}_{\mathbf{C}}]\| + \|\mathbb{E}_\zeta[\hat{\mathbf{Q}}_{-s}^2 \mathbf{A}_{\mathbf{C}}^\top \mathbf{A}_{\mathbf{C}}] - \mathbb{E}_\zeta[\hat{\mathbf{Q}}_{-s}^2 \hat{\mathbf{x}}_s \hat{\mathbf{x}}_s^\top]\|
$$

$$
\leq \underbrace{\|\mathbb{E}_\zeta[(\hat{\mathbf{Q}}^2 - \hat{\mathbf{Q}}_{-s}^2) \mathbf{A}_{\mathbf{C}}^\top \mathbf{A}_{\mathbf{C}}]\|}_{T_{21}} + \underbrace{\|\mathbb{E}_\zeta[\hat{\mathbf{Q}}_{-s}^2 (\mathbf{A}_{\mathbf{C}}^\top \mathbf{A}_{\mathbf{C}} - \hat{\mathbf{x}}_s \hat{\mathbf{x}}_s^\top)]\|}_{T_{22}}.
$$

Recalling $\mathbf{A}_{\mathbf{C}}^\top \mathbf{A}_{\mathbf{C}} = \mathbf{H}^{-1/2} \mathbf{A}^\top \mathbf{A} \mathbf{H}^{-1/2} \preceq \mathbf{I}_d$, we get, for the first term $T_{21}$, that

$$
T_{21} \leq \|\mathbb{E}_\zeta[\hat{\mathbf{Q}}^2 - \hat{\mathbf{Q}}_{-s}^2]\| \leq \|\mathbb{E}_\zeta[\hat{\mathbf{Q}}(\hat{\mathbf{Q}} - \hat{\mathbf{Q}}_{-s})]\| + \|\mathbb{E}_\zeta[(\hat{\mathbf{Q}} - \hat{\mathbf{Q}}_{-s})\hat{\mathbf{Q}}_{-s}]\|
$$

$$
\overset{(a)}{=} \left\| \mathbb{E}_\zeta \left[ \frac{F_{i_s i_s}}{m \check{\gamma}_s} \hat{\mathbf{Q}} \hat{\mathbf{Q}}_{-s} \hat{\mathbf{x}}_s \hat{\mathbf{x}}_s^\top \hat{\mathbf{Q}}_{-s} \right] \right\| + \left\| \mathbb{E}_\zeta \left[ \frac{F_{i_s i_s}}{m \check{\gamma}_s} \hat{\mathbf{Q}}_{-s} \hat{\mathbf{x}}_s \hat{\mathbf{x}}_s^\top \hat{\mathbf{Q}}_{-s}^2 \right] \right\|
$$

$$
\overset{(b)}{=} 2 \left\| \mathbb{E}_\zeta \left[ \frac{F_{i_s i_s}}{m \check{\gamma}_s} \hat{\mathbf{Q}}_{-s}^2 \hat{\mathbf{x}}_s \hat{\mathbf{x}}_s^\top \hat{\mathbf{Q}}_{-s} \right] \right\| + \left\| \mathbb{E}_\zeta \left[ \frac{F_{i_s i_s}^2}{m^2 \check{\gamma}_s^2} \hat{\mathbf{Q}}_{-s} \hat{\mathbf{x}}_s \hat{\mathbf{x}}_s^\top \hat{\mathbf{Q}}_{-s}^2 \hat{\mathbf{x}}_s \hat{\mathbf{x}}_s^\top \hat{\mathbf{Q}}_{-s} \right] \right\|, \tag{40}
$$

where we used (again) Sherman–Morrison formula twice in the step $(a)$ and $(b)$, respectively. For the first term in (40), using the fact that $F_{i_s i_s}/\check{\gamma}_s < F_{i_s i_s} < 2$, together with (22) and (39), we get

$$
\begin{aligned}
\left\| \mathbb{E}_\zeta \left[ \frac{F_{i_s i_s}}{m\check{\gamma}_s} \hat{\mathbf{Q}}_{-s}^2 \hat{\mathbf{x}}_s \hat{\mathbf{x}}_s^\top \hat{\mathbf{Q}}_{-s} \right] \right\| &= \sup_{\|\mathbf{u}\|=1,\|\mathbf{v}\|=1} \mathbb{E}_\zeta \left[ \frac{F_{i_s i_s}}{m\check{\gamma}_s} \mathbf{u}^\top \hat{\mathbf{Q}}_{-s}^2 \hat{\mathbf{x}}_s \hat{\mathbf{x}}_s^\top \hat{\mathbf{Q}}_{-s} \mathbf{v} \right] \\
&\leq \frac{4}{m} \sup_{\|\mathbf{u}\|=1,\|\mathbf{v}\|=1} \mathbb{E}_{\zeta'} \left[ |\mathbf{u}^\top \hat{\mathbf{Q}}_{-s}^2 \hat{\mathbf{x}}_s \hat{\mathbf{x}}_s^\top \hat{\mathbf{Q}}_{-s} \mathbf{v}| \right] \\
&\leq \frac{2}{m} \sup_{\|\mathbf{u}\|=1,\|\mathbf{v}\|=1} \mathbb{E}_{\zeta'} \left[ \mathbf{u}^\top \hat{\mathbf{Q}}_{-s}^2 \hat{\mathbf{x}}_s \hat{\mathbf{x}}_s^\top \hat{\mathbf{Q}}_{-s}^2 \mathbf{u} + \mathbf{v}^\top \hat{\mathbf{Q}}_{-s} \hat{\mathbf{x}}_s \hat{\mathbf{x}}_s^\top \hat{\mathbf{Q}}_{-s} \mathbf{v} \right] \\
&\leq \frac{2}{m} \sup_{\|\mathbf{u}\|=1,\|\mathbf{v}\|=1} \mathbb{E}_{\zeta'} \left[ \sum_{j=1}^n \mathbf{u}^\top \hat{\mathbf{Q}}_{-s}^2 \mathbf{a}_{\mathbf{C}_j} \mathbf{a}_{\mathbf{C}_j}^\top \hat{\mathbf{Q}}_{-s}^2 \mathbf{u} + \mathbf{v}^\top \hat{\mathbf{Q}}_{-s} \mathbf{a}_{\mathbf{C}_j} \mathbf{a}_{\mathbf{C}_j}^\top \hat{\mathbf{Q}}_{-s} \mathbf{v} \right] \\
&\leq \frac{2}{m} \sup_{\|\mathbf{u}\|=1,\|\mathbf{v}\|=1} \mathbb{E}_{\zeta'} \left[ \mathbf{u}^\top \hat{\mathbf{Q}}_{-s}^2 \mathbf{A}_{\mathbf{C}}^\top \mathbf{A}_{\mathbf{C}} \hat{\mathbf{Q}}_{-s}^2 \mathbf{u} + \mathbf{v}^\top \hat{\mathbf{Q}}_{-s} \mathbf{A}_{\mathbf{C}}^\top \mathbf{A}_{\mathbf{C}} \hat{\mathbf{Q}}_{-s} \mathbf{v} \right] \\
&= O\left( \frac{1}{m} \right).
\end{aligned}
$$

Analogously as above, we have, for the second term in (40) that

$$
\begin{aligned}
\left\| \mathbb{E}_\zeta \left[ \frac{F_{i_s i_s}^2}{m^2 \check{\gamma}_s^2} \hat{\mathbf{Q}}_{-s} \hat{\mathbf{x}}_s \hat{\mathbf{x}}_s^\top \hat{\mathbf{Q}}_{-s}^2 \hat{\mathbf{x}}_s \hat{\mathbf{x}}_s^\top \hat{\mathbf{Q}}_{-s} \right] \right\| &\leq \frac{8}{m^2} \| \mathbb{E}_{\zeta'} [\hat{\mathbf{Q}}_{-s} \hat{\mathbf{x}}_s \hat{\mathbf{x}}_s^\top \hat{\mathbf{Q}}_{-s}^2 \hat{\mathbf{x}}_s \hat{\mathbf{x}}_s^\top \hat{\mathbf{Q}}_{-s}] \| \\
&= \frac{8}{m^2} \left\| \mathbb{E}_{\zeta'} \left[ \sum_{j=1}^n \frac{\hat{\mathbf{Q}}_{-s} \mathbf{a}_{\mathbf{C}_j} \mathbf{a}_{\mathbf{C}_j}^\top \hat{\mathbf{Q}}_{-s}^2 \mathbf{a}_{\mathbf{C}_j} \mathbf{a}_{\mathbf{C}_j}^\top \hat{\mathbf{Q}}_{-s}}{\pi_j} \right] \right\| \\
&= \frac{288\rho_{\max} d_{\mathrm{eff}}}{m^2} \left\| \mathbb{E}_{\zeta'} \left[ \hat{\mathbf{Q}}_{-s} \mathbf{A}_{\mathbf{C}}^\top \mathbf{A}_{\mathbf{C}} \hat{\mathbf{Q}}_{-s} \right] \right\| = O\left( \frac{\rho_{\max} d_{\mathrm{eff}}}{m^2} \right).
\end{aligned}
$$

We thus conclude that $T_{21} = O(1/m)$.

Following the methodology used to bound $\|\mathbf{H}^{1/2}\check{\mathbf{Z}}_2\mathbf{H}^{-1/2}\|$ and $\|\mathbf{H}^{1/2}\check{\mathbf{Z}}_3\mathbf{H}^{-1/2}\|$ in Appendix E.2, together with (39), we get similarly

$$
T_{22} = O\left( \sqrt{\frac{\rho_{\max}^3 d_{\mathrm{eff}}^3}{m^3}} \right), \quad \|\mathbf{T}_3\| = O\left( \sqrt{\frac{\rho_{\max}^3 d_{\mathrm{eff}}^3}{m^3}} \right).
$$

Utilizing the fact that $\bar{F}_{ii} \leq 36\rho_{\max} d_{\mathrm{eff}}/m$, together with (32) and (39), it follows that

$$
\|\mathbf{T}_5\| \leq \|\mathbb{E}_\zeta[\hat{\mathbf{Q}}_{-s} - \hat{\mathbf{Q}}]\| \|\mathbf{A}_{\mathbf{C}}^\top \bar{\mathbf{F}} \mathbf{A}_{\mathbf{C}}\| \leq O\left( \frac{\rho_{\max} d_{\mathrm{eff}}}{m^2} \right).
$$

Since (39) and $T_{21} = O(1/m)$, we can bound $\|\mathbf{T}_6\|$ as

$$
\begin{aligned}
\|\mathbf{T}_6\| &\leq \|\mathbb{E}_\zeta[\hat{\mathbf{Q}}_{-s}]\| \|\mathbf{A}_{\mathbf{C}}\|^2 \|\bar{\mathbf{F}}' - \bar{\mathbf{F}}\| \leq 6 \max_{1 \leq j \leq n} \frac{|\mathbf{a}_{\mathbf{C}_j}^\top (\mathbb{E}_\zeta[\hat{\mathbf{Q}}_{-s}^2] - \mathbb{E}_\zeta[\hat{\mathbf{Q}}^2]) \mathbf{a}_{\mathbf{C}_j}|}{m\pi_j} \\
&\leq \frac{6\rho_{\max} d_{\mathrm{eff}}}{m} \|\mathbb{E}_\zeta[\hat{\mathbf{Q}}_{-s}^2] - \mathbb{E}_\zeta[\hat{\mathbf{Q}}^2]\| = O\left( \frac{\rho_{\max} d_{\mathrm{eff}}}{m^2} \right).
\end{aligned}
$$

We then move on to bound $\|\mathbf{T}_7\|$. We start by rewriting

$$
\begin{aligned}
\|\mathbf{T}_7\| = \left\| \mathbb{E}_\zeta \left[ \hat{\mathbf{Q}}_{-s} \left( \mathbf{A}_{\mathbf{C}}^\top \bar{\mathbf{F}}' \mathbf{A}_{\mathbf{C}} - \frac{\hat{\mathbf{x}}_s^\top \hat{\mathbf{Q}}_{-s}^2 \hat{\mathbf{x}}_s \hat{\mathbf{x}}_s \hat{\mathbf{x}}_s^\top}{m} \right) \right] \right\| &\leq \underbrace{\left\| \mathbb{E}_\zeta \left[ \hat{\mathbf{Q}}_{-s} \left( \mathbf{A}_{\mathbf{C}}^\top \bar{\mathbf{F}}' \mathbf{A}_{\mathbf{C}} - \frac{\hat{\mathbf{x}}_s^\top \mathbb{E}_\zeta[\hat{\mathbf{Q}}_{-s}^2] \hat{\mathbf{x}}_s \hat{\mathbf{x}}_s \hat{\mathbf{x}}_s^\top}{m} \right) \right] \right\|}_{T_{71}} \\
+ \underbrace{\left\| \mathbb{E}_\zeta \left[ \hat{\mathbf{Q}}_{-s} \frac{\hat{\mathbf{x}}_s^\top (\mathbb{E}_{\zeta'}[\hat{\mathbf{Q}}_{-s}^2] - \hat{\mathbf{Q}}_{-s}^2) \hat{\mathbf{x}}_s \hat{\mathbf{x}}_s \hat{\mathbf{x}}_s^\top}{m} \right] \right\|}_{T_{72}} &+ \underbrace{\left\| \mathbb{E}_\zeta \left[ \hat{\mathbf{Q}}_{-s} \frac{\hat{\mathbf{x}}_s^\top (\mathbb{E}_\zeta[\hat{\mathbf{Q}}_{-s}^2] - \mathbb{E}_{\zeta'}[\hat{\mathbf{Q}}_{-s}^2]) \hat{\mathbf{x}}_s \hat{\mathbf{x}}_s \hat{\mathbf{x}}_s^\top}{m} \right] \right\|}_{T_{73}}.
\end{aligned}
$$

Note that $\mathbb{E}_{\zeta'}\left[\hat{\mathbf{Q}}_{-s}\left(\mathbf{A}_{\mathbf{C}}^\top\bar{\mathbf{F}}'\mathbf{A}_{\mathbf{C}} - \frac{\hat{\mathbf{x}}_s^\top\mathbb{E}_\zeta[\hat{\mathbf{Q}}^2_{-s}]\hat{\mathbf{x}}_s\hat{\mathbf{x}}_s\hat{\mathbf{x}}_s^\top}{m}\right)\right] = 0$. By adapting the techniques of bounding $\|\mathbf{H}^{1/2}\check{\mathbf{Z}}_2\mathbf{H}^{-1/2}\|$ in Appendix E.2, for $\delta_3 < m^{-3}$, we further get

$$T_{71} \leq 2\left\|\mathbb{E}_{\zeta'}\left[\hat{\mathbf{Q}}_{-s}\left(\mathbf{A}_{\mathbf{C}}^\top\bar{\mathbf{F}}'\mathbf{A}_{\mathbf{C}} - \frac{\hat{\mathbf{x}}_s^\top\mathbb{E}_\zeta[\hat{\mathbf{Q}}^2_{-s}]\hat{\mathbf{x}}_s\hat{\mathbf{x}}_s^\top}{m}\right)\cdot\mathbf{1}_{\neg\zeta_3}\right]\right\|$$

$$\leq O\left(\frac{\rho_{\max}d_{\text{eff}}}{m^4}\right) + 12\mathbb{E}_{\zeta'}\left[\frac{\hat{\mathbf{x}}_s^\top\mathbb{E}_\zeta[\hat{\mathbf{Q}}^2_{-s}]\hat{\mathbf{x}}_s\hat{\mathbf{x}}_s^\top\hat{\mathbf{x}}_s}{m}\cdot\mathbf{1}_{\neg\zeta_3}\right].$$

Furthermore, applying the Chebyshev's inequality again, and considering $\mathbb{E}_{\zeta'}[\hat{\mathbf{x}}_s^\top\mathbb{E}_\zeta[\hat{\mathbf{Q}}^2_{-s}]\hat{\mathbf{x}}_s\hat{\mathbf{x}}_s^\top\hat{\mathbf{x}}_s/m] \leq 36\rho_{\max}d_{\text{eff}}^2/m$, alongside

$$\text{Var}_{\zeta'}\left[\frac{\hat{\mathbf{x}}_s^\top\mathbb{E}_\zeta[\hat{\mathbf{Q}}^2_{-s}]\hat{\mathbf{x}}_s\hat{\mathbf{x}}_s^\top\hat{\mathbf{x}}_s}{m}\right] \leq \mathbb{E}_{\zeta'}\left[\frac{(\hat{\mathbf{x}}_s^\top\mathbb{E}_\zeta[\hat{\mathbf{Q}}^2_{-s}]\hat{\mathbf{x}}_s\hat{\mathbf{x}}_s^\top\hat{\mathbf{x}}_s)^2}{m^2}\right] \leq \frac{6^4\rho_{\max}^3d_{\text{eff}}^4}{m^2},$$

it follows that, for $x \geq 72\rho_{\max}d_{\text{eff}}^2/m$, $\Pr(\hat{\mathbf{x}}_s^\top\mathbb{E}_\zeta[\hat{\mathbf{Q}}^2_{-s}]\hat{\mathbf{x}}_s\hat{\mathbf{x}}_s^\top\hat{\mathbf{x}}_s/m \geq x \mid \zeta') \leq 6^4\rho_{\max}^3d_{\text{eff}}^4/(m^2x^2)$. Subsequently, by $\delta_3 < m^{-3}$, we deduce that

$$\mathbb{E}_{\zeta'}\left[\frac{\hat{\mathbf{x}}_s^\top\mathbb{E}_\zeta[\hat{\mathbf{Q}}^2_{-s}]\hat{\mathbf{x}}_s\hat{\mathbf{x}}_s^\top\hat{\mathbf{x}}_s}{m}\cdot\mathbf{1}_{\neg\zeta_3}\right] = \int_0^\infty\Pr\left(\frac{\hat{\mathbf{x}}_s^\top\mathbb{E}_\zeta[\hat{\mathbf{Q}}^2_{-s}]\hat{\mathbf{x}}_s\hat{\mathbf{x}}_s^\top\hat{\mathbf{x}}_s}{m}\cdot\mathbf{1}_{\neg\zeta_3} \geq x \mid \zeta'\right)dx$$

$$\leq 2m^2\delta_3 + \int_{2m^2}^\infty\Pr\left(\frac{\hat{\mathbf{x}}_s^\top\mathbb{E}_\zeta[\hat{\mathbf{Q}}^2_{-s}]\hat{\mathbf{x}}_s\hat{\mathbf{x}}_s^\top\hat{\mathbf{x}}_s}{m} \geq x \mid \zeta'\right)dx$$

$$\leq \frac{2}{m} + \frac{6^4\rho_{\max}^3d_{\text{eff}}^4}{m^2}\int_{2m^2}^\infty\frac{1}{x^2}dx = \frac{2}{m} + \frac{6^4\rho_{\max}^3d_{\text{eff}}^4}{2m^4} = O\left(\sqrt{\frac{\rho_{\max}^3d_{\text{eff}}^3}{m^3}}\right).$$

This leads to $T_{71} = O\left(\sqrt{\rho_{\max}^3d_{\text{eff}}^3/m^3}\right)$.

Next, we show a bound on $T_{72}$. Using again (22) and (39), we get

$$T_{72} \leq \sup_{\|\mathbf{u}\|=1, \|\mathbf{v}\|=1}\mathbb{E}_\zeta\left[\left|\hat{\mathbf{x}}_s^\top(\mathbb{E}_{\zeta'}[\hat{\mathbf{Q}}^2_{-s}] - \hat{\mathbf{Q}}^2_{-s})\hat{\mathbf{x}}_s\right|\left|\frac{\mathbf{u}^\top\hat{\mathbf{Q}}_{-s}\hat{\mathbf{x}}_s\hat{\mathbf{x}}_s^\top\mathbf{v}}{m}\right|\right]$$

$$\leq 2\sup_{\|\mathbf{u}\|=1, \|\mathbf{v}\|=1}\mathbb{E}_{\zeta'}\left[\left|\hat{\mathbf{x}}_s^\top(\mathbb{E}_{\zeta'}[\hat{\mathbf{Q}}^2_{-s}] - \hat{\mathbf{Q}}^2_{-s})\hat{\mathbf{x}}_s\right|\left|\frac{\mathbf{u}^\top\hat{\mathbf{Q}}_{-s}\hat{\mathbf{x}}_s\hat{\mathbf{x}}_s^\top\mathbf{v}}{m}\right|\right]$$

$$\leq \frac{1}{m}\sup_{\|\mathbf{u}\|=1, \|\mathbf{v}\|=1}\mathbb{E}_{\zeta'}\left[\left|\hat{\mathbf{x}}_s^\top(\mathbb{E}_{\zeta'}[\hat{\mathbf{Q}}^2_{-s}] - \hat{\mathbf{Q}}^2_{-s})\hat{\mathbf{x}}_s\right|\left(\mathbf{u}^\top\hat{\mathbf{Q}}_{-s}\hat{\mathbf{x}}_s\hat{\mathbf{x}}_s^\top\hat{\mathbf{Q}}_{-s}\mathbf{u} + \mathbf{v}^\top\hat{\mathbf{x}}_s\hat{\mathbf{x}}_s^\top\mathbf{v}\right)\right]$$

$$\leq \frac{1}{m}\sup_{\|\mathbf{u}\|=1, \|\mathbf{v}\|=1}\mathbb{E}_{\zeta'}\left[\sum_{j=1}^n\frac{|\mathbf{a}_{\mathbf{C}j}^\top(\mathbb{E}_{\zeta'}[\hat{\mathbf{Q}}^2_{-s}] - \hat{\mathbf{Q}}^2_{-s})\mathbf{a}_{\mathbf{C}j}|}{\pi_j}\left(\mathbf{u}^\top\hat{\mathbf{Q}}_{-s}\mathbf{a}_{\mathbf{C}j}\mathbf{a}_{\mathbf{C}j}^\top\hat{\mathbf{Q}}_{-s}\mathbf{u} + \mathbf{v}^\top\mathbf{a}_{\mathbf{C}j}\mathbf{a}_{\mathbf{C}j}^\top\mathbf{v}\right)\right]$$

$$\leq \frac{1}{m}\sup_{\|\mathbf{u}\|=1}\mathbb{E}_{\zeta'}\left[\max_{1\leq j\leq n}\frac{|\mathbf{a}_{\mathbf{C}j}^\top(\mathbb{E}_{\zeta'}[\hat{\mathbf{Q}}^2_{-s}] - \hat{\mathbf{Q}}^2_{-s})\mathbf{a}_{\mathbf{C}j}|}{\pi_j}\sum_{j=1}^n\mathbf{u}^\top\hat{\mathbf{Q}}_{-s}\mathbf{a}_{\mathbf{C}j}\mathbf{a}_{\mathbf{C}j}^\top\hat{\mathbf{Q}}_{-s}\mathbf{u}\right]$$

$$+ \frac{1}{m}\sup_{\|\mathbf{v}\|=1}\mathbb{E}_{\zeta'}\left[\max_{1\leq j\leq n}\frac{|\mathbf{a}_{\mathbf{C}j}^\top(\mathbb{E}_{\zeta'}[\hat{\mathbf{Q}}^2_{-s}] - \hat{\mathbf{Q}}^2_{-s})\mathbf{a}_{\mathbf{C}j}|}{\pi_j}\sum_{j=1}^n\mathbf{v}^\top\mathbf{a}_{\mathbf{C}j}\mathbf{a}_{\mathbf{C}j}^\top\mathbf{v}\right]$$

$$\leq \max_{1\leq j\leq n}\frac{37}{m}\mathbb{E}_{\zeta'}\left[\frac{|\mathbf{a}_{\mathbf{C}j}^\top(\mathbb{E}_{\zeta'}[\hat{\mathbf{Q}}^2_{-s}] - \hat{\mathbf{Q}}^2_{-s})\mathbf{a}_{\mathbf{C}j}|}{\pi_j}\right]$$

$$\leq \max_{1\leq j\leq n}37\sqrt{\frac{\mathbb{E}_{\zeta'}\left[(\mathbf{a}_{\mathbf{C}j}^\top(\mathbb{E}_{\zeta'}[\hat{\mathbf{Q}}^2_{-s}] - \hat{\mathbf{Q}}^2_{-s})\mathbf{a}_{\mathbf{C}j})^2\right]}{(m\pi_j)^2}}.$$

Observe that adapting the proof of Lemma D.1, we can readily ascertain the following:

$$\mathbb{E}_{\zeta'}\left[(\mathbf{a}_{\mathbf{C}j}^\top(\mathbb{E}_{\zeta'}[\hat{\mathbf{Q}}_{-s}^2] - \hat{\mathbf{Q}}_{-s}^2)\mathbf{a}_{\mathbf{C}j})^2\right] = \mathrm{Var}_{\zeta'}\left[\mathbf{a}_{\mathbf{C}j}^\top\hat{\mathbf{Q}}_{-s}^2\mathbf{a}_{\mathbf{C}j}\right] = O\left(\frac{(\mathbf{a}_{\mathbf{C}j}^\top\mathbf{a}_{\mathbf{C}j})^2\rho_{\max}d_{\mathrm{eff}}}{m}\right),$$

so that $T_{72} = O\left(\sqrt{\rho_{\max}^3 d_{\mathrm{eff}}^3/m^3}\right)$. Recalling (39) again, we derive

$$T_{73} \le 12\mathbb{E}_{\zeta'}\left[\left|\frac{\hat{\mathbf{x}}_s^\top(\mathbb{E}_\zeta[\hat{\mathbf{Q}}_{-s}^2] - \mathbb{E}_{\zeta'}[\hat{\mathbf{Q}}_{-s}^2])\hat{\mathbf{x}}_s\hat{\mathbf{x}}_s^\top\hat{\mathbf{x}}_s}{m}\right|\right] \le 12\sum_{j=1}^n\left|\frac{\mathbf{a}_{\mathbf{C}j}^\top(\mathbb{E}_\zeta[\hat{\mathbf{Q}}_{-s}^2] - \mathbb{E}_{\zeta'}[\hat{\mathbf{Q}}_{-s}^2])\mathbf{a}_{\mathbf{C}j}\mathbf{a}_{\mathbf{C}j}^\top\mathbf{a}_{\mathbf{C}j}}{m\pi_j}\right|$$

$$\le \frac{12\rho_{\max}d_{\mathrm{eff}}}{m}\sum_{j=1}^n\left|\mathbf{a}_{\mathbf{C}j}^\top(\mathbb{E}_\zeta[\hat{\mathbf{Q}}_{-s}^2] - \mathbb{E}_{\zeta'}[\hat{\mathbf{Q}}_{-s}^2])\mathbf{a}_{\mathbf{C}j}\right|.$$

Along with

$$\mathbb{E}_\zeta[\hat{\mathbf{Q}}_{-s}^2] - \mathbb{E}_{\zeta'}[\hat{\mathbf{Q}}_{-s}^2] = \frac{\delta_3}{1-\delta_3}\mathbb{E}_{\zeta'}[\hat{\mathbf{Q}}_{-s}^2] - \frac{\delta_3}{1-\delta_3}\mathbb{E}_{\zeta'}[\hat{\mathbf{Q}}_{-s}^2 \mid \mathbf{1}_{\neg\zeta_3}],$$

we obtain, for $\delta_3 \le m^{-3}$,

$$\left|\mathbf{a}_{\mathbf{C}j}^\top(\mathbb{E}_\zeta[\hat{\mathbf{Q}}_{-s}^2] - \mathbb{E}_{\zeta'}[\hat{\mathbf{Q}}_{-s}^2])\mathbf{a}_{\mathbf{C}j}\right| \le 2\delta_3\mathbf{a}_{\mathbf{C}j}^\top\mathbf{a}_{\mathbf{C}j}(\|\mathbb{E}_{\zeta'}[\hat{\mathbf{Q}}_{-s}^2]\| + \|\mathbb{E}_{\zeta'}[\hat{\mathbf{Q}}_{-s}^2 \mid \mathbf{1}_{\neg\zeta_3}]\|) \le \frac{144\mathbf{a}_{\mathbf{C}j}^\top\mathbf{a}_{\mathbf{C}j}}{m^3}.$$

This results in $T_{73} = O(\rho_{\max}d_{\mathrm{eff}}^2/m^4)$.

Now, it remains to bound $\|\mathbf{T}_8\|$. Noting (39) and $\frac{F_{i_s i_s}}{\check{\gamma}_s} \le 2$, it follows that

$$\|\mathbf{T}_8\| \le \sup_{\|\mathbf{u}\|=1,\|\mathbf{v}\|=1}\mathbb{E}_\zeta\left[\left|\frac{F_{i_s i_s}^2}{\check{\gamma}_s^2} - 1\right|\left|\frac{\mathbf{u}^\top\hat{\mathbf{x}}_s^\top\hat{\mathbf{Q}}_{-s}^2\hat{\mathbf{x}}_s\hat{\mathbf{Q}}_{-s}\hat{\mathbf{x}}_s\hat{\mathbf{x}}_s^\top\mathbf{v}}{m}\right|\right]$$

$$\le \sup_{\|\mathbf{u}\|=1,\|\mathbf{v}\|=1}\mathbb{E}_\zeta\left[\left|\frac{F_{i_s i_s}}{\check{\gamma}_s}\left(\frac{F_{i_s i_s}}{\check{\gamma}_s} - 1\right)\right|\left|\frac{\mathbf{u}^\top\hat{\mathbf{x}}_s^\top\hat{\mathbf{Q}}_{-s}^2\hat{\mathbf{x}}_s\hat{\mathbf{Q}}_{-s}\hat{\mathbf{x}}_s\hat{\mathbf{x}}_s^\top\mathbf{v}}{m}\right|\right]$$

$$+ \sup_{\|\mathbf{u}\|=1,\|\mathbf{v}\|=1}\mathbb{E}_\zeta\left[\left|\frac{F_{i_s i_s}}{\check{\gamma}_s} - 1\right|\left|\frac{\mathbf{u}^\top\hat{\mathbf{x}}_s^\top\hat{\mathbf{Q}}_{-s}^2\hat{\mathbf{x}}_s\hat{\mathbf{Q}}_{-s}\hat{\mathbf{x}}_s\hat{\mathbf{x}}_s^\top\mathbf{v}}{m}\right|\right]$$

$$\le 3\sup_{\|\mathbf{u}\|=1,\|\mathbf{v}\|=1}\mathbb{E}_\zeta\left[\left|\frac{F_{i_s i_s}}{\check{\gamma}_s} - 1\right|\left|\frac{\mathbf{u}^\top\hat{\mathbf{x}}_s^\top\hat{\mathbf{Q}}_{-s}^2\hat{\mathbf{x}}_s\hat{\mathbf{Q}}_{-s}\hat{\mathbf{x}}_s\hat{\mathbf{x}}_s^\top\mathbf{v}}{m}\right|\right]$$

$$\le \frac{108\rho_{\max}d_{\mathrm{eff}}}{m}\sup_{\|\mathbf{u}\|=1,\|\mathbf{v}\|=1}\mathbb{E}_\zeta\left[\left|\frac{F_{i_s i_s}}{\check{\gamma}_s} - 1\right|\left|\mathbf{u}^\top\hat{\mathbf{Q}}_{-s}\hat{\mathbf{x}}_s\hat{\mathbf{x}}_s^\top\mathbf{v}\right|\right].$$

Further, applying the techniques of bounding $\|\mathbf{H}^{1/2}\check{\mathbf{Z}}_3\mathbf{H}^{-1/2}\|$ in Appendix E.2 again, we get $\|\mathbf{T}_8\| = O\left(\sqrt{\rho_{\max}^5 d_{\mathrm{eff}}^5/m^5}\right)$. Putting the above together, we conclude that

$$\|\mathbb{E}_\zeta[\hat{\mathbf{Q}}^2] - (\mathbf{I}_d + \mathbf{A}_{\mathbf{C}}^\top\bar{\mathbf{F}}\mathbf{A}_{\mathbf{C}})\| = O\left(\sqrt{\frac{\rho_{\max}^3 d_{\mathrm{eff}}^3}{m^3}}\right).$$

This thus concludes the proof of Proposition F.1. □

Based on the second inverse moment result in Proposition F.1, we follow the line of arguments in Corollary 3.4 and Corollary 3.7,to derive the following fine-grained second inverse moment results for exact and/or approximate leverage score sampling and SRHT using scalar debiasing. The proofs of these results follow from a direct combination of the proofs of Corollary 3.4, Corollary 3.7 and Proposition F.1, and are omitted here.

**Corollary F.2** (Second inverse moment using scalar debiasing under approximate leverage)**.** *Under the settings and notations of Proposition F.1, for any random sampling scheme with sampling distribution $\pi_i \in [\ell_i^{\mathbf{C}}/(d_{\text{eff}}\rho_{\max}), \ell_i^{\mathbf{C}}/(d_{\text{eff}}\rho_{\min})]$ with some $\rho_{\min} \in [1/2, 1]$ as in Definition 2.3 and diagonal matrix $\bar{\mathbf{F}} = \text{diag}\{\bar{F}_{ii}\}_{i=1}^n$ with*

$$\bar{F}_{ii} = \frac{\mathbf{a}_{\mathbf{C}i}{}^\top \mathbb{E}_\zeta[(\frac{m}{m-d_{\text{eff}}}\mathbf{A}_{\mathbf{C}}^\top \mathbf{S}^\top \mathbf{S}\mathbf{A}_{\mathbf{C}} + \mathbf{C}_{\mathbf{A}})^{-2}]\mathbf{a}_{\mathbf{C}i}}{m\pi_i}.$$

*Then, there exists universal constant $C > 0$ independent of $n, d_{\text{eff}}$ such that for $m \geq C\rho_{\max}d_{\text{eff}}\log(d_{\text{eff}}/\delta)$ with $\delta \leq m^{-3}$, $(\frac{m}{m-d_{\text{eff}}}\mathbf{A}_{\mathbf{C}}^\top\mathbf{S}^\top\mathbf{S}\mathbf{A}_{\mathbf{C}} + \mathbf{C}_{\mathbf{A}})^{-2}$ is an $(\epsilon, \delta)$-unbiased estimator of $\mathbf{I}_d + \mathbf{A}_{\mathbf{C}}{}^\top\bar{\mathbf{F}}\mathbf{A}_{\mathbf{C}}$ with inversion bias $\epsilon = \max\left\{O\left(\sqrt{\rho_{\max}^3 d_{\text{eff}}^3/m^3}\right), O(\epsilon_\rho\rho_{\max}d_{\text{eff}}/m)\right\}$ for $\epsilon_\rho = \max\{\rho_{\min}^{-1} - 1, 1 - \rho_{\max}^{-1}\}$.*

**Corollary F.3** (Second moment for SRHT using scalar debiasing )**.** *Under the settings and notations of Proposition F.1, for $\tilde{\mathbf{A}}_{\text{SRHT}} \in \mathbb{R}^{m \times n}$ the SRHT of $\mathbf{A}$ as in Definition 3.6, and diagonal matrix $\bar{\mathbf{F}} = \text{diag}\{\bar{F}_{ii}\}_{i=1}^n$ with*

$$\bar{F}_{ii} = \frac{\mathbf{e}_i^\top \mathbf{H}_n \mathbf{D}_n \mathbf{A}_{\mathbf{C}} \mathbb{E}_\zeta[(\frac{m}{m-d_{\text{eff}}}\mathbf{H}^{-1/2}\tilde{\mathbf{A}}_{\text{SRHT}}^\top \tilde{\mathbf{A}}_{\text{SRHT}}\mathbf{H}^{-1/2} + \mathbf{C}_{\mathbf{A}})^{-2}]\mathbf{A}_{\mathbf{C}}^\top\mathbf{D}_n\mathbf{H}_n\mathbf{e}_i}{nm\pi_i},$$

*then there exists universal constant $C > 0$ independent of $n, d_{\text{eff}}$ and $n\exp(-d_{\text{eff}}) < \delta < m^{-3}$ such that for $m \geq C\rho_{\max}d_{\text{eff}}\log(d_{\text{eff}}/\delta)$ with max factor $\rho_{\max} = \max_{1 \leq i \leq n} n\ell_i^{\mathbf{C}}(\mathbf{H}_n\mathbf{D}_n\mathbf{A}/\sqrt{n})/d_{\text{eff}}$, $(\frac{m}{m-d_{\text{eff}}}\mathbf{H}^{-1/2}\tilde{\mathbf{A}}_{\text{SRHT}}^\top\tilde{\mathbf{A}}_{\text{SRHT}}\mathbf{H}^{-1/2} + \mathbf{C}_{\mathbf{A}})^{-2}$ is an $(\epsilon, \delta)$-unbiased estimator of $\mathbf{I}_d + \frac{1}{n}\mathbf{A}_{\mathbf{C}}^\top\mathbf{D}_n\mathbf{H}_n\bar{\mathbf{F}}\mathbf{H}_n\mathbf{D}_n\mathbf{A}_{\mathbf{C}}$ with inversion bias $\epsilon = O\left(\sqrt{\rho_{\max}^3 d_{\text{eff}}^3/m^3} + \sqrt{d_{\text{eff}}\log(n/\delta)/m}\right)$.*

These results are extension of the results in Corollaries in 3.4 and 3.7, and can be used to derive SSN local convergence rates under exact/approximate leverage and SRHT similar to Theorem 4.3. See Corollaries F.5 and F.6 in Appendix F.3.

## F.2. Proof of Theorem 4.3

Here, we prove Theorem 4.3 by adapting the proof approaches of our Propositions 3.2 and F.1 and Dereziński et al. (2021a, Theorem 10). The proof of Theorem 4.3 comes in the following two steps:

- we first establish Lemma F.4 that connects the de-biased SSN iterations $\check{\boldsymbol{\beta}}_{t+1}$ to the true $\boldsymbol{\beta}_{t+1}$; and

- perform detailed convergence analysis of de-biased SSN.

To begin, we define a high probability event $\zeta_t$ for each $t = 0, \ldots, T-1$, as defined in (19). These events are established independently for each iteration, and we denote $\zeta = \bigcap_{t=0}^{T-1} \zeta_t$. Under the setting of Theorem 4.3, it follows from Lemma 2.7 that $\Pr(\zeta_t) \geq 1 - \delta/T$ and $\Pr(\zeta) \geq 1 - \delta$. These events will be constantly exploited in the remainder of the proof of Theorem 4.3.

Proceeding to the core of our proof, we present an auxiliary lemma, which uses Propositions 3.2 and F.1 to connect the de-biased iteration $\check{\boldsymbol{\beta}}_{t+1}$ to the true Newton iteration $\boldsymbol{\beta}_{t+1}$. This is pivotal to prove Theorem 4.3. Note that this result is universally applicable to any de-biased $\check{\boldsymbol{\beta}}_t$ and is independent of the smoothness assumption on $F$ in, e.g., Assumption 4.2.

**Lemma F.4.** *Let $\mathbf{H}_t = \nabla^2 f(\check{\boldsymbol{\beta}}_t) + \mathbf{C}(\check{\boldsymbol{\beta}}_t)$ with $\nabla^2 f(\check{\boldsymbol{\beta}}_t) = \mathbf{A}(\check{\boldsymbol{\beta}}_t)^\top\mathbf{A}(\check{\boldsymbol{\beta}}_t)$ and $\mathbf{C}(\check{\boldsymbol{\beta}}_t) = \nabla^2\Phi(\check{\boldsymbol{\beta}}_t)$ a p.s.d. matrix, and let $\check{\boldsymbol{\beta}}_{t+1}$ be the de-biased SSN iteration as in (10) with de-biased $\check{\mathbf{S}}_t$ as in Proposition 3.2. Denote $\Delta_t = \boldsymbol{\beta}_t - \boldsymbol{\beta}^*$ and $\check{\Delta}_t = \check{\boldsymbol{\beta}}_t - \boldsymbol{\beta}^*$. If the exact Newton step $\boldsymbol{\beta}_{t+1} = \check{\boldsymbol{\beta}}_t - \mu_t\mathbf{H}_t^{-1}\mathbf{g}_t$ (with $\mathbf{g}_t$ the gradient of $F$ at $\check{\boldsymbol{\beta}}_t$) is a descent direction, i.e., $\|\Delta_{t+1}\|_{\mathbf{H}_t} \leq \|\check{\Delta}_t\|_{\mathbf{H}_t}$, then, there exists universal constant $C > 0$ independent of $n, d_{\text{eff}}$ such that for $m \geq C\rho_{\max}d_{\text{eff}}(\log(d_{\text{eff}}T/\delta) + 1/\epsilon^{2/3})$ with $\epsilon > 0$, $\delta \leq m^{-3}$, we have*

$$\mathbb{E}_{\zeta_t}[\|\check{\Delta}_{t+1}\|_{\mathbf{H}_t}^2] \leq \|\Delta_{t+1}\|_{\mathbf{H}_t}^2 + \epsilon\|\check{\Delta}_t\|_{\mathbf{H}_t}^2 + \frac{36\rho_{\max}d_{\text{eff}}}{m}\|\boldsymbol{\beta}_{t+1} - \check{\boldsymbol{\beta}}_t\|_{\nabla^2 f(\check{\boldsymbol{\beta}}_t)}^2.$$

*Here, $\rho_{\max}$ is the max importance sampling approximation factor in Definition 2.3 for $\ell_i^{\mathbf{C}} = \max_{1 \leq t \leq T} \ell_i^{\mathbf{C}}(\check{\boldsymbol{\beta}}_t)$ and $d_{\text{eff}} = \max_{1 \leq t \leq T} d_{\text{eff}}(\check{\boldsymbol{\beta}}_t)$ with $\ell_i^{\mathbf{C}}(\check{\boldsymbol{\beta}}_t)$ and $d_{\text{eff}}(\check{\boldsymbol{\beta}}_t)$ the leverage scores and effective dimension of $\mathbf{A}(\check{\boldsymbol{\beta}}_t)$ given $\mathbf{C}(\check{\boldsymbol{\beta}}_t)$, respectively.*

*Proof of Lemma F.4.* In both this proof and the subsequent proof of Theorem 4.3, to simplify the notation, we abbreviate $\mathbf{A_C}(\check{\boldsymbol{\beta}}_t)$ as $\mathbf{A_C}$. We first recall some notations from (the proof of) Proposition F.1. Let $\mathbf{p}_t = \boldsymbol{\beta}_{t+1} - \check{\boldsymbol{\beta}}_t = -\mu_t \mathbf{H}_t^{-1}\mathbf{g}_t$, and $\hat{\mathbf{Q}}_t = (\mathbf{A_C}^\top \check{\mathbf{S}}_t^\top \check{\mathbf{S}}_t \mathbf{A_C} + \mathbf{C_A})^{-1}$ with the de-biased sampling matrix $\check{\mathbf{S}}_t = \mathrm{diag}\left\{\sqrt{m/(m - \ell_{i_s}^{\mathbf{C}}(\check{\boldsymbol{\beta}}_t)/\pi_{i_s})}\right\}_{s=1}^m \cdot \mathbf{S}_t$ as in Proposition 3.2 and $\mathbf{C_A} = \mathbf{H}_t^{-1/2}\mathbf{C}(\check{\boldsymbol{\beta}}_t)\mathbf{H}_t^{-1/2}$. Define the diagonal matrix $\bar{\mathbf{F}} = \mathrm{diag}\{\bar{F}_{ii}\}_{i=1}^n$ with $\bar{F}_{ii} = \mathbf{a_C}_i^\top \mathbb{E}_{\zeta_t}[(\mathbf{A_C}^\top \check{\mathbf{S}}^\top \check{\mathbf{S}} \mathbf{A_C} + \mathbf{C_A})^{-2}]\mathbf{a_C}_i/m\pi_i$ as in Proposition F.1 and $\mathbf{a_C}_i^\top \in \mathbb{R}^d$ the $i^{th}$ row of $\mathbf{A_C}$.

Building on the results from Proposition 3.2 and Proposition F.1, for $\epsilon = O\left(\sqrt{\rho_{\max}^3 d_{\mathrm{eff}}^3/m^3}\right)$, we obtain

$$
\begin{aligned}
\mathbb{E}_{\zeta_t}[\|\check{\Delta}_{t+1}\|_{\mathbf{H}_t}^2] - \|\Delta_{t+1}\|_{\mathbf{H}_t}^2 &= \mathbb{E}_{\zeta_t}[\|\check{\boldsymbol{\beta}}_{t+1} - \boldsymbol{\beta}_{t+1}\|_{\mathbf{H}_t}^2] + 2\Delta_{t+1}^\top \mathbf{H}_t \mathbb{E}_{\zeta_t}[\check{\boldsymbol{\beta}}_{t+1} - \boldsymbol{\beta}_{t+1}] \\
&= 2\mu_t \Delta_{t+1}^\top \mathbf{H}_t^{\frac{1}{2}} \mathbb{E}_{\zeta_t}[(\hat{\mathbf{Q}}_t - \mathbf{I}_d)]\mathbf{H}_t^{\frac{1}{2}} \mathbf{H}_t^{-1}\mathbf{g}_t + \mu_t^2 \mathbf{g}_t^\top \mathbf{H}_t^{-1}\mathbf{H}_t^{\frac{1}{2}}\mathbb{E}_{\zeta_t}[(\hat{\mathbf{Q}}_t - \mathbf{I}_d)^2]\mathbf{H}_t^{\frac{1}{2}}\mathbf{H}_t^{-1}\mathbf{g}_t \\
&\leq 2\|\Delta_{t+1}\|_{\mathbf{H}_t}\|\mathbf{p}_t\|_{\mathbf{H}_t}\|\mathbb{E}_{\zeta_t}[\hat{\mathbf{Q}}_t] - \mathbf{I}_d\| + \mathbf{p}_t^\top \mathbf{H}_t^{\frac{1}{2}}\mathbb{E}_{\zeta_t}[\hat{\mathbf{Q}}_t^2 - \mathbf{I}_d - 2(\hat{\mathbf{Q}}_t - \mathbf{I}_d)]\mathbf{H}_t^{\frac{1}{2}}\mathbf{p}_t \\
&\leq 2\|\Delta_{t+1}\|_{\mathbf{H}_t}\|\mathbf{p}_t\|_{\mathbf{H}_t}\|\mathbb{E}_{\zeta_t}[\hat{\mathbf{Q}}_t] - \mathbf{I}_d\| + 2\|\mathbf{p}_t\|_{\mathbf{H}_t}^2\|\mathbb{E}_{\zeta_t}[\hat{\mathbf{Q}}_t] - \mathbf{I}_d\| + \mathbf{p}_t^\top \mathbf{H}_t^{\frac{1}{2}}\mathbb{E}_{\zeta_t}[\hat{\mathbf{Q}}_t^2 - \mathbf{I}_d - \mathbf{A_C}^\top \bar{\mathbf{F}}_t \mathbf{A_C}]\mathbf{H}_t^{\frac{1}{2}}\mathbf{p}_t \\
&+ \mathbf{p}_t^\top \mathbf{H}_t^{\frac{1}{2}}\mathbf{A_C}^\top \bar{\mathbf{F}}_t \mathbf{A_C}\mathbf{H}_t^{\frac{1}{2}}\mathbf{p}_t \\
&\leq 2\epsilon\|\Delta_{t+1}\|_{\mathbf{H}_t}\|\mathbf{p}_t\|_{\mathbf{H}_t} + 2\epsilon\|\mathbf{p}_t\|_{\mathbf{H}_t}^2 + \|\mathbf{p}_t\|_{\mathbf{H}_t}^2\|\mathbb{E}_{\zeta_t}[\hat{\mathbf{Q}}_t^2] - \mathbf{I}_d - \mathbf{A_C}^\top \bar{\mathbf{F}}_t \mathbf{A_C}\| + \|\mathbf{p}_t\|_{\nabla^2 f(\check{\boldsymbol{\beta}}_t)}^2\|\bar{\mathbf{F}}_t\| \\
&\leq \epsilon\|\Delta_{t+1}\|_{\mathbf{H}_t}\|\mathbf{p}_t\|_{\mathbf{H}_t} + \epsilon\|\mathbf{p}_t\|_{\mathbf{H}_t}^2 + \|\mathbf{p}_t\|_{\nabla^2 f(\check{\boldsymbol{\beta}}_t)}^2 \max_{1 \leq i \leq n} \frac{\mathbf{a_C}_i^\top \mathbb{E}_{\zeta_t}[\hat{\mathbf{Q}}_t^2]\mathbf{a_C}_i}{m\pi_i} \\
&\leq \epsilon\|\check{\Delta}_t\|_{\mathbf{H}_t}\|\mathbf{p}_t\|_{\mathbf{H}_t} + \epsilon\|\mathbf{p}_t\|_{\mathbf{H}_t}^2 + \frac{36\rho_{\max}d_{\mathrm{eff}}}{m}\|\mathbf{p}_t\|_{\nabla^2 f(\check{\boldsymbol{\beta}}_t)}^2,
\end{aligned}
$$

which combined with $\|\mathbf{p}_t\|_{\mathbf{H}_t} \leq \|\Delta_{t+1}\|_{\mathbf{H}_t} + \|\check{\Delta}_t\|_{\mathbf{H}_t} \leq 2\|\check{\Delta}_t\|_{\mathbf{H}_t}$ and $\|\check{\Delta}_t\|_{\mathbf{H}_t}\|\mathbf{p}_t\|_{\mathbf{H}_t} \leq 2\|\check{\Delta}_t\|_{\mathbf{H}_t}^2$ leads to

$$
\begin{aligned}
\mathbb{E}_{\zeta_t}[\|\check{\Delta}_{t+1}\|_{\mathbf{H}_t}^2] - \|\Delta_{t+1}\|_{\mathbf{H}_t}^2 &\leq 2\epsilon\|\check{\Delta}_t\|_{\mathbf{H}_t}^2 + 4\epsilon\|\check{\Delta}_t\|_{\mathbf{H}_t}^2 + \frac{36\rho_{\max}d_{\mathrm{eff}}}{m}\|\mathbf{p}_t\|_{\nabla^2 f(\check{\boldsymbol{\beta}}_t)}^2 \\
&= \epsilon\|\check{\Delta}_t\|_{\mathbf{H}_t}^2 + \frac{36\rho_{\max}d_{\mathrm{eff}}}{m}\|\mathbf{p}_t\|_{\nabla^2 f(\check{\boldsymbol{\beta}}_t)}^2.
\end{aligned}
$$

This concludes the proof of Lemma F.4. $\qquad\square$

**Proof of Theorem 4.3.** First note that by Assumption 4.2, we have the following result

$$
\|\mathbf{H}_t\check{\Delta}_t - \mathbf{g}_t\|_{\mathbf{H}_t^{-1}} \leq \upsilon\|\check{\Delta}_t\|_{\mathbf{H}_t}, \quad \mathbf{H}_t \approx_\upsilon \mathbf{H}, \text{ for } \upsilon = O\left(\sqrt{\frac{\rho_{\max}^3 d_{\mathrm{eff}}^3}{m^3}}\right). \tag{41}
$$

Letting $\check{\Delta}_t = \check{\boldsymbol{\beta}}_t - \boldsymbol{\beta}^*$ and $\Delta_t = \boldsymbol{\beta}_t - \boldsymbol{\beta}^*$, we then show (41) holds under Assumption 4.2. Consider $\check{\boldsymbol{\beta}}_t \in U$ for all $t$, we get the following condition:

$$
\|\check{\Delta}_t\|_{\mathbf{H}} < \left(\frac{\rho_{\max}d_{\mathrm{eff}}\sigma_{\min}}{m}\right)^{3/2}/L,
$$

where $\sigma_{\min}$ denotes the smallest eigenvalue of $\mathbf{H}$. We further derive

$$
\|\mathbf{H}^{-\frac{1}{2}}(\mathbf{H}_t - \mathbf{H})\mathbf{H}^{-\frac{1}{2}}\| \leq \frac{1}{\sigma_{\min}}\|\mathbf{H}_t - \mathbf{H}\| \leq \frac{L}{\sigma_{\min}}\|\check{\boldsymbol{\beta}}_t - \boldsymbol{\beta}^*\| = \frac{L}{\sigma_{\min}}\|\check{\Delta}_t\| \leq \frac{L}{\sigma_{\min}^{3/2}}\|\check{\Delta}_t\|_{\mathbf{H}} \leq \sqrt{\frac{\rho_{\max}^3 d_{\mathrm{eff}}^3}{m^3}} = \upsilon,
$$

which leads to that $\mathbf{H}_t \approx_\upsilon \mathbf{H}$. We now advance to verify the second part of (41). Recalling that the standard analysis of Newton's method (Boyd & Vandenberghe, 2004), we have

$$
\begin{aligned}
\|\mathbf{H}_t\check{\Delta}_t - \mathbf{g}_t\| &= \left\|\mathbf{H}_t\check{\Delta}_t - \left(\int_0^1 \nabla^2 F(\boldsymbol{\beta}^* + \tau\check{\Delta}_t)d\tau\right)\check{\Delta}_t\right\| \leq \|\check{\Delta}_t\|\int_0^1 \|\nabla^2 F(\check{\boldsymbol{\beta}}_t) - \nabla^2 F(\boldsymbol{\beta}^* + \tau\check{\Delta}_t)\|d\tau \\
&\leq \|\check{\Delta}_t\|\int_0^1 (1 - \tau)L\|\check{\Delta}_t\|d\tau = \frac{L}{2}\|\check{\Delta}_t\|^2,
\end{aligned}
$$

which together with $\upsilon < 1/2$ and the fact that $\mathbf{H}_t \approx_\upsilon \mathbf{H}$ implies that $\|\mathbf{H}_t^{-1}\| \leq \frac{1+\upsilon}{\sigma_{\min}}$ achieves

$$\|\mathbf{H}_t \check{\Delta}_t - \mathbf{g}_t\|_{\mathbf{H}_t^{-1}} \leq \sqrt{\frac{1+\upsilon}{\sigma_{\min}}}\|\mathbf{H}_t \check{\Delta}_t - \mathbf{g}_t\| \leq \sqrt{\frac{L^2(1+\upsilon)}{4\sigma_{\min}}}\|\check{\Delta}_t\|^2 \leq \sqrt{\frac{L^2(1+\upsilon)}{4\sigma_{\min}^3}}\|\check{\Delta}_t\|_{\mathbf{H}}^2 \leq \frac{\upsilon\sqrt{1+\upsilon}}{2}\|\check{\Delta}_t\|_{\mathbf{H}}$$

$$\leq \frac{\upsilon(1+\upsilon)^{3/2}}{2}\|\check{\Delta}_t\|_{\mathbf{H}_t} \leq \upsilon\|\check{\Delta}_t\|_{\mathbf{H}_t}.$$

Thus, (41) holds.

Next, we proceed to the main part of the proof. We first rewrite $\|\Delta_{t+1}\|_{\mathbf{H}_t}^2$ as

$$\begin{aligned}
\|\Delta_{t+1}\|_{\mathbf{H}_t}^2 &= \Delta_{t+1}^\top \mathbf{H}_t(\boldsymbol{\beta}_{t+1} - \check{\boldsymbol{\beta}}_t + \check{\boldsymbol{\beta}}_t - \boldsymbol{\beta}^*) = \Delta_{t+1}^\top \mathbf{H}_t \check{\Delta}_t - \mu_t \Delta_{t+1}^\top \mathbf{g}_t \\
&= (1-\mu_t)\Delta_{t+1}^\top \mathbf{g}_t + \Delta_{t+1}^\top(\mathbf{H}_t \check{\Delta}_t - \mathbf{g}_t) \\
&= (1-\mu_t)\Delta_{t+1}^\top \mathbf{H}_t \check{\Delta}_t - (1-\mu_t)\Delta_{t+1}^\top(\mathbf{H}_t \check{\Delta}_t - \mathbf{g}_t) + \Delta_{t+1}^\top(\mathbf{H}_t \check{\Delta}_t - \mathbf{g}_t) \\
&= (1-\mu_t)(\check{\Delta}_t^\top \mathbf{H}_t \check{\Delta}_t - \mu_t \mathbf{g}_t^\top \check{\Delta}_t) + \mu_t \Delta_{t+1}^\top(\mathbf{H}_t \check{\Delta}_t - \mathbf{g}_t) \\
&= (1-\mu_t)^2 \|\check{\Delta}_t\|_{\mathbf{H}_t}^2 + \mu_t(\Delta_{t+1} + (1-\mu_t)\check{\Delta}_t)^\top(\mathbf{H}_t \check{\Delta}_t - \mathbf{g}_t).
\end{aligned}$$

Invoking (41) and triangle inequality, we get

$$\|\Delta_{t+1}\|_{\mathbf{H}_t}^2 \leq (1-\mu_t)^2 \|\check{\Delta}_t\|_{\mathbf{H}_t}^2 + \upsilon\mu_t\|\Delta_{t+1}\|_{\mathbf{H}_t}\|\check{\Delta}_t\|_{\mathbf{H}_t} + \upsilon\mu_t(1-\mu_t)\|\check{\Delta}_t\|_{\mathbf{H}_t}^2.$$

With the fact that if $x^2 \leq ax + b$ then $x^2 \leq a^2 + 2b$, it follows that:

$$\begin{aligned}
\|\Delta_{t+1}\|_{\mathbf{H}_t}^2 &\leq \upsilon^2\mu_t^2\|\check{\Delta}_t\|_{\mathbf{H}_t}^2 + 2(1-\mu_t)^2\|\check{\Delta}_t\|_{\mathbf{H}_t}^2 + 2\upsilon\mu_t(1-\mu_t)\|\check{\Delta}_t\|_{\mathbf{H}_t}^2 \\
&= (2(1-\mu_t)^2 + 2\upsilon\mu_t + \mu_t^2(\upsilon^2 - 2\upsilon))\|\check{\Delta}_t\|_{\mathbf{H}_t}^2 \\
&\leq 2((1-\mu_t)^2 + \upsilon\mu_t)\|\check{\Delta}_t\|_{\mathbf{H}_t}^2.
\end{aligned}$$

It then follows from Lemma F.4 and the inequality $\nabla^2 f(\check{\boldsymbol{\beta}}_t) \preceq \mathbf{H}_t$ that

$$\mathbb{E}_{\zeta_t}[\|\check{\Delta}_{t+1}\|_{\mathbf{H}_t}^2] - \|\Delta_{t+1}\|_{\mathbf{H}_t}^2 \leq \upsilon\|\check{\Delta}_t\|_{\mathbf{H}_t}^2 + \frac{36\rho_{\max}d_{\mathrm{eff}}}{m}\|\mathbf{p}_t\|_{\mathbf{H}_t}^2. \tag{42}$$

For the second term in (42), we rewrite

$$\begin{aligned}
\frac{36\rho_{\max}d_{\mathrm{eff}}}{m}\|\mathbf{p}_t\|_{\mathbf{H}_t}^2 &= \frac{36\rho_{\max}d_{\mathrm{eff}}}{m}\mu_t^2(\mathbf{g}_t^\top \check{\Delta}_t - \mathbf{g}_t^\top \mathbf{H}_t^{-1}(\mathbf{H}_t \check{\Delta}_t - \mathbf{g}_t)) \\
&= \frac{36\rho_{\max}d_{\mathrm{eff}}}{m}\mu_t^2(\check{\Delta}_t^\top \mathbf{H}_t \check{\Delta}_t - \check{\Delta}_t^\top(\mathbf{H}_t \check{\Delta}_t - \mathbf{g}_t) - \mathbf{g}_t^\top \mathbf{H}_t^{-1}(\mathbf{H}_t \check{\Delta}_t - \mathbf{g}_t)) \\
&= \frac{36\rho_{\max}d_{\mathrm{eff}}}{m}\mu_t^2\|\check{\Delta}_t\|_{\mathbf{H}_t}^2 - \frac{36\rho_{\max}d_{\mathrm{eff}}}{m}\mu_t^2(\check{\Delta}_t + \mathbf{H}_t^{-1}\mathbf{g}_t)^\top(\mathbf{H}_t \check{\Delta}_t - \mathbf{g}_t)).
\end{aligned}$$

Using again (41), we further get

$$\begin{aligned}
-\frac{36\rho_{\max}d_{\mathrm{eff}}}{m}\mu_t^2(\check{\Delta}_t + \mathbf{H}_t^{-1}\mathbf{g}_t)^\top(\mathbf{H}_t \check{\Delta}_t - \mathbf{g}_t) &= \frac{36\rho_{\max}d_{\mathrm{eff}}}{m}(-\mu_t^2\check{\Delta}_t + \mu_t(\Delta_{t+1} - \check{\Delta}_t))^\top(\mathbf{H}_t \check{\Delta}_t - \mathbf{g}_t) \\
&\leq \frac{36\rho_{\max}d_{\mathrm{eff}}\upsilon}{m}\mu_t(\mu_t + 2)\|\check{\Delta}_t\|_{\mathbf{H}_t}^2.
\end{aligned}$$

Then, putting the above together, we have

$$\begin{aligned}
\mathbb{E}_{\zeta_t}[\|\check{\Delta}_{t+1}\|_{\mathbf{H}_t}^2] &\leq \left(\left(2(1-\mu_t)^2 + 2\upsilon\mu_t + \upsilon + \frac{36\rho_{\max}d_{\mathrm{eff}}}{m}\mu_t^2 + \frac{36\rho_{\max}d_{\mathrm{eff}}\upsilon}{m}\mu_t(\mu_t + 2)\right)\|\check{\Delta}_t\|_{\mathbf{H}_t}^2 \\
&= \left(\left(2(1-\mu_t)^2 + \frac{36\rho_{\max}d_{\mathrm{eff}}}{m}\mu_t^2\right) + \upsilon\left(2\mu_t + 1 + \frac{36\rho_{\max}d_{\mathrm{eff}}}{m}\mu_t(\mu_t + 2)\right)\right)\|\check{\Delta}_t\|_{\mathbf{H}_t}^2,
\end{aligned}$$

which along with $\upsilon = O\left(\sqrt{\rho_{\max}^3 d_{\mathrm{eff}}^3/m^3}\right)$ and $\mu_t = \frac{1}{1+\rho_{\max}d_{\mathrm{eff}}/m} < 1$ results in

$$\mathbb{E}_{\zeta_t}[\|\check{\Delta}_{t+1}\|_{\mathbf{H}_t}^2] \leq \left(\left(2(1-\mu_t)^2 + \frac{36\rho_{\max}d_{\mathrm{eff}}}{m}\mu_t^2\right) + \upsilon\right)\|\check{\Delta}_t\|_{\mathbf{H}_t}^2 \leq \left(\frac{37\rho_{\max}d_{\mathrm{eff}}}{m} + \upsilon\right)\|\check{\Delta}_t\|_{\mathbf{H}_t}^2.$$

Applying $\upsilon < \frac{1}{2}$ and the fact that $\mathbf{H}_t \approx_\upsilon \mathbf{H}$ indicates $\|\mathbf{v}\|_{\mathbf{H}_t}^2 \approx_\upsilon \|\mathbf{v}\|_{\mathbf{H}}^2$, we get the following bound:

$$\mathbb{E}_{\zeta_t}[\|\check{\Delta}_{t+1}\|_{\mathbf{H}}^2] \leq (1+\upsilon)\mathbb{E}_{\zeta_t}[\|\check{\Delta}_{t+1}\|_{\mathbf{H}_t}^2] \leq (1+\upsilon)\left(\frac{37\rho_{\max}d_{\mathrm{eff}}}{m} + \upsilon\right)\|\check{\Delta}_t\|_{\mathbf{H}_t}^2$$

$$\leq (1+\upsilon)^2\left(\frac{37\rho_{\max}d_{\mathrm{eff}}}{m} + \upsilon\right)\|\check{\Delta}_t\|_{\mathbf{H}}^2 = \left(\frac{\rho_{\max}d_{\mathrm{eff}}}{m} + \upsilon\right)\|\check{\Delta}_t\|_{\mathbf{H}}^2, \tag{43}$$

where the constant "$(1+\upsilon)^2 \cdot 37$" is absorbed into $m$.

Now, it remains to check that $\check{\beta}_t \in U$ for all $t$ when conditioned on the event $\zeta = \bigcap_{t=0}^{T-1}\zeta_t$. Assuming that this holds for $t = 0$, then it remains to prove that, conditioned on the event $\zeta$, $\|\check{\Delta}_{t+1}\|_{\mathbf{H}} \leq \|\check{\Delta}_t\|_{\mathbf{H}}$ holds for each $t$ almost surely. Following the fact that Proposition 3.2 implies that, conditioned on $\zeta_t$, we obtain

$$\|\hat{\mathbf{Q}}_t - \mathbf{I}_d\| \leq \varepsilon,$$

where Lemma 2.7 guarantees that $\varepsilon$ is small. Adapting the techniques used in the proof of Lemma F.4 and by the analysis of the exact Newton step $\Delta_{t+1}$, we arrive at

$$\|\check{\Delta}_{t+1}\|_{\mathbf{H}_t}^2 \leq \|\Delta_{t+1}\|_{\mathbf{H}_t}^2 + 2\|\Delta_{t+1}\|_{\mathbf{H}_t}\|\mathbf{p}_t\|_{\mathbf{H}_t}\|\hat{\mathbf{Q}}_t - \mathbf{I}_d\| + \mathbf{p}_t^\top\mathbf{H}_t^{\frac{1}{2}}(\hat{\mathbf{Q}}_t - \mathbf{I}_d)^2\mathbf{H}_t^{\frac{1}{2}}\mathbf{p}_t$$

$$\leq \|\Delta_{t+1}\|_{\mathbf{H}_t}^2 + 8\|\check{\Delta}_t\|_{\mathbf{H}_t}^2\|\hat{\mathbf{Q}}_t - \mathbf{I}_d\| \leq \left(\varepsilon + \frac{\rho_{\max}d_{\mathrm{eff}}}{m} + \upsilon\right)\|\check{\Delta}_t\|_{\mathbf{H}_t}^2.$$

From Lemma 2.7, it is deduced that $\varepsilon + \frac{\rho_{\max}d_{\mathrm{eff}}}{m} + \upsilon$ is sufficiently small. Using $\mathbf{H}_t \approx_\upsilon \mathbf{H}$ again, we deduce $\|\check{\Delta}_{t+1}\|_{\mathbf{H}}^2 \leq \|\check{\Delta}_t\|_{\mathbf{H}}^2$. Consequently, we infer that every iterate lies within $U$, and the result (43) holds for $t = 0, 1, \ldots, T-1$. Putting the above together, we conclude that, conditioned on the event $\zeta$ that holds with probability at least $1 - \delta$, the union bound (11) is achieved, thereby completing the proof. $\square$

### F.3. Additional Results on SSN and Further Discussions

Based on the above analysis, for any sampling method whose sampling probabilities are close to the exact approximate leverage scores, we then show that using scalar debiasing $m/(m - d_{\mathrm{eff}})$ yields a slightly weaker local convergence result for SSN than that in Theorem 4.3, yet achieves enhanced computational efficiency. The result naturally follows by recalling Corollary 3.4 and Corollary F.2, and adapting the proof of Theorem 4.3 with a neighborhood $U = \{\boldsymbol{\beta}\colon \|\boldsymbol{\beta} - \boldsymbol{\beta}^*\|_{\mathbf{H}} < u_{\mathrm{lev}}\sigma_{\min}^{3/2}/L\}$ in place of $U = \{\boldsymbol{\beta}\colon \|\boldsymbol{\beta} - \boldsymbol{\beta}^*\|_{\mathbf{H}} < (\rho_{\max}d_{\mathrm{eff}}\sigma_{\min}/m)^{3/2}/L\}$, and the corresponding proof is omitted for brevity.

**Corollary F.5** (Local convergence of SSN using scalar debiasing under approximate leverage ). *Under the settings and notations of Theorem 4.3, for any random sampling scheme with sampling distribution $\pi_i \in [\ell_i^{\mathbf{C}}(\check{\beta}_t)/(d_{\mathrm{eff}}(\check{\beta}_t)\rho_{\max}(\check{\beta}_t)), \ell_i^{\mathbf{C}}(\check{\beta}_t)/(d_{\mathrm{eff}}(\check{\beta}_t)\rho_{\min}(\check{\beta}_t))]$ as in Corollary 3.4, there exists a neighborhood $U$ of $\beta^*$ such that the* de-biased *SSN iteration $\check{\beta}_{t+1} = \check{\beta}_t - \mu_t\left(\frac{m}{m-d_{\mathrm{eff}}(\check{\beta}_t)}\mathbf{A}(\check{\beta}_t)^\top\mathbf{S}_t^\top\mathbf{S}_t\mathbf{A}(\check{\beta}_t) + \mathbf{C}(\check{\beta}_t)\right)^{-1}\mathbf{g}_t$ starting from $\check{\beta}_0 \in U$ satisfies, for $U = \{\boldsymbol{\beta}\colon \|\boldsymbol{\beta} - \boldsymbol{\beta}^*\|_{\mathbf{H}} < u_{\mathrm{lev}}\sigma_{\min}^{3/2}/L\}$, step size $\mu_t = 1 - \frac{\rho_{\max}}{m/d_{\mathrm{eff}}+\rho_{\max}}$, and $m \geq C\rho_{\max}d_{\mathrm{eff}}\log(d_{\mathrm{eff}}T/\delta)$, that*

$$\left(\mathbb{E}_\zeta\left[\frac{\|\check{\beta}_T - \beta^*\|_{\mathbf{H}}}{\|\check{\beta}_0 - \beta^*\|_{\mathbf{H}}}\right]\right)^{1/T} \leq \frac{\rho_{\max}d_{\mathrm{eff}}}{m}(1 + \epsilon + \epsilon_\rho),$$

*holds for $\epsilon = O\left(\sqrt{\rho_{\max}d_{\mathrm{eff}}/m}\right), \epsilon_\rho = \max\{\rho_{\min}^{-1} - 1, 1 - \rho_{\max}^{-1}\}$ and conditioned on an event $\zeta$ that happens with probability at least $1 - \delta$. Here, $u_{\mathrm{lev}} = (\rho_{\max}d_{\mathrm{eff}}/m)^{3/2} + \epsilon_\rho\rho_{\max}d_{\mathrm{eff}}/m$, $\sigma_{\min}$ is the smallest singular value of $\mathbf{H} \equiv \mathbf{A}(\beta^*)^\top\mathbf{A}(\beta^*) + \mathbf{C}(\beta^*)$, $d_{\mathrm{eff}} = \max_{1\leq t\leq T}d_{\mathrm{eff}}(\check{\beta}_t)$ with $\ell_i^{\mathbf{C}}(\check{\beta}_t)$ and $d_{\mathrm{eff}}(\check{\beta}_t)$ the leverage scores and effective dimension of $\mathbf{A}(\check{\beta}_t)$ given $\mathbf{C}(\check{\beta}_t)$, respectively, $\rho_{\max} = \max_{1\leq t\leq T}\rho_{\max}(\check{\beta}_t)$ and $\rho_{\min} = \min_{1\leq t\leq T}\rho_{\min}(\check{\beta}_t)$ with $\rho_{\max}(\check{\beta}_t)$ and $\rho_{\min}(\check{\beta}_t)$ the (max and min) importance sampling approximation factors (of $\mathbf{A}(\check{\beta}_t)$ given $\mathbf{C}(\check{\beta}_t)$) in Definition 2.3.*

Similarly, for SRHT, applying the scalar debiasing $m/(m - d_{\text{eff}})$ to SSN also leads to a slightly weaker local convergence than that in Theorem 4.3 but offers greater computational efficiency. This follows readily by recalling Corollary 3.7 and Corollary F.3, and adapting the proof of Theorem 4.3 with a neighborhood $U = \{\boldsymbol{\beta}\colon \|\boldsymbol{\beta} - \boldsymbol{\beta}^*\|_{\mathbf{H}} < u_{\text{SRHT}}\sigma_{\min}^{3/2}/L\}$ instead of $U = \{\boldsymbol{\beta}\colon \|\boldsymbol{\beta} - \boldsymbol{\beta}^*\|_{\mathbf{H}} < (\rho_{\max}d_{\text{eff}}\sigma_{\min}/m)^{3/2}/L\}$, and we omit the detailed proof for brevity.

**Corollary F.6** (Local convergence of SSN for SRHT using scalar debiasing). *Under the settings and notations of Theorem 4.3, for $\tilde{\mathbf{A}}_{\text{SRHT}}(\check{\boldsymbol{\beta}}_t) = \mathbf{S}_t\mathbf{H}_n\mathbf{D}_n\mathbf{A}(\check{\boldsymbol{\beta}}_t)/\sqrt{n} \in \mathbb{R}^{m\times n}$ the SRHT of $\mathbf{A}$ as in Definition 3.6, there exists a neighborhood $U$ of $\boldsymbol{\beta}^*$ such that the* de-biased *SSN iteration $\check{\boldsymbol{\beta}}_{t+1} = \check{\boldsymbol{\beta}}_t - \mu_t \left(\frac{m}{m-d_{\text{eff}}(\check{\boldsymbol{\beta}}_t)}\tilde{\mathbf{A}}_{\text{SRHT}}^{\top}(\check{\boldsymbol{\beta}}_t)\mathbf{S}_t^{\top}\mathbf{S}_t\tilde{\mathbf{A}}_{\text{SRHT}}(\check{\boldsymbol{\beta}}_t) + \mathbf{C}(\check{\boldsymbol{\beta}}_t)\right)^{-1}\mathbf{g}_t$ starting from $\check{\boldsymbol{\beta}}_0 \in U$ satisfies, for $U = \{\boldsymbol{\beta}\colon \|\boldsymbol{\beta} - \boldsymbol{\beta}^*\|_{\mathbf{H}} < u_{\text{SRHT}}\sigma_{\min}^{3/2}/L\}$, step size $\mu_t = 1 - \frac{\rho_{\max}}{m/d_{\text{eff}}+\rho_{\max}}$, $m \geq C\rho_{\max}d_{\text{eff}}\log(d_{\text{eff}}T/\delta)$, and $Tn\exp(-d_{\text{eff}}) < \delta < Tm^{-3}$, that*

$$\left(\mathbb{E}_\zeta\left[\frac{\|\check{\boldsymbol{\beta}}_T - \boldsymbol{\beta}^*\|_{\mathbf{H}}}{\|\check{\boldsymbol{\beta}}_0 - \boldsymbol{\beta}^*\|_{\mathbf{H}}}\right]\right)^{1/T} \leq \frac{\rho_{\max}d_{\text{eff}}}{m}(1 + \epsilon), \tag{44}$$

*holds for $\epsilon = O\left(\sqrt{\rho_{\max}d_{\text{eff}}/m} + \rho_{\max}^{-1}\sqrt{d_{\text{eff}}^{-1}\log(nT/\delta)}\right)$ and conditioned on an event $\zeta$ that happens with probability at least $1 - \delta$. Here, $u_{\text{SRHT}} = (\rho_{\max}d_{\text{eff}}/m)^{3/2} + \sqrt{d_{\text{eff}}\log(nT/\delta)}/m$, $\sigma_{\min}$ is the smallest singular value of $\mathbf{H} \equiv \mathbf{A}(\boldsymbol{\beta}^*)^{\top}\mathbf{A}(\boldsymbol{\beta}^*) + \mathbf{C}(\boldsymbol{\beta}^*)$, $d_{\text{eff}} = \max_{1\leq t\leq T} d_{\text{eff}}(\check{\boldsymbol{\beta}}_t)$ with $\ell_i^{\mathbf{C}}(\check{\boldsymbol{\beta}}_t)$ and $d_{\text{eff}}(\check{\boldsymbol{\beta}}_t)$ the leverage scores and effective dimension of $\mathbf{H}_n\mathbf{D}_n\mathbf{A}(\check{\boldsymbol{\beta}}_t)/\sqrt{n}$ given $\mathbf{C}(\check{\boldsymbol{\beta}}_t)$, respectively, $\rho_{\max} = \max_{1\leq t\leq T}\rho_{\max}(\check{\boldsymbol{\beta}}_t)$ with $\rho_{\max}(\check{\boldsymbol{\beta}}_t)$ the max importance sampling approximation factor (of $\mathbf{H}_n\mathbf{D}_n\mathbf{A}(\check{\boldsymbol{\beta}}_t)/\sqrt{n}$ given $\mathbf{C}(\check{\boldsymbol{\beta}}_t)$) in Definition 2.3.*

*Remark* F.7 (Comparison between Corollary F.6 and Lacotte et al. (2021, Theorem 2)). Using Corollary 3.7 and Corollary F.3 with $\nu = 0$, it follows, by adapting the proof of Theorem 4.3, that for a number $m > Cd_{\text{eff}}$ of samples, the linear convergence rate in Corollary F.6 holds *but* with a right-hand side term of $O(1)$ in (44). That is similar to the *problem-dependent* linear convergence rate obtained in e.g., Lacotte et al. (2021, Theorem 2) for self-concordant $f$, $\Phi$, and $F$ in (7). Notably, using a larger number of samples $m$ as in Corollary F.6 (than $\Theta(d_{\text{eff}})$), the (linear) convergence rate in Corollary F.6 becomes dependent on $\rho_{\max}d_{\text{eff}}/m$, whereas the linear or quadratic rates stated in Lacotte et al. (2021, Theorem 2) cannot be characterized using $n$, $d_{\text{eff}}$ or $m$.

## G. Additional Numerical Experiments and Implementation Details

Here, we provide in Appendix G.1 and Appendix G.2 implementation details of the numerical experiments on SSN in Section 5, and then in Appendix G.3 additional numerical results on the inversion bias under approximate versus exact leverage score sampling.

### G.1. Sketching Matrices and Step Size

For a data matrix $\mathbf{A} \in \mathbb{R}^{n\times d}$, the approximate ridge leverage scores and leverage scores are computed using the method provided in Drineas et al. (2012) and Cohen et al. (2017). The LESS-uniform sketching matrix is constructed as in Dereziński et al. (2021a, Section E.1). Further implementation details, including those for the SRHT, are available in the public repository provided by Dereziński et al. (2021a) at `https://github.com/lessketching/newtonsketch`. Additionally, the Shrinkage Leverage Score sampling probability is formulated by combining uniform sampling probability and approximate leverage score sampling probability in equal proportions.

In our experiments, the first-order methods, Gradient Descent and Stochastic Gradient Descent, are used with a fixed step size. As done in Dereziński et al. (2021a), second-order methods, specifically Sub-sampled Newton and Newton Sketch with Less-uniform Sketches, employ step sizes that are dynamically adjusted using a line search algorithm based on the Armijo condition (Bonnans et al., 2006, Chapter 3).

For a fair comparison of the "convergence-complexity" trade-off across different optimization methods, the time reported in Figures 2 and 3 include the time for input data pre-processing, e.g., the computation of exact or approximate leverage scores, and Walsh–Hadamard transform, as well as the computational overhead associated with the sketching process. And we fix in Figures 2 and 3 the ridge regularization parameter to $\lambda = 10^{-2}$ for both MNIST and CIFAR-10 data in Section 5.

### G.2. Datasets

For MNIST data matrix, we have $n = 2^{13}$ and $d = 2^7$, and for CIFAR-10 data, we have $n = 2^{14}$ and $d = 2^8$. We use torchvision.transforms from PyTorch to pre-process each image. We divide the ten classes of MNIST and CIFAR-10 datasets into two groups, assigning them labels of $-1$ and $+1$, respectively.

### G.3. Inversion Bias for Exact versus Approximate Leverage Score Sampling under Scalar Debiasing

In this section, we present empirical experiments to compare the inversion bias of exact versus approximate leverage score sampling under scalar debiasing $m/(m - d_{\text{eff}})$, as shown in Corollary 3.4 and discussed in Remark 3.5, for both MNIST and CIFAR-10 data.

Figure 4 depicts the inversion bias (with and without the scalar debiasing $m/(m - d_{\text{eff}})$) measured in spectral norm: $\|\mathbf{H}^{1/2}(\mathbb{E}[(\frac{m}{m-d_{\text{eff}}}\mathbf{A}^\top\mathbf{S}^\top\mathbf{S}\mathbf{A} + \lambda\mathbf{I}_d)^{-1}] - (\mathbf{A}^\top\mathbf{A} + \lambda\mathbf{I}_d)^{-1})\mathbf{H}^{1/2}\|$ and $\|\mathbf{H}^{1/2}(\mathbb{E}[(\mathbf{A}^\top\mathbf{S}^\top\mathbf{S}\mathbf{A} + \lambda\mathbf{I}_d)^{-1}] - (\mathbf{A}^\top\mathbf{A} + \lambda\mathbf{I}_d)^{-1})\mathbf{H}^{1/2}\|$, $\mathbf{H} = \mathbf{A}^\top\mathbf{A} + \lambda\mathbf{I}_d$ ) as a function of the sketch size $m$, using the following random sampling schemes:

1. **RLev**: Exact $\lambda$-ridge leverage score sampling with scalar debiasing $m/(m - d_{\text{eff}})$.

2. **NO-RLev**: Standard exact $\lambda$-ridge leverage score sampling *without* scalar debiasing $m/(m - d_{\text{eff}})$.

3. **ARLev**: Approximate $\lambda$-ridge leverage score sampling (identical to that used in Section 5) with scalar debiasing $m/(m - d_{\text{eff}})$, where the approximate ridge leverage scores $\hat{\ell}_i^{\mathbf{C}} \approx \|\mathbf{e}_i^\top\mathbf{A}(\mathbf{A}^\top\mathbf{S}_1^\top\mathbf{S}_1\mathbf{A} + \lambda\mathbf{I}_d)^{-1/2}\|^2$ are computed using sparse Johnson Lindenstrauss transform (SJLT) (Clarkson & Woodruff, 2017) $\mathbf{S}_1 \in \mathbb{R}^{m_1 \times n}$ of size $m_1$.

4. **NO-ARLev**: Standard approximate $\lambda$-ridge leverage score sampling without scalar debiasing $m/(m - d_{\text{eff}})$.

5. **DARLev**: A more efficient double-sketches variant of approximate $\lambda$-ridge leverage score sampling, together with scalar debiasing $m/(m - d_{\text{eff}})$, where the approximate ridge leverage scores $\hat{\ell}_i^{\mathbf{C}} \approx \|\mathbf{e}_i^\top\mathbf{A}((\mathbf{A}^\top\mathbf{S}_1^\top\mathbf{S}_1\mathbf{A} + \lambda\mathbf{I}_d)^{-1/2}\mathbf{S}_2^\top)\|^2$ are constructed using two SJLT matrices $\mathbf{S}_1 \in \mathbb{R}^{m_1 \times n}$ and $\mathbf{S}_2 \in \mathbb{R}^{m_2 \times d}$, with $m_2 < m_1$, see (Drineas et al., 2012; Cohen et al., 2017).

6. **NO-DARLev**: As **DARLev** but without the scalar debiasing $m/(m - d_{\text{eff}})$.

We observe from Figure 4 that the inversion bias of all random sampling methods consistently decreases as $m$ increases. Comparing solid to dashed lines, we see that the proposed scalar debiasing effectively reduces the inversion bias by a significant margin. We also find that **RLev** yields a lower inversion bias compared to its approximate counterparts **ARLev** and **DARLev**, under the scalar debiasing. These results corroborate the conclusions in Corollary 3.4 and Remark 3.5.

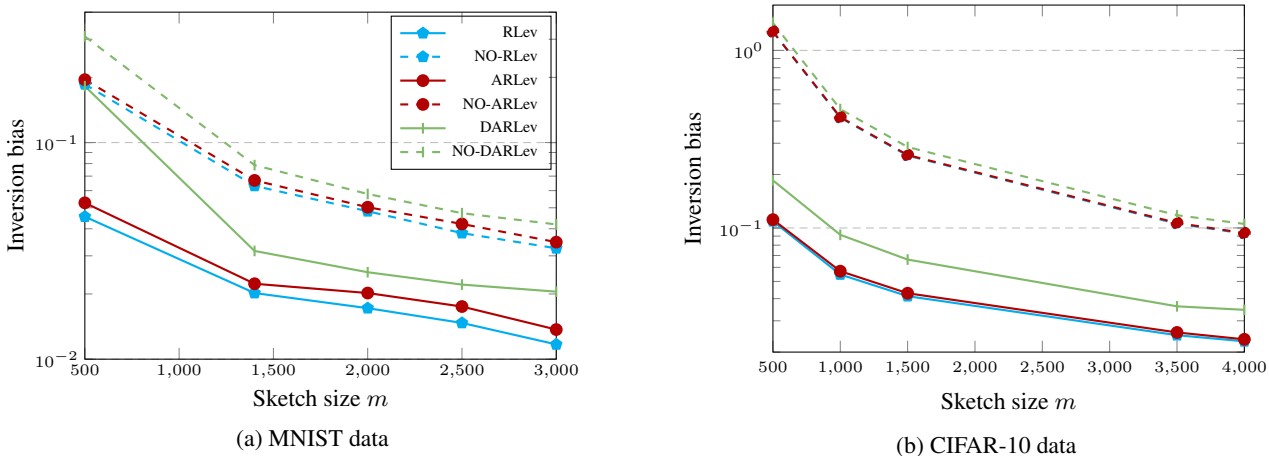

(a) MNIST data

(b) CIFAR-10 data

Figure 4: Inversion bias as a function of the sketch size $m$, for various sampling methods on both MNIST and CIFAR-10 data, with ridge parameter $\lambda = 10^{-1}$ for MNIST data and $\lambda = 10^{-6}$ for CIFAR-10 data. The results are averaged over 500 independent runs.

