# OpenReview forum: "Fundamental Bias in Inverting Random Sampling Matrices with Application to Sub-sampled Newton"
_ICML.cc/2025/Conference — ICML 2025 oral_

### Official Review · Reviewer_vs1h · 2025-03-01

**Overall Recommendation:** 4

**Summary:**

This paper looks at how inversion bias (where the inverses of random sketches of a matrix are not unbiased) can be corrected for several random sampling methods. The paper gives an outline of how this can be done, and gives bounds for an $(\epsilon,\delta)$ estimator in Theorem 3.1 The paper makes a commentary on Derezinski's bounds (if I understand this correctly, the paper's derivation uses an alternative proof which beats the bounds given by Derezinski et al (2021a;b)). The paper then gives practical applications of these bounds to de-biased sub-sampled Newton with a better convergence rate (or convergence-complexity) trade-off.

**Claims And Evidence:**

Each claim is supported by proofs in the Appendix. While I did not have the opportunity to check each line thoroughly, each part of the (proofs) in the appendix gives the outline of the proof, and clear explanation for each subsequent step. This makes it straightforward for readers with more detailed background knowledge to follow and check.

**Essential References Not Discussed:**

No

**Experimental Designs Or Analyses:**

I would be interested to see the boxplots of the relative errors over the 30 independent runs in Fig 2, and 10 independent runs in Fig 3 for both Newton-LESS and SSN-ARLev, as the relative error for Newton-LESS and SSN-ARLev seem close to each other as the sketch size $m$ increases. This is not to say that the experimental design / analyses is flawed, but I'm curious if the relative errors are skewed in one direction or equally symmetric about the solid lines in both Figures.

**Methods And Evaluation Criteria:**

The datasets are standard benchmark datasets (MNIST and CIFAR-10).

I do have a question about lines 400-402: "Figures 2 and 3 do not include the time for input data pre-processing......"
What is the relative magnitude of the time for input data pre-processing? That is to say, if the relative magnitudes of pre-processing is greater than the reduction in wall clock time for Newton-LESS / SSN-ARLev, then I don't think wall-clock time would be a meaningful metric.

**Other Comments Or Suggestions:**

I will admit I am not very familiar with this paper, so I am basing my review solely on the clarity of the writing, some papers I referred to from the references in this paper, and questions from the experimental side.  But what I like about this paper is that the proofs are "reproducible", in the sense that the very detailed appendix explains what is happening in each line of the proof, as well as a general outline, hence anyone with sufficient background should be able to check what is happening at each step - unfortunately I am not the best person to do this.

Hence, please take this review with a grain of salt.

**Other Strengths And Weaknesses:**

N/A

**Questions For Authors:**

Please see above for my questions

**Relation To Broader Scientific Literature:**

The contributions extend the work of randomized numerical algebra, where the paper looks at reducing the inversion bias of random sketches for random sampling methods. Based on the theorems and discussion in the main paper, the paper builds upon and extends (and critiques?) the results in the papers Derezinski 2021a;b using leverage based sampling.

**Theoretical Claims:**

The proofs look reasonable, however I did not carefully check the correctness of them.

---

> ### Author Rebuttal · Authors · 2025-03-31
>
> We thank the reviewer for his/her constructive and thoughtful comments. Please find our point-by-point responses below.
>
> **Methods And Evaluation Criteria**:
>
> 	The datasets are standard benchmark datasets (MNIST and CIFAR-10).
> 	I do have a question about lines 400-402: "Figures 2 and 3 do not include the time for input data pre-processing......" What is the relative magnitude of the time for input data pre-processing? That is to say, if the relative magnitudes of pre-processing is greater than the reduction in wall clock time for Newton-LESS / SSN-ARLev, then I don't think wall-clock time would be a meaningful metric.
>
> **Reply**:
> We thank the reviewer for the constructive question (which is in fact shared by Reviewer GJpF).
> It is worth noting that various second-order baselines in Figures 2 and 3, including Newton-LESS, ARLev, ALev, SLev, and SRHT, **all** require preprocessing of the data.
>
> Specifically, Newton-LESS requires a Hadamard transform preprocessing with a complexity of $O(nd \log n)$, see [Derezinski2021a].
> To ensure a fair comparison, we treat SRHT as a uniform sampling method that undergoes the same random Hadamard transform preprocessing as Newton-LESS.
> The leverage-based sampling methods, ARLev, ALev, and SLev, require computing approximate leverage scores before constructing the sketches $\check{\mathbf{S}}\mathbf{A}$ via random sampling, with complexity $O(\text{nnz}(\mathbf{A}) \log n + d^3 \log d)$.
>
> As such, the preprocessing time are in fact **comparable** across these methods, and it makes sense to exclude the time for the Hadamard transform and leverage score computation ensures a fair comparison.
> Considering the computational overhead associated with the sketching process itself (rather than the preprocessing steps) allows for a clearer illustration of the "convergence-complexity" trade-off across different stochastic optimization methods.
>
> We have reproduced Figures 1 and 2 by adding (for all methods) the preprocessing time, and observed **a similar trends**.
> In particular, SSN-ARLev achieves a significantly lower overall runtime compared to Newton-LESS and establishes better complexity-convergence trade-off than other baselines.
> These additional numerical results and discussions will be included in the revision.
>
> Additionally, for certain random Newton-type solvers, preprocessing is done **only once** at the beginning of the iteration and then reused in subsequent iterations.
> For instance, in [Li2024], distributed least squares regression based on federated Newton iterations only performs data preprocessing (such as the Hadamard transform and computing leverage scores) once, and then reuse the obtained results alongside the optimization procedure.
> In such cases, the preprocessing time becomes negligible compared to the iteration computation and communication costs.
> Thus, we believe wall-clock time remains a meaningful metric.
>
>
> [Derezinski2021a] Derezinski M, Lacotte J, Pilanci M, et al. Newton-LESS: Sparsification without trade-offs for the sketched Newton update. Advances in Neural Information Processing Systems, 2021, 34: 2835-2847.
>
> [Li2024] Li J, Liu Y, Wang W. Fedns: A fast sketching newton-type algorithm for federated learning. Proceedings of the AAAI Conference on Artificial Intelligence. 2024, 38(12): 13509-13517.
>
>
> **Experimental Designs Or Analyses**:
>
> 	I would be interested to see the boxplots of the relative errors over the 30 independent runs in Fig 2, and 10 independent runs in Fig 3 for both Newton-LESS and SSN-ARLev, as the relative error for Newton-LESS and SSN-ARLev seem close to each other as the sketch size m increases. This is not to say that the experimental design / analyses is flawed, but I'm curious if the relative errors are skewed in one direction or equally symmetric about the solid lines in both Figures.
>
> **Reply**:
> We thank the reviewer for the insightful comment.
> Our theoretical results say that the convergence rate of SSN-ARLev can approximate, but **not** exceed, the convergence rate of Newton-LESS, see Theorem 4.3.
> Moreover, as $m$ increases, the convergence rate of the de-biased sub-sampled Newton method approaches that of Newton-LESS.
> Therefore, it is reasonable to see in the experimental results that the relative errors for both Newton-LESS and SSN-ARLev becoming increasingly similar as $m$ increases.
>
> Our contribution in fact relies in showing that SSN-type methods can indeed achieve (local) **problem-independent convergence rates** (which, to the best of our knowledge, has **never** been established for SSN-type methods) that
>
> (1) can be significantly faster than *standard* SSN; and
>
> (2) matches that of Newton-LESS, but with significantly reduced computation complexity, and thus better complexity-convergence trade-off, as confirmed by Figures 1 and 2.
>
> As suggested by the reviewer, we will include boxplots of the relative errors for both methods also converge as $m$ increases.

---

> > ### Comment · Reviewer_vs1h · 2025-04-01
> >
> > Thank you for the detailed rebuttal. I will maintain my score.

---

### Official Review · Reviewer_LRsm · 2025-03-08

**Overall Recommendation:** 5

**Summary:**

The paper studies the row-sampling of matrices in the context of approximating the inverse. Specifically, given an $n \times d$ matrix $A$ and a row-sampling matrix $S$ with $m$ rows, sketching is commonly used to approximate $A^T A$ by $A^T S^T S A$. Prior work by Derezinski et al. showed that the inverse of the (rescaled) sketched matrix, $(\frac{m}{m-d} A^T S^T S A)^{-1}$ is a ($1+\epsilon$)-approximation to $(A^T A)^{-1}$ with only $m = O(d + \sqrt{d}/\epsilon)$. In contrast, a direct approximation of $A^T A$ typically requires $m = d \log d / \epsilon^2$ rows.

This paper generalizes the result of Derezinski et al. by studying how well $(A^T S^T SA + C)^{-1}$ approximates $(A^T A + C)^{-1}$, where $C$ is a PSD matrix. When $C$ is the identity matrix, this encompasses ridge leverage scores. This paper shows that $(A^T S^T SA + C)^{-1}$ is an ($1+\epsilon$)-approximation to (A^T D A + C)^{-1} for some diagonal matrix D depending on ridge leverage scores and sampling probabilities, when the number of sampled rows $m$ satisfies $m \gtrsim d \log(m d) + d/\epsilon^{2/3}$. When $C = 0$ and the sampling probabilities are exactly leverage scores, $D = (m-d)/m \cdot I$, recovering the result of Derezinski et al. The paper then obtains results for some specific constructions of S, and applies the new results to the update step of Newton’s method, where it is used to approximate the inverse of the Hessian, and evaluates its performance through numerical experiments.

Overall, this submission makes a solid contribution to the study of the inverse of row-sampled matrices, which also benefits the acceleration of optimization methods. I recommend acceptance, provided that my questions are adequately answered.

**Claims And Evidence:**

The claims are theoretical and proofs are provided.

**Essential References Not Discussed:**

NIL

**Experimental Designs Or Analyses:**

I do not see issues in general but I do not think one can claim “improved convergence” because the convergence is not improved. The main thing seems to be improving the runtime at the same convergence. The numerical experiments also show this: the convergence rate of SSN-ARLeve looks about the same (at least the rate looks similar) as Newton-LESS in Figure 1 on both datasets.

**Methods And Evaluation Criteria:**

The numerical experiments look reasonable.

**Other Comments Or Suggestions:**

- Page 4, left column, line 204: add a comma after $m$
- Page 4, left column, line 207: as follow -> as follows
- Page 5, left column, line 264: “n, eff” -> “n and d_eff”
- Page 7, right column, line 338: It is better to say “Suppose that F, f: …”.
- Page 16, Eq.(20): A right square bracket is missing for \tilde{Z}_0.
- Page 16, Lines 862-871: to be consistent with (22), it is better to write the event in the subscript, that is, $\mathbf{1}_{\neg\zeta_3}$.
- Page 21, the long list of derivation: Once u and v are separated, there is no need to carry both u and v for supremum; it suffices to have one supremum over u and the other one over v.

**Other Strengths And Weaknesses:**

NIL

**Questions For Authors:**

Please see the sections above.

**Relation To Broader Scientific Literature:**

Research on the inverse of row-sampling matrices appears to be less developed than that on row-sampling matrices themselves. This submission is a valuable contribution to the literature on the inverse of row-sampling matrices and also makes a nice addition to the literature on using sketching to accelerate numerical methods.

**Theoretical Claims:**

I did not have time to check all the proofs. The following are a few questions:
- Page 4, left column, Line 199: what does it mean to say “subspace embeddings … ensure unbiasedness up to the same level of epsilon but are not guaranteed to remain unbiased for a smaller epsilon”?
- Page 21, inequality (a) (in the middle of the page), can you explain how you get the constant factor of 38?

---

> ### Author Rebuttal · Authors · 2025-03-31
>
> We thank the reviewer for his/her constructive and thoughtful comments. Please find our point-by-point responses below.
>
> **Theoretical Claims:**
>
> 	I did not have time to check all the proofs. The following are a few questions:
>
> 	1. Page 4, left column, Line 199: what does it mean to say “subspace embeddings … ensure unbiasedness up to the same level of epsilon but are not guaranteed to remain unbiased for a smaller epsilon”?
>
> **Reply:** Sorry for this discussion that may be misleading.
> Here, we meant that sketching matrices satisfying the subspace embedding property, with an error $\epsilon$ (say of order $\epsilon = O\left(\sqrt{\rho_{\max} d_{\mathrm{eff}} \log(d_{\mathrm{eff}}/\delta)/m}\right)$ ), can guarantee the **same** level $\epsilon$ of the inversion bias, see Lemma B.1 in the supplementary material for a proof.
> It is, however, unclear whether this (upper bound of) inversion bias is tight and/or can be made smaller with refined analysis.
> This is in fact the very motivation of the proposed  de-biased sampling matrix $\check{S}$, which, under the same sampling size, achieves a **smaller** inversion bias of order $\epsilon = O\left(\sqrt{\rho_{\max}^3 d_{\mathrm{eff}}^3 / m^3}\right)$, see Proposition 3.2, offering an improvement over the standard subspace embedding-type analysis.
>
> 	2. Page 21, inequality (a) (in the middle of the page), can you explain how you get the constant factor of 38?
>
> **Reply:** We thank the reviewer for the constructive comments.
> Regarding the constant factor of 38 in inequality (a) on Page 21, we will provide a detailed explanation.
> By combining $H^{1/2} Q_{-s} H^{1/2} \preceq 6 I_d$ and the bound $|| H^{\frac12} \widetilde{H}^{-1} H^{\frac{1}{2}}\ || \leq 2$, along with $||v || = 1$ and $||u || = 1$, we can derive the following two inequalities:
> $$
> \sum^n_{j=1} u^\top \widetilde{H}^{\frac12} Q_{-s} a_{j} a_{j}^\top Q_{-s} \widetilde{H}^{\frac12} u=u^\top \widetilde{H}^{\frac12} Q_{-s} A^\top A Q_{-s} \widetilde{H}^{\frac12} u \leq 36
> $$
> and
> $$
> \sum^n_{j=1} v^\top \widetilde{H}^{-\frac12} a_{j} a_{j}^\top \widetilde{H}^{-\frac{1}{2}} v = v^\top \widetilde{H}^{-\frac12} A^\top A \widetilde{H}^{-\frac12} v  \leq 2.
> $$
>
> Adding these two results gives the constant factor of 38. We will provide these proof details in Section D.2 of the supplementary material in the revised version for clarification.
>
> **Experimental Designs Or Analyses:**
>
> 	I do not see issues in general but I do not think one can claim “improved convergence” because the convergence is not improved. The main thing seems to be improving the runtime at the same convergence. The numerical experiments also show this: the convergence rate of SSN-ARLev looks about the same (at least the rate looks similar) as Newton-LESS in Figure 1 on both datasets.
>
> **Reply:**
> We greatly appreciate the reviewer's helpful feedback and agree that the claim of "improved convergence" is indeed a bit misleading.
> As discussed in Section 4 of the submission, the proposed de-biased SSN improves the local convergence rate over **standard** sub-sampled Newton method, to achieve problem-independent convergence rates (that, to the best of our knowledge, has **never** been established for SSN).
> It is worth noting that the theoretical convergence rate of de-biasd SSN can approximately match, but **not** exceed, that of Newton-LESS (that is a closely-related, but different approach replying on random projection), but will **significantly reduced** computational complexity.
>
> In particular, the proposed SSN-ARLev (that replies on random sampling) offers significant computational efficiency advantages over Newton-LESS, and thus outperforms Newton-LESS in terms of the complexity-convergence trade-off, see confirmed by Figures 1 and 2.
>
> While the numerical experiments in Figure 1 suggest that SSN-ARLev may offer **even improved** convergence in certain practical scenarios over the Newton-LESS approach, we are unable yet provide a theoretical proof of such superiority in convergence rate.
>
> Additionally, we believe that combining the proposed de-biased approach with more efficient sub-sampled newton algorithms, such as the adaptive SSN-type algorithm proposed by [Lacotte2021], could further  accelerate convergence.
> However, exploring this is beyond the scope of the present work.
>
>
> [Lacotte2021] Lacotte J, Wang Y, Pilanci M. Adaptive newton sketch: Linear-time optimization with quadratic convergence and effective hessian dimensionality. International Conference on Machine Learning. PMLR, 2021: 5926-5936.
>
> **Other Comments Or Suggestions:**
>
> **Reply:**  We thank the reviewer again for the thorough and thoughtful review.  All minor issues raised will be addressed in the revised version of the paper.

---

> > ### Comment · Reviewer_LRsm · 2025-04-02
> >
> > Thanks for the clarifications.
> >
> > 1. Please rephrase the sentence to mention explicitly that the subspace embedding has an error $\epsilon$ and the unbiasedness refers to the inversion bias. The current phrasing is mostly unclear.
> >
> > 2. It might be good to mention at some early point in the proof that $A^\top A\preceq H$ and $\tilde{H}\preceq H$. The inequality is then clear.
> >
> > 3. Please make your point clear in the revised version what the "improved convergence" actually means. Also, I do not see from Figure 1 that SSN-ARLev has a clear better convergence than Newton-LESS. They are almost the same.

---

> > > ### Author Response · Authors · 2025-04-02
> > >
> > > We sincerely appreciate the reviewer’s timely feedback and constructive comments. Below are our responses to the points raised:
> > > - In the revised paper, we will explicitly mention that the subspace embedding introduces an error $\epsilon$, and clarify that the term *“unbiasedness”* specifically refers to the *inversion bias*. The corresponding explanation will be  added after Definition 2.6.
> > > - Furthermore, we will include the technical details that $ \mathbf{A}^\top \mathbf{A}\preceq \mathbf{H}$ and $ \widetilde{\mathbf{H}}\preceq \mathbf{H}$  in Section D.2 of the supplementary material to make the derivation easier to follow.
> > > - We appreciate the reviewer’s comment and apologize again for the lack of clarity in the original claim of *“improved convergence”*. In the revised version, we will clarify this claim at the end of Section 4. Specifically, we will state that compared to the **standard** sub-sampled Newton method, our de-biased SSN method *improves* the local convergence rate; while its theoretical rate is comparable to, but **not better** than, that of Newton-LESS, it **significantly lowers** the computational cost, resulting in a more **favorable complexity–convergence** trade-off.

---

### Official Review · Reviewer_GJpF · 2025-03-12

**Overall Recommendation:** 4

**Summary:**

The paper studies debiasing scheme for sub-sampled random matrix sketching. It improves prior work by providing a novel random sampling method which has better convergence result. Experiments are presented to corroborate theoretic results.

**Claims And Evidence:**

Yes

**Essential References Not Discussed:**

Newton Meets Marchenko-Pastur: Massively Parallel Second-Order Optimization with Hessian Sketching and Debiasing (ICLR 2025, https://arxiv.org/abs/2410.01374)

Optimal Shrinkage for Distributed Second-Order Optimization (ICML 2023, https://arxiv.org/abs/2402.01956)

**Experimental Designs Or Analyses:**

Not in full details

**Methods And Evaluation Criteria:**

Yes

**Other Comments Or Suggestions:**

n/a

**Other Strengths And Weaknesses:**

The paper is well-written, theoretically sound, and the experiments are clearly presented. I personally appreciate that the authors provide a lot of comments and discussions / remarks linking to prior findings, pointing out what's the similarity and what's the difference, which offers a comprehensive perspective for readers.

**Questions For Authors:**

1. In Definition 2.3, $\rho_\min=\min_{1\leq i\leq n} \ell_i^C/(\pi_id_{\text{eff}})$, isn't $\pi_id_{\text{eff}}=\ell_i^C$ and thus cancels the numerator? Also, how does this sampling factor related to $S$, I think it's just how much we weigh each row of $A$ when we sample, but the Definition sounds like $\rho_\min$ is also related to $S$?

2. In Figure 2, the authors present numerical results showing the proposed method is much more efficient compare to Newton-LESS, I feel it would be more helpful if the authors can discuss more about the baseline and what's the main computation overhead there.

3. Also in the experiments, the authors mention they exclude the time for computation of exact or approximate leverage score, does is mean the main workload of computing $\check S$ is removed? Is there any justification for such exclusion, i.e., is it a fair thing to do?

**Relation To Broader Scientific Literature:**

The key contribution is mainly a new random sampling scheme achieving better debiasing result, and connection between the new scheme and prior method (the scalar debiasing scheme, and phase transition, etc). It's a more refined version of prior debiasing literature.

**Theoretical Claims:**

Not in full details

---

> ### Author Rebuttal · Authors · 2025-03-31
>
> We thank the reviewer for his/her constructive and thoughtful comments. Please find our point-by-point responses below.
>
> **Essential References Not Discussed**:
>
> 	Newton Meets Marchenko-Pastur: Massively Parallel Second-Order Optimization with Hessian Sketching and Debiasing (ICLR 2025, https://arxiv.org/abs/2410.01374)
>
> 	Optimal Shrinkage for Distributed Second-Order Optimization (ICML 2023, https://arxiv.org/abs/2402.01956)
>
> **Reply**:  We thank the reviewer for pointing to the two helpful references.
> They will be included and discussed in the revised version of the paper.
>
>
> **Questions For Authors**:
>
> 	1. In Definition 2.3,$ \rho_{\min} \equiv \min_{1\leq i \leq n}\ell_i^{C}/(\pi_i d_{\mathrm{eff}})$, isn't $\pi_i d_{\mathrm{eff}}=\ell_i^{C}$ and thus cancels the numerator? Also, how does this sampling factor related to  $S$, I think it's just how much we weigh each row of $A$ when we sample, but the Definition sounds like $\rho_{\min} $ is also related to  $S$?
>
> **Reply**:  We thank the reviewer for this question.
> In Definition 2.3, we define $\rho_{\min} \equiv \min_{1 \leq i \leq n} \frac{\ell_i^{\mathbf{C}}}{\pi_i d_{\mathrm{eff}}}$.
> While for **exact (ridge) leverage score** (when the random sampling probability $\pi_i$ is chosen according to $\ell_i^{\mathbf{C}}/d_{\mathrm{eff}}$), one has $\pi_i d_{\mathrm{eff}} = \ell_i^{\mathbf{C}}$ and thus cancels the numerator, this is **not** true for generic random sampling methods.
>
> The parameter $\rho_{\min}$ is introduced to capture the interaction between the choice of sampling probabilities $\pi_i$ and the data structure of $\mathbf{A}$ (e.g., its leverage scores) and how it affects the associated random sampling inversion bias.
>
>
> 	2. In Figure 2, the authors present numerical results showing the proposed method is much more efficient compare to Newton-LESS, I feel it would be more helpful if the authors can discuss more about the baseline and what's the main computation overhead there.
> 	3. Also in the experiments, the authors mention they exclude the time for computation of exact or approximate leverage score, does is mean the main workload of computing  $\check S$ is removed? Is there any justification for such exclusion, i.e., is it a fair thing to do?
>
> **Reply**:
> We thank  the reviewer for the comments.
> It is worth noting that various second-order baselines in Figures 2 and 3, including Newton-LESS, ARLev, ALev, SLev, and SRHT, **all** require preprocessing of the data.
>
> Specifically, as performed in [Derezinski2021a], Newton-LESS first performs the Hadamard transform preprocessing on the data, and then generates LESS-uniform sketching matrix $\mathbf{S}$ to get the sketch $\tilde{\mathbf{A}} = \mathbf{S} \mathbf{A}$.
> The computational costs of these two steps are $O(nd \log n)$ and $O(msd)$, where $s$ represents the sparsity level of (each row in) $\mathbf{S}$.  To ensure a fair comparison, we treat SRHT as a uniform sampling method that undergoes the same random Hadamard transform preprocessing as Newton-LESS.
>
> The three leverage-based sampling methods, ARLev, ALev, and SLev, all involve the computation of approximate leverage scores and the sampling process, and has complexities of $O(\text{nnz}(\mathbf{A})\log n + d^3 \log d)$ and $O(m + md)$, respectively.
>
> Thus, the preprocessing time is in fact **comparable** across all baseline methods, and by
> excluding the time for the Hadamard transform and leverage score computation, we ensure a **fair comparison** when evaluating these methods.
> Considering the computational overhead associated with the sketching process itself (rather than the preprocessing steps) allows for a clearer illustration of the "convergence-complexity" trade-off across different stochastic optimization methods.
>
> We have reproduced Figures 1 and 2 by adding (for all methods) the preprocessing time, and observed a similar trends.
> In particular, SSN-ARLev achieves a significantly lower overall runtime compared to Newton-LESS and establishes better complexity-convergence trade-off than other baselines.
>
> These additional numerical results and discussions will be included in the revision.
>
> [Derezinski2021a] Derezinski M, Lacotte J, Pilanci M, et al. Newton-LESS: Sparsification without trade-offs for the sketched Newton update. Advances in Neural Information Processing Systems, 2021, 34: 2835-2847.

---

> > ### Comment · Reviewer_GJpF · 2025-04-02
> >
> > Thanks for the reply. I'll keep my evaluation.

---

### Official Review · Reviewer_2Kx2 · 2025-03-17

**Overall Recommendation:** 4

**Summary:**

This paper analyzes and corrects the inversion bias arising from common random sketching methods, including uniform/weighted sampling (based on leverage scores) and structured fast Johnson-Lindenstrauss transform like the sub-sampled randomized Hadamard transform (SRHT). Via a fine-grained characterization of inversion bias, it is demonstrated that for certain structured methods like (approximate) leverage-based sampling and SRHT, the bias can be effectively corrected through careful diagonal-wise rescaling. Leveraging these theoretical insights, the paper further establishes the first problem-independent local convergence rates for sub-sampled Newton methods, closely matching those obtained with dense Gaussian sketches. Empirical evaluations support these theoretical findings.

**Claims And Evidence:**

Yes, the main theoretical claims regarding the characterization and correction of inversion bias, as well as the convergence rates for sub-sampled Newton, are clearly presented. While I couldn't verify all the proofs in the appendix, with a quick scan, the proofs seem reasonable, and the intuitions provided at the beginning are helpful.  The empirical results are also convincing and support the theoretical claims.

**Essential References Not Discussed:**

To my knowledge, the paper discussed the essential references in the field.

**Experimental Designs Or Analyses:**

This is a theoretical paper with a few synthetic numerical experiments. The experiments are simple but sufficient to support the theoretical claims.

**Methods And Evaluation Criteria:**

Yes, the application of the proposed de-biasing method to sub-sampled Newton is well-aligned, providing convincing evidence for the broader applicability of the theoretical results.

**Other Comments Or Suggestions:**

Here are some minor points/questions:
1. Are the constants $C$ in the theorems, propositions, and lemmas in Section 2 and 3 the same, or at least on the same order? It would be helpful to clarify this in the paper.
2. The preliminary in Section 2 seems to be a bit long, especially given the 8-page limit of the main text. Compared to the preliminaries, some proof sketches and intuitions in the appendix seem to be more insightful and may deserve a place in the main text.

**Other Strengths And Weaknesses:**

Overall, I think this is a valuable theoretical study on inversion bias in random sketching methods. The paper is well-structured and easy to follow. The proposed de-biasing method is theoretically sound and practically useful. The application to sub-sampled Newton provides a good example of the broader applicability of the proposed de-biasing method. Some minor weaknesses are discussed below in the "Other Comments" and "Questions For Authors" sections.

**Questions For Authors:**

Is the log factors in $m$ in most of the theorems, propositions, and lemmas in Sections 2 and 3 a consequence due to independent sampling of the $m$ rows (uniformly or according to leverage scores)? If so, would it be possible to remove the log factors via dependent sampling, e.g., volume sampling, or some faster alternatives like adaptive leverage score sampling (see, e.g., [1])? It would be insightful to provide some discussions on this, even just as future directions.

[1] Cortinovis, Alice, and Daniel Kressner. "Adaptive randomized pivoting for column subset selection, DEIM, and low-rank approximation." arXiv preprint arXiv:2412.13992 (2024).

**Relation To Broader Scientific Literature:**

Yes, this work provides a fine-grained characterization of inversion bias in common random sketching methods leveraging RMT techniques. It extends the existing results in Derezinski et al. (2021a;b) on approximately unbiased sparse embedding to more common random sketching methods. Applying the resulted de-biasing method to sub-sampled Newton leads to considerably improvement compared to Lacotte & Pilanci, (2019), Derezi ́nski et al., (2021a)., etc., both theoretically in terms of the convergence guarantee and empirically.

**Theoretical Claims:**

While I couldn't verify all the proofs in the appendix, the main theoretical results in the paper seem reasonable. The proofs are well-structured, and the intuitions provided at the beginning are helpful.

---

> ### Author Rebuttal · Authors · 2025-03-31
>
> We thank the reviewer for his/her constructive and thoughtful comments. Please find our point-by-point responses below.
>
> **Other Comments Or Suggestions**:
>
> 	1. Are the constants C in the theorems, propositions, and lemmas in Section 2 and 3 the same, or at least on the same order? It would be helpful to clarify this in the paper.
>
> **Reply**:
> While the constants $C$ in the theorems, propositions, and lemmas in Sections 2 and 3 are not identical, they are all *absolute constants* independent of the dimensions $n$, $d_{\mathrm{eff}}$, and $m$.
>
>
> 	2. The preliminary in Section 2 seems to be a bit long, especially given the 8-page limit of the main text. Compared to the preliminaries, some proof sketches and intuitions in the appendix seem to be more insightful and may deserve a place in the main text.
>
> **Reply**: We thank the reviewer for this constructive comment.
> We agree that the current section of preliminaries are a bit too long and that the proof sketches and intuitions (currently in the appendix) should be moved to the main text to better highlight the connection between RMT and RandNLA.
> Due to the 8-page limit for the submission, we were unable to include them in the current version.
>
> If the paper gets accepted, we will revise the content accordingly (with the one additional page)  in the camera-ready version by incorporating the proof sketches and intuitions from Appendices D.1 and E.1 into the main text, placing them directly after Theorem 3.1 and Proposition 3.2, respectively.
>
>
> **Questions For Authors**:
>
> 	Is the log factors in m in most of the theorems, propositions, and lemmas in Sections 2 and 3 a consequence due to independent sampling of the m rows (uniformly or according to leverage scores)? If so, would it be possible to remove the log factors via dependent sampling, e.g., volume sampling, or some faster alternatives like adaptive leverage score sampling (see, e.g., [1])? It would be insightful to provide some discussions on this, even just as future directions.
> [1] Cortinovis, Alice, and Daniel Kressner. "Adaptive randomized pivoting for column subset selection, DEIM, and low-rank approximation." arXiv preprint arXiv:2412.13992 (2024).
>
> **Reply**:
> We thank the reviewer for the constructive comment and the reference.
> We agree with the reviewer's perspective.
> As shown in Section B of the supplementary materials, the subspace embedding property in Lemma 2.7 relies on the independence of the random row sampling procedure.
> This implies that the logarithmic factors in $m$ present in most of the theorems, propositions, and lemmas in Sections 2 and 3 arise from the independent sampling of the $m$ rows (including, but not limited to, uniform sampling or leverage score-based sampling).
>
> As such, it is very possible that dependent sampling methods, such as volume sampling or more efficient techniques like adaptive leverage score sampling (see, e.g., [1]), may provide tighter bounds and potentially eliminate the logarithmic factors.
> For now, we are not able to include these efficient methods in the proposed analysis framework, since our current proof approach strongly relies on the independence of random sampling.
>
> We also agree with the reviewer that these dependent sampling methods could yield better computational efficiency and/or convergence rate for SSN.
> We leave them for future work.
> We will include a discussion of these methods in the conclusion section (Section 6) of the revised version, addressing the potential benefits and challenges of using volume sampling, adaptive leverage score sampling, or other dependent sampling approaches.

---

> > ### Comment · Reviewer_2Kx2 · 2025-04-01
> >
> > I would like to thank the authors for addressing my questions and concerns. I will maintain my evaluation.

---

### Decision · Program_Chairs · 2025-05-01

**Decision:**

Accept (oral)

**Comment:**

This paper considers the problem of inversion bias for random sampling matrices, which play a central role in randomized numerical linear algebra. Though previous works have shown how to handle specific samplings instances (e.g., Gaussian projection), this paper addresses the problem for general instances of random sampling. This is a very nice and well-written paper that provides a clear overview of the previous techniques and the reasons they do not immediately extend to the more general setting, along with their novel approach to debiasing based on approximate leverage score sampling. The reviewers unanimously agree that this paper should be accepted, and I also believe it provides important contributions that should be included in the conference.